# FlowDAS: A Stochastic Interpolant-based Framework for Data Assimilation

**Siyi Chen**[*]
University of Michigan
siyiche@umich.edu

**Yixuan Jia**[*]
University of Michigan
jiayx@umich.edu

**Qing Qu**
University of Michigan
qingqu@umich.edu

**He Sun**[†]
Peking University
hesun@pku.edu.cn

**Jeffrey A. Fessler**
University of Michigan
fessler@umich.edu

## Abstract

Data assimilation (DA) integrates observations with a dynamical model to estimate states of PDE-governed systems. Model-driven methods (e.g., Kalman Filter, Particle Filter) presuppose full knowledge of the true dynamics, which is not always satisfied in practice, while purely data-driven solvers learn a deterministic mapping between observations and states and therefore miss the intrinsic stochasticity of real processes. Recently, score-based diffusion models have shown promise for DA by learning a global diffusion prior to represent stochastic dynamics. However, their one-shot generation lacks stepwise physical consistency and struggles with complex stochastic processes. To address these issues, we propose **FlowDAS**, a generative DA framework that employs stochastic interpolants to learn state transition dynamics through step-by-step stochastic updates. By incorporating observations into each transition, FlowDAS can produce stable, measurement-consistent forecasts. Experiments on Lorenz-63, Navier–Stokes super-resolution/sparse-observation scenarios, and large-scale weather forecasting——where dynamics are partly or wholly unknown—show that FlowDAS surpasses model-driven methods, neural operators, and score-based baselines in accuracy and physical plausibility. Our implementation is available at https://github.com/umjiayx/FlowDAS.

## 1 Introduction

Recovering state variables in complex dynamical systems is a fundamental problem in science and engineering. Accurate state estimation from noisy, incomplete data is critical in weather forecasting [1, 2], oceanography [3, 4], seismology [5, 6], and many other fields [7], where reliable predictions depend on understanding the underlying physics [8–10]. A representative example is fluid dynamics [11–13], where one aims to reconstruct a continuous velocity field from sparse, noisy observations governed by nonlinear, time-dependent partial differential equations (PDEs)—a task complicated by stochasticity and high dimensionality. To meet these challenges, **data assimilation** (DA) combines model forecasts with observations to produce physically consistent state estimates; developed first in atmospheric and oceanic forecasting, DA is now ubiquitous across many scientific and engineering domains [14–20].

Mathematically, a discrete-time stochastic dynamical system can be described by:

$$\boldsymbol{x}_{k+1} = \Psi(\boldsymbol{x}_k) + \boldsymbol{\xi}_k, \tag{1}$$

$$\boldsymbol{y}_{k+1} = \mathcal{A}(\boldsymbol{x}_{k+1}) + \boldsymbol{\eta}_{k+1}, \tag{2}$$

---

[*]Equal contribution. Work done while Siyi Chen was an undergraduate student at Peking University.

[†]Corresponding author. This author is affiliated with the National Biomedical Imaging Center, Peking University, Beijing and Academy for Advanced Interdisciplinary Studies, Peking University, Beijing.

39th Conference on Neural Information Processing Systems (NeurIPS 2025).

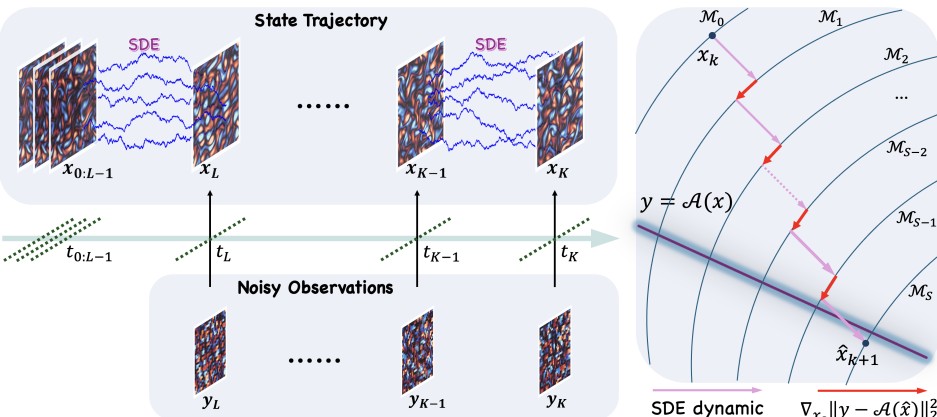

Figure 1: **An overview of FlowDAS.** We introduce a stochastic interpolant-based framework for data assimilation, named FlowDAS, to estimate states $\boldsymbol{x}_{L:K}$ from the noisy (sparse or low-resolution) observations $\boldsymbol{y}_{L:K}$. FlowDAS models the stochastic dynamics of the system with a flow-based stochastic differential equation (SDE) and incorporates the observations to improve the prediction accuracy. On the right, we show a conceptual illustration of the geometry of the process to estimate $\hat{\boldsymbol{x}}_{k+1}$ from $\boldsymbol{x}_k$. $\mathcal{M}_s$ denotes the generative manifold at interpolation step $s$. The gradient guidance $\nabla_{\boldsymbol{x}_s}\|\boldsymbol{y} - \mathcal{A}(\hat{\boldsymbol{x}})\|_2^2$ enforces observation consistency during the generation process.

where $\boldsymbol{x}_k \in \mathbb{R}^D$ is the state vector at time step $k$, $\Psi(\cdot)$ is the state transition map, and $\boldsymbol{\xi}_k \in \mathbb{R}^D$ denotes the stochastic force. The observations $\boldsymbol{y}_k \in \mathbb{R}^M$ are related to the state through the measurement map $\mathcal{A}(\cdot)$, with observation noise $\boldsymbol{\eta}_k \sim \mathcal{N}(\boldsymbol{0}, \gamma^2 \boldsymbol{I}_M)$. In many practical settings we have an initial window of known states $\boldsymbol{x}_{0:L-1}$, after which observations $\boldsymbol{y}_{L:K}$ will only be noisy and partial. The filtering problem in DA seeks to infer the posterior of the state trajectory $\boldsymbol{x}_{L:K}$ given observations $\boldsymbol{y}_{L:K}$ and the initial states $\boldsymbol{x}_{0:L-1}$, i.e., $p(\boldsymbol{x}_{L:K} \mid \boldsymbol{y}_{L:K}, \boldsymbol{x}_{0:L-1})$, as shown in Figure 1. Moreover, when the observations are absent, the task reduces to *probabilistic forecasting* [21], where we predict $p(\boldsymbol{x}_{L:K} \mid \boldsymbol{x}_{0:L-1})$.

Existing DA methods are split into two categories. Model-driven methods—Kalman variants for quasi-linear/Gaussian systems [22] and particle filters for fully nonlinear cases [23]—need an accurate physical model and become costly or unstable in high dimensions [24]. Data-driven surrogates learn unknown or partially known dynamics directly from observations and embed them within Bayesian inversion frameworks. However, the intrinsic stochasticity of complex systems exposes limitations: neural operators such as FNO [25] and Transolver [26] produce only deterministic forecasts and thus cannot quantify uncertainty, while diffusion models [21, 27, 28] struggle to learn physically faithful distributions due to the long Markov chain required to map Gaussian noise into realistic data.

**Our contributions.** We address the above limitations by building on recent flow-based *stochastic interpolant* methods [29–31], which learn a short-step conditional transition $p(\boldsymbol{x}_{k+1} \mid \boldsymbol{x}_k)$ instead of a global noise-to-data map. Our main contributions are:

- **FlowDAS**: We introduce a generative data-assimilation framework that treats a learned stochastic interpolant as the forward surrogate model and assimilates new observations on the fly at inference time, without retraining or auxiliary filtering steps.

- **Efficient, interpretable learning**: By modeling the transition between adjacent states, FlowDAS trains faster and more stably than long-horizon diffusion bridges, rolls out autoregressively, and provides clear physical insight into one-step dynamics.

- **Extensive validation**: Across the Lorenz-63 system, incompressible Navier–Stokes super-resolution and sparse-observation tasks, and large-scale SEVIR weather forecasting—where dynamics range from known to partially or fully unknown—FlowDAS consistently outperforms strong baselines in both accuracy and physical plausibility.

## 2 Preliminaries

### 2.1 Related Work

Data assimilation alternates between a *forecasting* step—propagating the current state forward under a dynamical model—and an *inverse* step that corrects this forecast with new observations.

**Model-driven methods**  Traditional model-driven approaches include the Kalman filter and its many variants [32]. The Kalman filter provides an optimal minimum-variance estimate when the dynamics are linear and all errors are Gaussian, but its performance degrades once these assumptions are not met. Also, there are some variants for nonlinear settings like the Extended Kalman Filter [33, 34], applying the nonlinear transition to predict states and its Jacobian to the covariance, struggling in discontinuity situations and causing computational expense to calculate the Jacobian. The Unscented Kalman Filters [35] bypass the computation of Jacobian by propagating a set of sample points with the nonlinear transformations and deriving the mean and covariance, making it sensitive to sample point parameters and non-Gaussian problems. For fully nonlinear, non-Gaussian problems, the bootstrap particle filter (BPF) [23] approximates the posterior with a set of weighted particles that are advanced by the dynamics, reweighted by the observation likelihood, and resampled to prevent weight collapse. Although flexible, the BPF requires an exponential number of particles as the state dimension grows, making it impractical for large physical systems [24]. Besides, both the Kalman family and particle filters rely on having accurate knowledge of the underlying dynamical models, a requirement that is often unmet in real-world applications.

**Neural operator-based methods**  Neural-operator solvers treat forecasting as a deterministic map from the current state to a single future state. The Fourier Neural Operator (FNO; [25]) learns this map in Fourier space, giving fast and accurate predictions, while Transolver ([26]) replaces Fourier layers with a physics-attention Transformer that can effectively capture complex physical correlations. Both models are deterministic and they are not designed for dynamics that are inherently stochastic. Recent work has sought to introduce stochasticity by fitting probabilistic surrogates—such as stochastic processes, graphical models, or stochastic Koopman operators [36, 37]—but these efforts typically focus on low-dimensional settings or assume simple Gaussian statistics, limiting their applicability to high-dimensional, strongly nonlinear systems. Researchers have also adapted neural operator-based models for DA tasks [38–40]. However, these adaptations are typically tailored to narrow use cases—such as arbitrary-resolution data assimilation—and have not demonstrated robust generalization across diverse real-world scenarios.

**Generative model-based methods**  Recent works have leveraged score-based methods, in particular diffusion models, for dynamical system modeling. For instance, the Conditional Diffusion Schrödinger Bridge (CDSB) framework extends diffusion models to conditional simulation and demonstrates strong performance on filtering in state-space models [41]. Score-based data assimilation (SDA) is a recent data-driven approach that employs score-based diffusion models to estimate state trajectories in dynamical systems [28]. It bypasses explicit physical modeling by learning the joint distribution of short state trajectory segments (e.g., $2k + 1$ time steps) via a score network $s_\theta(x_{i-k:i+k})$. By integrating the score network with the observation model in a diffusion posterior sampling (DPS) framework [42], SDA can generate entire state trajectories in zero-shot and non-autoregressive manners. However, SDA captures state transition implicitly from state concatenation, the learned dynamics of SDA may lack physical interpretability, leading to potential inaccuracies. And its posterior approximations may be less reliable in systems where the physical model is highly sensitive, as illustrated by the double well potential problem in Appendix D.1. Additionally, non-autoregressive diffusion models may struggle with long sequence forecasting in high-dimensional systems, though autoregressive diffusion approaches, like GenCast [43], have shown promising results in such settings. PDEDiff extends SDA by introducing a universal amortized conditional diffusion framework for PDE dynamics, where a conditional score network is trained to predict local state transitions given variable-length histories, enabling both forecasting and data assimilation across different PDE systems [21]. Another closely related line of work is the stochastic interpolant-based forecasting framework recently proposed in [31], which formulates the prediction task as conditional sampling of future system states given the current state, using a fictitious, non-physical stochastic process governed by a learned SDE. While this approach enables probabilistic forecasting, it does not incorporate observation-driven corrections during inference. In contrast, FlowDAS augments the stochastic interpolant dynamics with a measurement-consistent correction term at each step, enabling integration of observational data and thus unifying generative modeling with data assimilation.

## 2.2 Stochastic Interpolants

Stochastic interpolants [29–31] is a generative modeling framework that unifies flow-based and diffusion-based models, providing a smooth, controlled transition between arbitrary probability densities over a finite time horizon.

Consider a stochastic process $X_s$ defined over the interval $s \in [0, 1]$, which evolves from an initial state $X_0 \sim \pi(X_0)$ to the target state $X_1 \sim q(X_1)$. A stochastic interpolant can be described as:

$$\tilde{I}_s = \alpha_s X_0 + \beta_s X_1 + \sigma_s W_s, \tag{3}$$

where $(X_0, X_1) \sim p(X_0, X_1)$. $W_s$ is a Wiener process for $s \in [0, 1]$ introduced after $X_0$ and $X_1$ are sampled, ensuring that $W_s$ is independent of $X_0$ and $X_1$. The time-varying coefficients $\alpha_s, \beta_s, \sigma_s \in C^1([0, 1])$ satisfy boundary conditions $\alpha_0 = \beta_1 = 1$ and $\alpha_1 = \beta_0 = \sigma_1 = 0$, ensuring that $\tilde{I}_0 = X_0$ and $\tilde{I}_1 = X_1$, thereby creating a smooth interpolation from $X_0$ to $X_1$. Moreover, for all $(s, X_0) \in [0, 1] \times \mathbb{R}^D$, $\tilde{I}_s \mid X_0$ has the same distribution as $X_s$, which is the solution to the following SDE [31]:

$$dX_s = b_s(X_s, X_0) ds + \sigma_s dW_s, \tag{4}$$

where the drift term $b_s(X, X_0)$ is optimized by minimizing the cost function:

$$\mathcal{L}_b(\hat{b}_s) = \int_0^1 \mathbb{E}\left[\|\hat{b}_s(\tilde{I}_s, X_0) - R_s\|^2\right] ds. \tag{5}$$

The "velocity" of the interpolant path, $R_s$, is given by $R_s = \dot{\alpha}_s X_0 + \dot{\beta}_s X_1 + \dot{\sigma}_s W_s$. Furthermore, the drift term $b_s(X_s, X_0)$ is related to the score function $\nabla \log p(X_s \mid X_0)$:

$$b_s(X_s, X_0) = \frac{c_s(X_s, X_0)}{\beta_s} + \frac{\nabla \log p(X_s \mid X_0)}{\lambda_s \beta_s}, \tag{6}$$

where $\lambda_s$ and $c_s(X_s, X_0)$ are defined to control the score-based dynamics (Appendix A.2).

Building on this framework, stochastic interpolants can estimate the transition $p(x_{k+1} \mid x_k)$ by evolving a latent path $X_s$ smoothly from $s = 0$ (current state) to $s = 1$ (next state) in dynamical systems. It provides a probabilistic yet compact surrogate that is more aligned with the true step-to-step evolution of complex dynamical systems. While stochastic interpolants capture complex system dynamics, they do not yet enforce consistency with measurements.

# 3 Method

## 3.1 Stochastic Interpolants for Data Assimilation

Using stochastic interpolants as the engine of data assimilation is attractive but *not* a plug-and-play replacement for existing forward models: it raises *three fundamental challenges* that FlowDAS must overcome. Below we formulate each challenge and describe the corresponding FlowDAS solution.

**Challenge I: Observation-consistent state generation**    Stochastic interpolants approximate the state transition $p(x_{k+1} \mid x_k)$ by interpolating between the state variables $x_k$ and $x_{k+1}$ using the SDE defined in Equation (4) and Equation (6) with boundary conditions $X_0, X_1 = x_k, x_{k+1}$. Moreover, DA forward surrogate must respect the observation $y_{k+1}$ while drawing the future state $x_{k+1}$. For stochastic interpolants, this requires an *observation-conditioned* SDE. FlowDAS augments the original drift $b_s(X_s, X_0)$ via Bayes' rule (Appendix A.1):

$$b_s(X_s, y, X_0) = b_s(X_s, X_0) + \frac{\nabla \log p(y \mid X_s, X_0)}{\lambda_s \beta_s}. \tag{7}$$

**Challenge II: Estimating the observation-informed drift**    The term $\nabla \log p(y \mid X_s, X_0)$ captures the observation information, however, it is intractable because the observation model only directly links $y = y_{k+1}$ and $X_1 = x_{k+1}$. FlowDAS produces unbiased estimate of it by Monte-Carlo marginalization, i.e., integrating with respect to $X_1$, which can be approximated by $J$ Monte Carlo samples $X_1^{(j)} \sim p(X_1 \mid X_s, X_0)$:

$$\nabla \log p(y \mid X_s, X_0) = \frac{\nabla \mathbb{E}_{X_1 \sim p(X_1 \mid X_s, X_0)}[p(y \mid X_1)]}{\mathbb{E}_{X_1 \sim p(X_1 \mid X_s, X_0)}[p(y \mid X_1)]} \approx \sum_{j=1}^{J} w_j \nabla \log p(y \mid X_1^{(j)}), \tag{8}$$

where we apply a softmax function to the $J$ scalars $\{\log p(y \mid X_1^{(j)})\}_{j=1}^{J}$ to compute the sample weights $w_j = p(y \mid X_1^{(j)}) / \sum_{j=1}^{J} p(y \mid X_1^{(j)})$. We leave detailed derivation in Appendix A.3.

**Challenge III: Efficient Monte Carlo sampling** Accurate sampling from $p(\boldsymbol{X}_1 \mid \boldsymbol{X}_s, \boldsymbol{X}_0)$ requires solving the SDE in Equation (4), which can be computationally intensive. FlowDAS accelerates this step with low-order stochastic integrators:

First-order Milstein method [44, p. 317]:

$$\hat{\boldsymbol{X}}_1 = \boldsymbol{X}_s + \boldsymbol{b}_s(\boldsymbol{X}_s, \boldsymbol{X}_0)(1 - s) + \int_s^1 \sigma_\tau \, \mathrm{d}\boldsymbol{W}_\tau \tag{9}$$

and second-order stochastic Runge-Kutta method [44, p. 324]:

$$\hat{\boldsymbol{X}}_1' = \boldsymbol{X}_s + \frac{\boldsymbol{b}_s(\boldsymbol{X}_s, \boldsymbol{X}_0) + \boldsymbol{b}_1(\hat{\boldsymbol{X}}_1, \boldsymbol{X}_0)}{2}(1 - s) + \int_s^1 \sigma_\tau \, \mathrm{d}\boldsymbol{W}_\tau. \tag{10}$$

Both approximations introduce slight numerical bias (i.e., $\mathcal{O}((1-s)^2)$ and $\mathcal{O}((1-s)^3)$, respectively) but significantly accelerated sampling speed. Appendix C.2 further compares these approximations.

**Advantages of FlowDAS.** Compared with other data-driven surrogates, FlowDAS offers two key advantages.

- **Observation-consistent probabilistic forecasts.** Neural operators such as FNO and Transolver learn a deterministic map $\boldsymbol{x}_k \mapsto \boldsymbol{x}_{k+1}$; they yield a single forecast and require a separate optimisation step to reconcile that forecast with the observation $\boldsymbol{y}_{k+1}$. FlowDAS learns the full conditional distribution $p(\boldsymbol{x}_{k+1} \mid \boldsymbol{x}_k, \boldsymbol{y}_{k+1})$. It therefore generates an ensemble of forecasts that are already consistent with the incoming measurement, eliminating the need for any post-hoc update.

- **Local and physics-aligned transport.** FlowDAS directly learns a *local* bridge between adjacent states, so the diffusion process spans a short distance in state space. Training therefore remains numerically stable and provides clear physical insight into one-step dynamics. By contrast, score-based diffusion models construct a global path from Gaussian noise to data; the network must master a far longer transformation and learn the state transition dynamics through channel correlations, which often obscures physical interpretability and can degrade accuracy.

### 3.2 Implementation Details

**Training** We train a user-defined neural network as the drift model $\boldsymbol{b}_s(\boldsymbol{X}_s, \boldsymbol{X}_0)$, which outputs the velocity given the current interpolant $\boldsymbol{X}_s$ and conditioned on the initial state $\boldsymbol{X}_0$. The network architecture of the drift model may vary across tasks, for example, a Multi-Layer Perceptron (MLP) is adopted for the low-dimensional Lorenz-63 system in Section 4.1, a U-Net for high-dimensional fluid system governed by incompressible Navier-Stokes equations in Section 4.2, and a FNO for the weather forecasting task in Section 4.3. The model is optimized using collected trajectories $\boldsymbol{x}_{0:K}$, minimizing an empirical loss between predicted and target velocities over sampled interpolation times $s \in [0, 1]$. The discrepancy metric (e.g., $\ell_1$ or squared $\ell_2$ norm, etc.) is also task-dependent.

**Inference** Given a trained drift model $\hat{\boldsymbol{b}}_s(\boldsymbol{X}_s, \boldsymbol{X}_0)$, FlowDAS performs inference in an autoregressive manner to estimate the state trajectory $\hat{\boldsymbol{x}}_{1:K}$. At each time step $k$, we begin from the known state $\hat{\boldsymbol{x}}_k$ and predict the next state $\hat{\boldsymbol{x}}_{k+1}$ using a discretized stochastic interpolant over a time grid $s_0 = 0 < s_1 < \cdots < s_N = 1$. We use $N = 500$ in our experiments. We set the interpolant state $\boldsymbol{X}_{s_0} \leftarrow \hat{\boldsymbol{x}}_k$ and retrieve the next observation $\boldsymbol{y}_{k+1}$. For each interpolation step $s_n$, we simulate a forward transition using the learned drift and noise. To enforce observation consistency, we generate $J$ posterior endpoint samples $\hat{\boldsymbol{X}}_1^{(j)}$, and compute their likelihoods under the measurement model, i.e., $\|\boldsymbol{y} - \mathcal{A}(\hat{\boldsymbol{X}}_1^{(j)})\|_2^2$. A softmax over these values gives importance weights $\{w_{1:J}\}$. We then apply a correction to the interpolant using a weighted gradient $-\zeta_n \nabla_{\boldsymbol{X}_{s_n}} \sum_{j=1}^J w_j \|\boldsymbol{y} - \mathcal{A}(\hat{\boldsymbol{X}}_1^{(j)})\|_2^2$, where $\zeta_n$ denotes the step size. After reaching $s = 1$, we set $\hat{\boldsymbol{x}}_{k+1} \leftarrow \boldsymbol{X}_{s_N}$ and proceed to the next time step. Repeating this procedure over all $K$ yields the full trajectory.

**Conditioning** While the original stochastic interpolants framework which only models the $p(\boldsymbol{x}_{k+1} \mid \boldsymbol{x}_k)$, we take in a sequence of previous states to achieve the transition from $p(\boldsymbol{x}_k \mid \boldsymbol{x}_{k-1}...\boldsymbol{x}_{k-l})$ to $p(\boldsymbol{x}_{k+1} \mid \boldsymbol{x}_k, \boldsymbol{x}_{k-1}...\boldsymbol{x}_{k-l})$ inspired by [45–48], thus achieving probabilistic prediction conditioned on several previous states. This straightforward extension allows FlowDAS to handle non-Markovian dynamics and empirically leads to markedly improved performance on weather-forecasting task (Section 4.3).

# 4 Experiments and Results

This section evaluates our proposed framework, FlowDAS, on a range of low- and high-dimensional stochastic dynamical systems, including low-dimensional problems with high-order observation models, such as the double-well potential (Appendix D.1.5) and the chaotic Lorenz 1963 system, as well as high-dimensional tasks involving the incompressible Navier-Stokes equations and a realistic problem: Particle Image Velocimetry (PIV). Additionally, we demonstrate its applicability on a real-world large-scale weather forecasting task, where governing dynamics are not available. These results underscore the versatility and robustness of FlowDAS.

## 4.1 Lorenz 1963

In this experiment, we evaluate the performance of FlowDAS using the Lorenz 1963 system, a simplied mathematical model for atmospheric convection that is widely studied in the DA community [28, 49, 50]. The state vector of the Lorenz system, $\boldsymbol{x} = (a, b, c) \in \mathbb{R}^3$, evolves according to the following nonlinear stochastic ordinary differential equations (ODEs):

$$
\begin{aligned}
\mathrm{d}a/\mathrm{d}t &= \mu(b - a) + \xi_1, \\
\mathrm{d}b/\mathrm{d}t &= a(\rho - c) - b + \xi_2, \\
\mathrm{d}c/\mathrm{d}t &= ab - \tau c + \xi_3,
\end{aligned}
\tag{11}
$$

where $\mu = 10$, $\rho = 28$, and $\tau = 8/3$ define the ODE parameters, and $\boldsymbol{\xi} = (\xi_1, \xi_2, \xi_3) \in \mathbb{R}^3$ is the Brownian process noise, with each component having a standard deviation $\sigma = 0.025$. This chaotic system poses a significant challenge for numerical methods, so we use the fourth-order Runge-Kutta (RK4) method [51] to simulate its state transition (see Appendix D.2 for details).

We observe only the arctangent-transformed value of the first state component $a$, so the observation model of the system is defined as

$$
y = \mathcal{A}(\boldsymbol{x}) + \boldsymbol{\eta} = \arctan(a) + \eta,
\tag{12}
$$

wehre $\eta$ is the observation noise with a standard deviation $\gamma = 0.25$.

**Dataset and experiments**   We generate 1,024 independent trajectories, each containing 1,024 states, and split the data into training (80%), validation (10%), and evaluation (10%) sets. Initial states are sampled from the statistically stationary regime of the Lorenz system, with additional data generation details provided in Appendix D.2. During inference, we independently estimate 64 trajectories over 15 time steps using FlowDAS and the baseline methods. A total of $L = 1$ previous state is conditioned for each autoregressive generation. For this low-dimensional problem, we use a fully connected neural network to approximate the drift term in stochastic interpolants; the network architecture is also described in the same appendix.

**Baselines and metrics**   We compare our method against two baselines: the SDA solver with a fixed window size of 2 and the classic BPF [23]. Appendix D.2 details the score network architecture for SDA and particle density settings for BPF.

We evaluate the performance of FlowDAS and baselines using four metrics: the expectation of log-prior $\mathbb{E}_{q(\boldsymbol{x}_{1:K}|\boldsymbol{y}_{1:K})}[\log p(\boldsymbol{x}_{2:K} \mid \boldsymbol{x}_1)]$; the expectation of log data likelihood $\mathbb{E}_{q(\boldsymbol{x}_{1:K}|\boldsymbol{y}_{1:K})}[\log p(\boldsymbol{y}_{1:K} \mid \boldsymbol{x}_{1:K})]$; the Wasserstein distance [52] $W_1(\cdot, \cdot)$ between the true trajectory $\boldsymbol{x}_{1:K}$ and the estimated trajectory $\hat{\boldsymbol{x}}_{1:K}$; and the RMSE between the true and estimated states.

|  |  | FLOWDAS | SDA | BPF |
|---|---|---|---|---|
| $\log p(\hat{\boldsymbol{x}}_{2:K} \mid \hat{\boldsymbol{x}}_1)$ | ↑ | 17.29 | -332.7 | **17.88** |
| $\log p(\boldsymbol{y} \mid \hat{\boldsymbol{x}}_{1:K})$ | ↑ | **-0.228** | -6.112 | -1.572 |
| $W_1(\boldsymbol{x}_{1:K}, \hat{\boldsymbol{x}}_{1:K})$ | ↓ | **0.106** | 0.528 | 0.812 |
| $\mathrm{RMSE}(\boldsymbol{x}_{1:K}, \hat{\boldsymbol{x}}_{1:K})$ | ↓ | **0.202** | 1.114 | 0.270 |

Table 1: **Data assimilation of Lorenz 1963 system.** This table summarizes the performance of FlowDAS, SDA, and BPF on the Lorenz 1963 experiment over 15 time steps. FlowDAS outperforms SDA across all evaluation metrics and is competitive with BPF, despite BPF utilizing the true transition equations, which are unknown to FlowDAS and SDA. The best results for each metric are highlighted in **bold**.

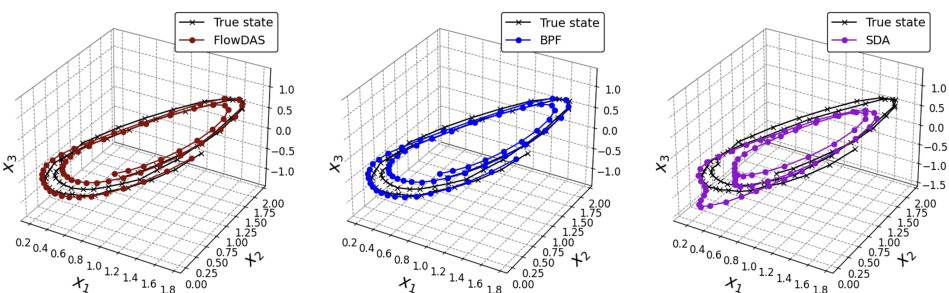

Figure 2: **Data assimilation of Lorenz 1963 system.** FlowDAS achieved results comparable to the state-of-the-art model-based BPF method, significantly outperforming the data-driven SDA method in recovering the underlying dynamics of this chaotic system. This highlights the efficiency and robustness of FlowDAS in capturing complex, nonlinear dynamics while maintaining accuracy and stability. The variables $x_1$, $x_2$ and $x_3$ correspond to $a$, $b$ and $c$ in the ODEs of the Lorenz system Equation (11), respectively.

**Results**    As shown in Table 1 and Figure 2, FlowDAS outperforms SDA across all metrics. FlowDAS is only slightly less effective than BPF in the expected log-prior, as BPF directly incorporates the true system dynamics into its state estimation. Appendix D.2 provides additional comparisons and results.

The success of FlowDAS is primarily due to its accurate mapping from current to future states $(\boldsymbol{x}_k \rightarrow \boldsymbol{x}_{k+1})$. Despite lacking explicit transition equations, FlowDAS effectively captures the system dynamics through stochastic interpolants, enabling a closer approximation of state trajectories compared to SDA, which models joint distributions across sequential states using diffusion models. Additionally, stochastic interpolants allow FlowDAS to produce accurate state estimates while managing inherent variability, avoiding over-concentration on high-probability regions, and effectively dealing with rare events. This advantage is further illustrated in the double well potential experiment (Appendix D.1.5) where FlowDAS outperforms BPF, because BPF tends to be trapped by high-probability point estimates.

### 4.2   Incompressible Navier-Stokes Flow

This section considers a high-dimensional dynamical system: incompressible fluid flow governed by the 2D Navier-Stokes (NS) equations with random forcing on the torus $\mathbb{T}^2 = [0, 2\pi]^2$. The state transition, $\Psi$, is described using the stream function formulation,

$$\mathrm{d}\boldsymbol{\omega} + \boldsymbol{v} \cdot \nabla\boldsymbol{\omega} \ \mathrm{d}t = \nu\Delta\boldsymbol{\omega} \ \mathrm{d}t - \alpha\boldsymbol{\omega} \ \mathrm{d}t + \varepsilon \, \mathrm{d}\boldsymbol{\xi}, \tag{13}$$

where $\boldsymbol{\omega}$ represents the vorticity field, the state variable in this fluid dynamics system ($\boldsymbol{x} = \boldsymbol{\omega}$). The velocity $\boldsymbol{v} = \nabla^\perp\psi = (-\partial_y\psi, \partial_x\psi)$ is expressed in terms of the stream function $\psi(x, y)$, which satisfies $-\Delta\psi = \boldsymbol{\omega}$. The term $d\boldsymbol{\xi}$ represents white-in-time random forcing acting on a few Fourier modes, with parameters $\nu, \alpha, \varepsilon > 0$ specified in Appendix D.3.1.

The observation operator $\mathcal{A}$ linearly downsamples or selects partial pixels from the simulated vorticity fields ($\boldsymbol{\omega}$),

$$\boldsymbol{y} = \mathcal{A}(\boldsymbol{\omega}) + \boldsymbol{\eta}, \tag{14}$$

where the observation noise $\boldsymbol{\eta}$ has a standard deviation of $\gamma = 0.05$.

**Dataset and experiments**    In this experiment, system dynamics are simulated by solving Equation (13) using a pseudo-spectral method [53] with a resolution of $256^2$ and a timestep $\Delta t = 10^{-4}$. We simulate 200 flow conditions over $t \in [0, 100]$, saving snapshots of fluid vorticity field ($\boldsymbol{\omega} = \nabla \times v$) at the second half of each trajectory ($t \in [50, 100]$) at intervals of $\Delta t = 0.5$ with a reduced resolution of $128^2$. The data are divided into training (80%), validation (10%), and evaluation (10%) sets.

We conduct experiments across different observation resolutions ($32^2$, $16^2$) and observation sparsity levels (5%, 1.5625%). For the super-resolution task, the goal is to reconstruct high-resolution vorticity data ($128^2$) from low-resolution observations. In the inpainting task, only 5% or 1.5625% of pixel values are retained, with the rest set to zero, and we attempt to recover the complete vorticity field.

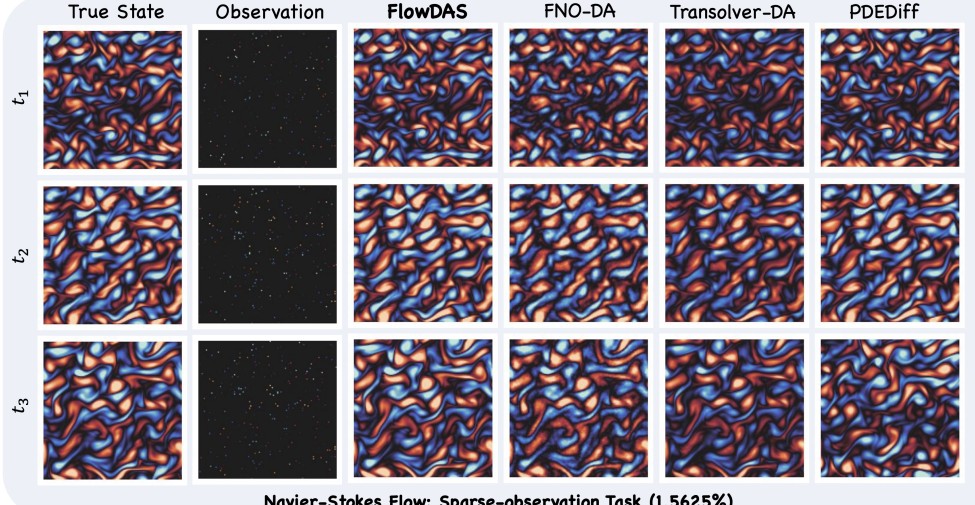

Figure 3: **Data assimilation of incompressible Navier-Stokes flow.** The positive values (red) of the state, i.e., vorticity field, indicate clockwise rotation and negative values (blue) indicate counter-clockwise rotation. FlowDAS achieved results with more accurate details and higher accuracy than all baselines, showing the efficiency of FlowDAS in tackling DA tasks with highly non-linear complex systems. Additionally, FlowDAS is also better at recovering high-frequency information, evidenced by the spectral analysis in Figure S.8.

|  | $32^2 \rightarrow 128^2$ | $16^2 \rightarrow 128^2$ | 5% | 1% |
|---|---|---|---|---|
| FLOWDAS | **0.077** $\pm$ 0.005 | **0.174** $\pm$ 0.006 | **0.156** $\pm$ 0.005 | **0.373** $\pm$ 0.012 |
| PDEDIFF | 0.112 $\pm$ 0.017 | 0.242 $\pm$ 0.035 | 0.221 $\pm$ 0.024 | 0.575 $\pm$ 0.068 |
| TRANSOLVER-DA | 0.597 $\pm$ 0.001 | 0.657 $\pm$ 0.001 | 0.660 $\pm$ 0.001 | 0.669 $\pm$ 0.001 |
| FNO-DA | 0.653 $\pm$ 0.001 | 0.732 $\pm$ 0.001 | 0.695 $\pm$ 0.001 | 0.753 $\pm$ 0.001 |

Table 2: **RMSE ($\pm$ std) of FlowDAS and baselines on incompressible Navier–Stokes super-resolution and sparse-observation tasks.** Each value is computed from 50 independently sampled states, with 30 repeated runs per state. All results correspond to single-step assimilation. Figure S.12 presents the RMSE and standard deviations evaluated along a FlowDAS-generated trajectory.

The model is evaluated on four unseen datasets ($2\times$ super-resolution, $2\times$ inpainting), with 64 samples for each configuration. A total of $L = 10$ previous states are conditioned for each autoregressive generation.

**Baselines and metrics** We benchmark FlowDAS against three data-driven baselines. First, we implement PDEDiff solver with a fixed eleven-step window (10 previous states as conditions). Second, we adapt the deterministic neural operators FNO and Transolver to the DA setting, namely 'FNO-DA' and 'Transolver-DA'. Each network is trained to minimise RMSE between adjacent states; at inference time the raw prediction $\boldsymbol{x}'_{k+1} = f_\theta(\boldsymbol{x}_k)$ is reconciled with the observation $\boldsymbol{y}_{k+1}$ by solving $\hat{\boldsymbol{x}}_{k+1} = \arg\min_{\boldsymbol{x}} \alpha_1 \|\boldsymbol{x} - \boldsymbol{x}'_{k+1}\|_2^2 + \alpha_2 \|\boldsymbol{y}_{k+1} - \mathcal{A}(\boldsymbol{x})\|_2^2$, an optimization step analogous to a Kalman update, where $\alpha_1$ and $\alpha_2$ are weighting coefficients. The $\alpha_1$ and $\alpha_2$ are chosen by grid searching, and we report the best results. BPF is not included in our testing, as its particle requirements grow exponentially with system dimensions, making it impractical for high-dimensional fluid dynamics systems. Additional details on the model architecture and training for baselines and FlowDAS are provided in Appendix D.3.3.

We evaluate performance using the RMSE between the predicted and ground-truth vorticity fields. Additionally, we assess the reconstruction of the kinetic energy spectrum to determine whether the physical characteristics of the fluid are accurately preserved. Appendix D.3.5 and Figure S.8 provide the definitions and results for the kinetic energy spectrum metric.

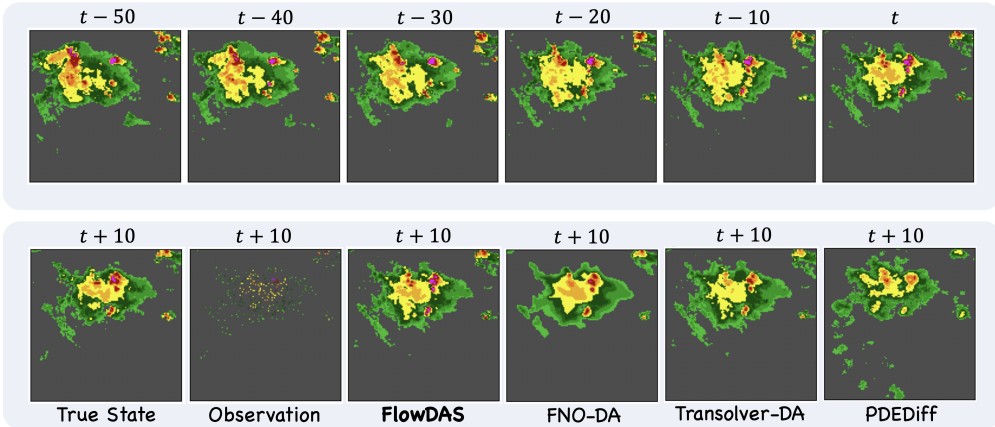

Figure 4: **Data assimilation of weather forecasting on SEVIR Vertical Integrated Liquid dataset under sparse observations.** All DA models take previous six states (displayed in the first row; $t-50$ min to $t$ min) as conditions and estimate the future state at $t + 10$ min. FlowDAS preserves storm cores and spatial texture better than score-based diffusion models and neural operator-based methods.

**Results** Figure 3 compares FlowDAS with baselines under the 1.5625% sparse-observation setting. See Appendix D.3 for additional results. Quantitative metrics, including RMSE and kinetic energy spectrum comparisons, are provided in Table 2 and Figure S.8. Our method consistently outperforms all baselines in terms of reconstruction accuracy, capturing high-frequency information with greater precision. This advantage is further validated by the kinetic energy spectrum in Figure S.8. The RMSE scores in Table 2 further highlight the effectiveness of FlowDAS in accurately estimating the underlying fluid dynamics from observational data. Appendix D.3.7 provides additional results on more challenging observation cases. Appendix D.3.8 presents the performance of FlowDAS when evaluated along an estimated trajectory.

**Particle Image Velocimetry** To demonstrate that our framework generalizes beyond the synthetic incompressible NS setting, we also apply FlowDAS to Particle Image Velocimetry, a widely used optical technique that infers sparse planar velocity vectors by tracking tracer particles in sequential images. In this scenario the DA problem is to reconstruct the full vorticity (or velocity) field from noisy, sparsely sampled velocities. We generate synthetic PIV frames from the same NS simulations, extract particle displacements with a standard PIV pipeline, and feed the resulting observations into the pretrained FlowDAS model. A comparison with baseline methods confirms that FlowDAS maintains its performance advantage in this practical setting. The full experimental protocol, quantitative metrics, and qualitative results are reported in Appendix D.4.

### 4.3 Weather Forecasting

This experiment evaluates FlowDAS on a large-scale, real-world weather–forecasting problem using the Storm EVent Imagery and Radar (SEVIR) dataset [54]. SEVIR provides multi-modal observations of severe convective storms across the continental United States. We focus on the *Vertically Integrated Liquid* (VIL) product, a 2-D proxy for precipitation intensity. Each sample is a $128 \times 128$ grid covering $384$ km $\times 384$ km at 2 km resolution and recorded every 10 min for four hours. Similar to [48], six consecutive VIL frames ($t-50$ min to $t$ min) constitute the input, and the task is to predict the next frame at $t + 10$ min. In our DA settings, the state variable is the VIL field $x$, and the observation is aquired by randomly sampling 10% of grid cells to emulate sparse radar coverage. A total of $L = 6$ previous states are conditioned for next-frame generation. We split the dataset into 80%/10%/10% for training, validation, and testing.

**Baselines and metrics** We compare FlowDAS with the PDEDiff solver (fixed window size of 7 with 6 previous states as conditions). Neural-operator baselines, e.g., FNO and Transolver, are similarly obtained as Navier-Stokes experiments. Both models takes a concatenation (along channel dimension) of states as inputs and output the state at target time step. Performance is measured by RMSE and the Critical Success Index (CSI) at thresholds of $\tau_{20}$ and $\tau_{40}$ dBZ, which are standard verification metrics for precipitation nowcasting.

| METHOD | RMSE ↓ | CSI($\tau_{20}$) (0.3) ↑ | CSI($\tau_{40}$) (0.5) ↑ |
|---|---|---|---|
| FLOWDAS | **0.053±0.004** | **0.746±0.022** | **0.614±0.044** |
| PDEDIFF | 0.071±0.007 | 0.549±0.033 | 0.387±0.065 |
| TRANSOLVER-DA | 0.062±0.001 | 0.663±0.001 | 0.499±0.002 |
| FNO-DA | 0.064±0.001 | 0.641±0.001 | 0.493±0.002 |

Table 3: **Comparison of FlowDAS and other baselines on RMSE and Critical Success Index (CSI) of the weather forecasting task.** All metrics are averaged over 30 repeated runs with standard deviations.

**Inplementation Details** Different from [31], we adopted FNO [25] rather than U-Net as the backbone of the stochastic interpolants framework. We provide the comparison between the FNO-based and the U-Net-based FlowDAS in the Appendix D.5, and found that the FNO-based FlowDAS demonstrates better performance.

**Results** FlowDAS achieves lower RMSE and higher CSI at both thresholds, indicating more accurate intensity estimates and better hit rates for heavy precipitation as shown in Appendix D.5. Qualitative comparisons in Figure 4 (and Figures S.15 to S.17) show that FlowDAS reconstructs coherent *precipitation structures* and *peak intensities* that other baselines either smooth out or mis-localize. These results demonstrate that the stochastic-interpolant surrogate scales to high-resolution, real meteorological data, even when underlying governed PDEs are unknown, and retains its advantage over diffusion models and neural operators baselines.

## 5 Limitations and Future Work

FlowDAS has so far been validated on controlled dynamical systems (e.g., Lorenz–63, Navier–Stokes), and its generalizability to more complex, real-world environments such as numerical weather prediction remains to be evaluated. In particular, we have not yet examined its performance under Sim2Real conditions or applied it to operational-scale, high-dimensional systems such as those targeted by GEN_BE [55]. Although FlowDAS supports probabilistic inference through sampling, its current sampling speed is a limitation. A promising direction is to employ post-hoc distillation, training a lightweight surrogate model to approximate the FlowDAS sampler. Another avenue is to explore hybrid formulations that combine stochastic interpolants with variational inference techniques to further improve computational efficiency.

## 6 Conclusion and Future Work

This work introduced **FlowDAS**, a stochastic interpolant-based data assimilation framework designed to address the challenges of high-dimensional, nonlinear dynamical systems. By leveraging stochastic interpolants, FlowDAS effectively integrates complex transition dynamics with observational data, enabling accurate state estimation without relying on explicit physical simulations. Through experiments on both low- and high-dimensional systems—including the Lorenz 1963 system, incompressible Navier-Stokes flow, and weather forecasting—FlowDAS demonstrated strong performance in recovering accurate state variables from sparse, noisy observations. These results highlight Flow-DAS as a robust alternative to traditional model-driven methods (e.g., particle filters) and data-driven approaches (e.g., score-based diffusion models and neural operators) for data assimilation, offering improved accuracy, efficiency, and adaptability.

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

# A  Mathematical Derivation

## A.1  Conditional Drift

Stochastic interpolants approximate the state transition $p(\boldsymbol{x}_{k+1} \mid \boldsymbol{x}_k)$ by interpolating between the state variables $\boldsymbol{x}_k$ and $\boldsymbol{x}_{k+1}$ using the SDE defined in Equation (4) and Equation (6) with boundary conditions $\boldsymbol{X}_0, \boldsymbol{X}_1 = \boldsymbol{x}_k, \boldsymbol{x}_{k+1}$. The drift term $\boldsymbol{b}_s(\boldsymbol{X}_s, \boldsymbol{X}_0)$ is related to the score function $\nabla \log p(\boldsymbol{X}_s \mid \boldsymbol{X}_0)$ of state transition distribution (Appendix A.2):

$$\boldsymbol{b}_s(\boldsymbol{X}_s, \boldsymbol{X}_0) = \frac{c_s(\boldsymbol{X}_s, \boldsymbol{X}_0)}{\beta_s} + \frac{\nabla \log p(\boldsymbol{X}_s \mid \boldsymbol{X}_0)}{\lambda_s \beta_s}. \tag{S.15}$$

To generate observation-consistent states, we modify the SDE to a process conditioned on observational data, where the drift term $\boldsymbol{b}_s(\boldsymbol{X}_s, \boldsymbol{y}, \boldsymbol{X}_0)$ incorporates observation information via Bayes' rule:

$$\begin{aligned}
\boldsymbol{b}_s(\boldsymbol{X}_s, \boldsymbol{y}, \boldsymbol{X}_0) &= \frac{c_s(\boldsymbol{X}_s, \boldsymbol{X}_0)}{\beta_s} + \frac{\nabla \log p(\boldsymbol{X}_s \mid \boldsymbol{y}, \boldsymbol{X}_0)}{\lambda_s \beta_s} \\
&= \frac{c_s(\boldsymbol{X}_s, \boldsymbol{X}_0)}{\beta_s} + \frac{\nabla \log p(\boldsymbol{y} \mid \boldsymbol{X}_s, \boldsymbol{X}_0) + \nabla \log p(\boldsymbol{X}_s \mid \boldsymbol{X}_0)}{\lambda_s \beta_s} \\
&= \boldsymbol{b}_s(\boldsymbol{X}_s, \boldsymbol{X}_0) + \frac{\nabla \log p(\boldsymbol{y} \mid \boldsymbol{X}_s, \boldsymbol{X}_0)}{\lambda_s \beta_s}.
\end{aligned} \tag{S.16}$$

## A.2  Relating Drift and Score

By extending Stein's formula [30], we have:

$$\nabla \log p(\boldsymbol{X}_s \mid \boldsymbol{X}_0) = -\frac{1}{\sqrt{s}\sigma_s} \mathbb{E}_{\boldsymbol{X}_0 \sim \pi(\boldsymbol{X}_0)}[\boldsymbol{z} \mid \boldsymbol{X}_s], \tag{S.17}$$

where $\boldsymbol{X}_s = \alpha_s \boldsymbol{X}_0 + \beta_s \boldsymbol{X}_1 + \sqrt{s}\sigma_s \boldsymbol{z}$. In addition,

$$\begin{aligned}
\boldsymbol{b}_s(\boldsymbol{X}_s, \boldsymbol{X}_0) &= \dot{\alpha}_s \boldsymbol{X}_0 + \dot{\beta}_s \mathbb{E}_{\boldsymbol{X}_0 \sim \pi(\boldsymbol{X}_0)}[\boldsymbol{X}_1 \mid \boldsymbol{X}_s] + \sqrt{s}\dot{\sigma}_s \mathbb{E}_{\boldsymbol{X}_0 \sim \pi(\boldsymbol{X}_0)}[\boldsymbol{z} \mid \boldsymbol{X}_s], \\
\boldsymbol{X}_s &= \alpha_s \boldsymbol{X}_0 + \beta_s \mathbb{E}_{\boldsymbol{X}_0 \sim \pi(\boldsymbol{X}_0)}[\boldsymbol{X}_1 \mid \boldsymbol{X}_s] + \sqrt{s}\sigma_s \mathbb{E}_{\boldsymbol{X}_0 \sim \pi(\boldsymbol{X}_0)}[\boldsymbol{z} \mid \boldsymbol{X}_s].
\end{aligned} \tag{S.18}$$

Solve $\mathbb{E}_{\boldsymbol{X}_0 \sim \pi(\boldsymbol{X}_0)}[\boldsymbol{z} \mid \boldsymbol{X}_s]$ from Equation (S.18):

$$\mathbb{E}_{\boldsymbol{X}_0 \sim \pi(\boldsymbol{X}_0)}[\boldsymbol{z} \mid \boldsymbol{X}_s] = \frac{\beta_s \boldsymbol{b}_s(\boldsymbol{X}_s, \boldsymbol{X}_0) - \dot{\beta}_s \boldsymbol{X}_s - (\beta_s \dot{\alpha}_s - \dot{\beta}_s \alpha_s) \boldsymbol{X}_0}{\sqrt{s}(\dot{\sigma}_s \beta_s - \sigma_s \dot{\beta}_s)}. \tag{S.19}$$

Define $\lambda_s = \frac{1}{\sqrt{s}(\dot{\sigma}_s \beta_s - \sigma_s \dot{\beta}_s)}$ and $\boldsymbol{c}_s(\boldsymbol{X}_s, \boldsymbol{X}_0) = \dot{\beta}_s \boldsymbol{X}_s + (\beta_s \dot{\alpha}_s - \dot{\beta}_s \alpha_s) \boldsymbol{X}_0$, and insert Equation (S.19) into Equation (S.17), we relate the score function $\nabla \log p(\boldsymbol{X}_s \mid \boldsymbol{X}_0)$ and the drift $\boldsymbol{b}_s(\boldsymbol{X}_s, \boldsymbol{X}_0)$ [31] by:

$$\nabla \log p(\boldsymbol{X}_s \mid \boldsymbol{X}_0) = \lambda_s [\beta_s \boldsymbol{b}_s(\boldsymbol{X}_s, \boldsymbol{X}_0) - \boldsymbol{c}_s(\boldsymbol{X}_s, \boldsymbol{X}_0)]. \tag{S.20}$$

## A.3  Estimation of the Gradient Log Likelihood of Observation

The term $\nabla \log p(\boldsymbol{y} \mid \boldsymbol{X}_s, \boldsymbol{X}_0)$ in Equation (7) captures the observation information. Since the observation model only directly links $\boldsymbol{y} = \boldsymbol{y}_{k+1}$ and $\boldsymbol{X}_1 = \boldsymbol{x}_{k+1}$, we compute this term by integrating with respect to $\boldsymbol{X}_1$:

$$\begin{aligned}
\nabla \log p(\boldsymbol{y} \mid \boldsymbol{X}_s, \boldsymbol{X}_0) &= \frac{\nabla p(\boldsymbol{y} \mid \boldsymbol{X}_s, \boldsymbol{X}_0)}{p(\boldsymbol{y} \mid \boldsymbol{X}_s, \boldsymbol{X}_0)} = \frac{\nabla \int p(\boldsymbol{y} \mid \boldsymbol{X}_1) p(\boldsymbol{X}_1 \mid \boldsymbol{X}_s, \boldsymbol{X}_0) \, \mathrm{d}\boldsymbol{X}_1}{\int p(\boldsymbol{y} \mid \boldsymbol{X}_1) p(\boldsymbol{X}_1 \mid \boldsymbol{X}_s, \boldsymbol{X}_0) \, \mathrm{d}\boldsymbol{X}_1} \\
&= \frac{\nabla \mathbb{E}_{\boldsymbol{X}_1 \sim p(\boldsymbol{X}_1 \mid \boldsymbol{X}_s, \boldsymbol{X}_0)}[p(\boldsymbol{y} \mid \boldsymbol{X}_1)]}{\mathbb{E}_{\boldsymbol{X}_1 \sim p(\boldsymbol{X}_1 \mid \boldsymbol{X}_s, \boldsymbol{X}_0)}[p(\boldsymbol{y} \mid \boldsymbol{X}_1)]}.
\end{aligned} \tag{S.21}$$

In practice, we approximate the above expectations by $J$ Monte Carlo samples, $\boldsymbol{X}_1^{(j)} \sim p(\boldsymbol{X}_1 \mid \boldsymbol{X}_s, \boldsymbol{X}_0)$:

$$\mathbb{E}_{\boldsymbol{X}_1 \sim p(\boldsymbol{X}_1 \mid \boldsymbol{X}_s, \boldsymbol{X}_0)}[p(\boldsymbol{y} \mid \boldsymbol{X}_1)] \approx \frac{1}{J} \sum_{j=1}^{J} p(\boldsymbol{y} \mid \boldsymbol{X}_1^{(j)}). \tag{S.22}$$

---

**Algorithm 1** Training

---
1: **Input:** Dataset $\boldsymbol{x}_{0:K}$; minibatch size $K' \leq K$; coefficients $\alpha_s, \beta_s, \sigma_s$
2: **repeat**
3:  Compute $\tilde{\boldsymbol{I}}_s^k$ and $\boldsymbol{R}_s^k$ using (S.25) for $k \in \mathcal{B}_{K'}$
4:  Compute the empirical loss $\mathcal{L}_b^{\text{emp}}(\hat{\boldsymbol{b}})$ in Equation (S.24)
5:  Take the gradient step on $\mathcal{L}_b^{\text{emp}}(\hat{\boldsymbol{b}})$ to update $\hat{\boldsymbol{b}}_s$
6: **until** converged
7: **return** drifts $\hat{\boldsymbol{b}}_s$

---

Plug Equation (S.22) into Equation (S.21):

$$
\begin{aligned}
\nabla \log p(\boldsymbol{y} \mid \boldsymbol{X}_s, \boldsymbol{X}_0) &\approx \frac{\nabla \frac{1}{J} \sum_{j=1}^J p(\boldsymbol{y} \mid \boldsymbol{X}_1^{(j)})}{\frac{1}{J} \sum_{j=1}^J p(\boldsymbol{y} \mid \boldsymbol{X}_1^{(j)})} = \frac{\sum_{j=1}^J p(\boldsymbol{y} \mid \boldsymbol{X}_1^{(j)}) \nabla \log p(\boldsymbol{y} \mid \boldsymbol{X}_1^{(j)})}{\sum_{j=1}^J p(\boldsymbol{y} \mid \boldsymbol{X}_1^{(j)})} \\
&= \sum_{j=1}^J w_j \nabla \log p(\boldsymbol{y} \mid \boldsymbol{X}_1^{(j)}),
\end{aligned}
\tag{S.23}
$$

where we apply a softmax function to the $J$ scalars $\{\log p(\boldsymbol{y} \mid \boldsymbol{X}_1^{(j)})\}_{j=1}^J$ to compute the sample weights $w_j = p(\boldsymbol{y} \mid \boldsymbol{X}_1^{(j)}) / \sum_{j=1}^J p(\boldsymbol{y} \mid \boldsymbol{X}_1^{(j)})$.

## B Implementation Details

**Training** We used a dataset consisting of multiple simulated state trajectories $\boldsymbol{x}_{0:K}$ to train the drift function $\boldsymbol{b}_s(\boldsymbol{X}_s, \boldsymbol{X}_0)$ in stochastic interpolants (see Appendix B.1 for more details). We approximate the cost function in Equation (5) by the empirical loss:

$$
\mathcal{L}_b^{\text{emp}}(\hat{\boldsymbol{b}}) = \frac{1}{K'} \sum_{k \in \mathcal{B}_{K'}} \int_0^1 \ell\big(\hat{\boldsymbol{b}}_s(\tilde{\boldsymbol{I}}_s^k, \boldsymbol{x}_k), \boldsymbol{R}_s^k\big) \, \mathrm{d}s,
\tag{S.24}
$$

where $\ell(\cdot, \cdot)$ denotes a user-chosen discrepancy metric, depending on the experiment. $\mathcal{B}_{K'} \subset \{0 : K\}$ is a subset of indices of cardinality $K' \leq K$, with

$$
\begin{aligned}
\tilde{\boldsymbol{I}}_s^k &= \alpha_s \boldsymbol{x}_k + \beta_s \boldsymbol{x}_{k+1} + \sqrt{s} \sigma_s \boldsymbol{z}_k \\
\boldsymbol{R}_s^k &= \dot{\alpha}_s \boldsymbol{x}_k + \dot{\beta}_s \boldsymbol{x}_{k+1} + \sqrt{s} \dot{\sigma}_s \boldsymbol{z}_k,
\end{aligned}
\tag{S.25}
$$

where $\boldsymbol{z}_k \sim \mathcal{N}(0, \boldsymbol{I}_D)$ and satisfy $\boldsymbol{W}_s \overset{d}{=} \sqrt{s} \boldsymbol{z}$ with $\boldsymbol{z} \sim \mathcal{N}(0, \boldsymbol{I}_D)$ for all $s \in [0, 1]$. We approximate the integral over $s$ in Equation (S.24) via an empirical expectation sampling from $s \sim U([0, 1])$. Algorithm 1 provides a detailed description of the model training process.

**Inference** Our inference procedure generates trajectories conditioned on observations $\boldsymbol{y}_{L:K}$ using the learned drift model $\hat{\boldsymbol{b}}_s(\boldsymbol{X}_s, \boldsymbol{X}_0)$. We start by setting a specific state $\boldsymbol{x}_k$ as initial $\boldsymbol{X}_0$ and iterate over a predefined temporal grid $s_0 = 0 < s_1 < \cdots < s_N = 1$. Within each iteration, we first compute posterior estimates $\{\hat{\boldsymbol{X}}_1^{(j)}\}_{j=1}^J$ using Equations (9) and (10) for $J$ times. Then, we move one step further towards $s_N$ on $\boldsymbol{X}_{s_n}$ by solving Equation (7) which involves backpropagating the gradient $\nabla_{\boldsymbol{X}_{s_n}} \sum_{j=1}^J w_j \|\boldsymbol{y} - \mathcal{A}(\hat{\boldsymbol{X}}_1^{(j)})\|_2^2$ to enforce consistency with observations $\boldsymbol{y}_k$. We iterated this process autoregressively, by setting $\boldsymbol{X}_{s_0}^{k+1} = \boldsymbol{X}_{s_N}^k$. Empirically, we found that using a constant step size $\zeta_n$ across the inference process produced generally good results, although fine-tuning $\zeta_n$ at each step can slightly improve the performance [42, 56]. The chosen $J$ values crossed all experiments, reported in Table S.4, were sufficient large for stable performance, with subtle variation across the dimensionality of problems. An ablation study, detailed in Appendix C.1, further confirmed that larger $J$ offers diminishing returns. Overall, Algorithm 2 summarizes the inference process.

### B.1 Constructing Training Dataset

In the training stage for all experiments, we require pairs of consecutive states to train the velocity model $\boldsymbol{b}_s$. To generate these pairs, we proceed as follows:

**Algorithm 2** Inference

1: **Input:** Observation $\boldsymbol{y}_{L:K}$, the measurement map $\mathcal{A}$, initial state $\boldsymbol{x}_0$, model $\hat{\boldsymbol{b}}_s(\boldsymbol{X}, \boldsymbol{X}_0)$, noise coefficient $\sigma_s$, grid $s_0 = 0 < s_1 < \cdots < s_N = 1$, i.i.d. $\boldsymbol{z}_n \sim \mathcal{N}(0, \boldsymbol{I}_D)$ for $n = 0 : N - 1$, step size $\zeta_n$, Monte Carlo sampling times $J$
2: Set $\hat{\boldsymbol{x}}_{L-1} \leftarrow \boldsymbol{x}_{L-1}$
3: Set the $(\Delta s)_n = s_{n+1} - s_n$, $n = 0 : N - 1$
4: **for** $k = L - 1$ **to** $K - 1$ **do**
5:     $\boldsymbol{X}_{s_0}, \boldsymbol{y} \leftarrow \hat{\boldsymbol{x}}_k, \boldsymbol{y}_{k+1}$
6:     **for** $n = 0$ **to** $N - 1$ **do**
7:         $\boldsymbol{X}'_{s_{n+1}} = \boldsymbol{X}_{s_n} + \hat{\boldsymbol{b}}_s(\boldsymbol{X}_{s_n}, \boldsymbol{X}_{s_0})(\Delta s)_n + \sigma_{s_n}\sqrt{(\Delta s)_n}\boldsymbol{z}_n$
8:         $\{\hat{\boldsymbol{X}}_1^{(j)}\}_{j=1}^J \leftarrow$ Posterior estimation $(\hat{\boldsymbol{b}}_s, s_n, \boldsymbol{X}_0, \boldsymbol{X}_{s_n})$
9:         $\{w_j\}_{j=1}^J \leftarrow$ Softmax$\left(\{\|\boldsymbol{y} - \mathcal{A}(\hat{\boldsymbol{X}}_1^{(j)})\|_2^2\}_{j=1}^J\right)$
10:        $\boldsymbol{X}_{s_{n+1}} = \boldsymbol{X}'_{s_{n+1}} - \zeta_n \nabla_{\boldsymbol{X}_{s_n}} \sum_{j=1}^J w_j \|\boldsymbol{y} - \mathcal{A}(\hat{\boldsymbol{X}}_1^{(j)})\|_2^2$
11:     **end for**
12:     $\hat{\boldsymbol{x}}_{k+1} \leftarrow \boldsymbol{X}_{s_N}$
13: **end for**
14: **return** $\{\hat{\boldsymbol{x}}_k\}_{k=L}^K$

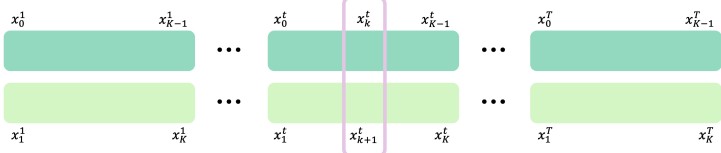

Figure S.5: **Structure of training data**. Consecutive states are paired across multiple simulated trajectories to construct the $\tilde{\boldsymbol{I}}_s$ and $\boldsymbol{R}_s$ defined in Equation (S.24) for training the velocity model $\boldsymbol{b}_s$.

1. **Trajectory simulation.** We simulate $T$ independent trajectories, denoted as $\boldsymbol{x}_{0:K}^{1:T}$, where each trajectory starts from a unique initial state $\boldsymbol{x}_0^t$. The state transitions within each trajectory follow the dynamics defined in Equation (1).

2. **Consecutive state pairs formation.** For each trajectory $t$, form two aligned sequences:

   (a) $\boldsymbol{x}_{0:K-1}^t$: The original sequence of states with each last state $x_K^t$ discarded.

   (b) $\boldsymbol{x}_{1:K}^t$: The sequence of states shifted by one time step with each initial state $x_0^t$ discarded.

   The two sequences form pairs of consecutive states $(\boldsymbol{x}_k^t, \boldsymbol{x}_{k+1}^t)$ for $k = 0, 1, \cdots, K - 1$.

3. **Concatenating trajectories.** Then, we concatenate $T$ sequences of $\boldsymbol{x}_{0:K-1}^t$ and $T$ sequences $\boldsymbol{x}_{1:K}^t$ end-to-end, respectively, into two long sequences: $\boldsymbol{x}_{0:K-1}^{1:T}$ and $\boldsymbol{x}_{1:K}^{1:T}$, where each state in the second sequence is the corresponding successor states in the first sequence.

4. **Sampling training data.** During training, batches of paired consecutive states are sampled. For each batch, we sample training data pairs as follows:

   (a) sample states from $\boldsymbol{x}_{0:K-1}^{1:T}$ as $\boldsymbol{X}_0$'s.

   (b) retrieve the counterpart states from $\boldsymbol{x}_{1:K}^{1:T}$ as $\boldsymbol{X}_1$'s.

   The batches of state pairs $(\boldsymbol{X}_0, \boldsymbol{X}_1)$ are used to construct $\tilde{\boldsymbol{I}}_s$ and $\boldsymbol{R}_s$ in Equation (S.24) for training the velocity model $\boldsymbol{b}_s$ as described in Algorithm 1.

In summary, we draw many state pairs $(\boldsymbol{X}_0, \boldsymbol{X}_1)$ and estimate the integral over $s$ in Equation (S.24) by approximated via an empirical expectation over draws of $s \sim U(0, 1)$, and for every $s$ and every state pairs $(\boldsymbol{X}_0, \boldsymbol{X}_1)$, we independently compute Equation (S.25) with $S$ samples of $\boldsymbol{z}_k$, so the training loss can be rewritten as:

$$\mathcal{L}_b^{\text{emp}}(\hat{\boldsymbol{b}}) = \frac{1}{K'} \sum_{k \in \mathcal{B}_{K'}} \frac{1}{S} \sum_{s=1}^S \|\hat{\boldsymbol{b}}_s(\tilde{\boldsymbol{I}}_s^k, \boldsymbol{x}_k) - \boldsymbol{R}_s^k\|^2, \tag{S.26}$$

| | | MONTE CARLO | |
| | | SAMPLING TIMES $J$ | SAMPLING STEP SIZE $\zeta$ |
|---|---|---|---|
| DOUBLE-WELL | | 17 | 1 |
| LORENZ 1963 | | 21 | 0.0002 |
| INCOMPRESSIBLE NAVIER STOKES | SR 4x | 25 | 1 |
| | SR 8x | 25 | 2 |
| | SO 5% | 25 | 1 |
| | SO 1.5625% | 25 | 1.75 |
| PARTICLE IMAGE VELOCIMETRY | | 25 | 1 |
| WEATHER FORECASTING | | 25 | 0.1 |

Table S.4: **Hyperparameters for FlowDAS in the inference stage** of all experiments presented in this study. SR stands for super-resolution task and SO represents sparse observation (inpainting) task.

where we randomly draws $s$ from $U(0, 1)$. Note that the training batch size equals $S \times K'$. Figure S.5 illustrates the structure of training data, showing how consecutive states are paired across trajectories.

### B.2 FlowDAS Inference Hyperparameters

We present key hyperparameters for FlowDAS in the inference stage, including the Monte Carlo sampling times $J$ and the sampling step size $\xi$ in Table S.4.

## C Ablation Study

This section examines different aspects of the proposed FlowDAS, including an alternative to the method and the hyperparameter settings. We also provide various evaluation results. To simplify notation, we omit the explicit sampling distribution $p(\boldsymbol{x})$ in the expectation operator, writing $\mathbb{E}_{\boldsymbol{x} \sim p(\boldsymbol{x})}[f(\boldsymbol{x})]$ simply as $\mathbb{E}_{\boldsymbol{x}}[f(\boldsymbol{x})]$. The sampling distribution for $\boldsymbol{x}$ will be specified at the end of the equation where necessary.

### C.1 Monte Carlo Sampling and An Alternative

In Appendix A.3, we estimate $\nabla \log p\left(\boldsymbol{y} \mid \boldsymbol{X}_s, \boldsymbol{X}_0\right)$ by

$$\nabla \log p\left(\boldsymbol{y} \mid \boldsymbol{X}_s, \boldsymbol{X}_0\right) = \frac{\nabla \mathbb{E}_{\boldsymbol{X}_1}\left[p\left(\boldsymbol{y}|\boldsymbol{X}_1\right)\right]}{\mathbb{E}_{\boldsymbol{X}_1}\left[p\left(\boldsymbol{y}|\boldsymbol{X}_1\right)\right]}, \tag{S.27}$$

where we approximate the expectation term by averaging $J$ Monte Carlo samples:

$$\mathbb{E}_{\boldsymbol{X}_1}\left[p\left(\boldsymbol{y}|\boldsymbol{X}_1\right)\right] \approx \frac{1}{J} \sum_{j=1}^{J} p\left(\boldsymbol{y}|\boldsymbol{X}_1^{(j)}\right), \boldsymbol{X}_1^{(j)} \sim p\left(\boldsymbol{X}_1|\boldsymbol{X}_s, \boldsymbol{X}_0\right). \tag{S.28}$$

This Monte Carlo estimate is an unbiased estimate and the error is proportional to $\frac{1}{\sqrt{J}}$. When the number of Monte Carlo samples, i.e., $J$, is sufficiently large, the estimation error converges to zero. Alternatively, one can also apply Jensen's inequality to estimate $\nabla \log p\left(\boldsymbol{y} \mid \boldsymbol{X}_s, \boldsymbol{X}_0\right)$, which provides a biased estimate:

$$\begin{aligned} \nabla \log p\left(\boldsymbol{y} \mid \boldsymbol{X}_s, \boldsymbol{X}_0\right) &= \nabla \log \int p\left(\boldsymbol{y}|\boldsymbol{X}_1\right) p\left(\boldsymbol{X}_1|\boldsymbol{X}_s, \boldsymbol{X}_0\right) \mathrm{d}\boldsymbol{X}_1 \\ &= \nabla \log \mathbb{E}_{\boldsymbol{X}_1}\left[p\left(\boldsymbol{y}|\boldsymbol{X}_1\right)\right] \geq \nabla \mathbb{E}_{\boldsymbol{X}_1}\left[\log p\left(\boldsymbol{y}|\boldsymbol{X}_1\right)\right]. \end{aligned} \tag{S.29}$$

The "$\geq$" arises from Jensen's inequality.

**Unbiased vs. biased estimation** Let $Z = p(\boldsymbol{y}|\boldsymbol{X}_1) = \frac{1}{\sqrt{2\pi}\gamma} e^{-\frac{(\boldsymbol{y} - \mathcal{A}(\boldsymbol{X}_1))^2}{2\gamma^2}}$, where $Z$ is a bounded random variable within the finite range $[0, \frac{1}{\sqrt{2\pi}\gamma}]$. As a result, the logarithm function is $\alpha$-Hölder continuous with $\alpha = 1$ and the gap introduced by Jensen's inequality, i.e., the Jensen gap, can be explicitly bounded [57] and given by

$$\left|\log \mathbb{E}_{\boldsymbol{X}_1}\left[p\left(\boldsymbol{y}|\boldsymbol{X}_1\right)\right] - \mathbb{E}_{\boldsymbol{X}_1}\left[\log p(\boldsymbol{y}|\boldsymbol{X}_1)\right]\right| \leq M\gamma_1^1 \tag{S.30}$$

|   |   | UNBIASED | BIASED |
|---|---|---|---|
| ↑ | $\log p(\hat{\boldsymbol{x}}_{2:K} \mid \hat{\boldsymbol{x}}_1)$ | **17.29** | -36.98 |
| ↑ | $\log p(\boldsymbol{y} \mid \hat{\boldsymbol{x}}_{1:K})$ | **-0.228** | -1.530 |
| ↓ | $W_1(\boldsymbol{x}_{1:K}, \hat{\boldsymbol{x}}_{1:K})$ | **0.106** | 0.111 |
| ↓ | $\mathrm{RMSE}(\boldsymbol{x}_{1:K}, \hat{\boldsymbol{x}}_{1:K})$ | **0.202** | 0.363 |

Table S.5: **Evaluation of unbiased vs. biased estimation**. Comparison of metrics between unbiased and biased estimation in the Lorenz experiments. The results demonstrate that the unbiased estimation outperforms the biased estimation.

where $M$ is a constant that satisfying $|\log Z - \log \mathbb{E}[Z]| \leq M |Z - \mathbb{E}[Z]|$ and $\gamma_1^1 = \mathbb{E}[|Z - \mathbb{E}[Z]|]$. Additionally, because $Z \in [0, \frac{1}{\sqrt{2\pi}\gamma}]$, we have $M < 1$ and $\gamma_1^1 \leq \max|Z - \mathbb{E}[Z]| \leq \frac{1}{\sqrt{2\pi}\gamma}$.

And for this upper-bound $M\gamma_1^1$, when $s \to 1$, $p(\boldsymbol{X}_1 \mid \boldsymbol{X}_s, \boldsymbol{X}_0)$ will become a delta distribution concentrated on $\hat{\boldsymbol{X}}_1$ and $Z = p(\boldsymbol{y}|\boldsymbol{X}_1)$ will also have a delta distribution concentrated on $p(\boldsymbol{y}|\hat{\boldsymbol{X}}_1)$, and the $\gamma_1^1 \approx 0$ finally. In conclusion, this bias should be controlled by

$$|\log \mathbb{E}_{\boldsymbol{X}_1}[p(\boldsymbol{y}|\boldsymbol{X}_1)] - \mathbb{E}_{\boldsymbol{X}_1}[\log p(\boldsymbol{y}|\boldsymbol{X}_1)]| \leq \frac{1}{\sqrt{2\pi}\gamma} \tag{S.31}$$

Although this bias is theoretically bounded, it still results in a slight degradation in performance. Table S.5 shows the comparison for the Lorenz experiment to illustrate this point.

**Hyperparameters for Monte Carlo Sampling**  We examine the estimation process and associated hyperparameters in Equation (S.28), where the expectation is computed using a Monte Carlo method. The hyperparameter $J$, is referred to as the number of Monte Carlo sampling iterations in Algorithm 2. It is important to note that increasing $J$ does not lead to an increase in neural network evaluations but only involves additional Gaussian noise simulations, which are computationally lightweight. To illustrate the effect of $J$, we use numerical results from the Lorenz experiment. As shown in Table S.6, increasing $J$ can improve performance by approximately 20%, with negligible impact on computational time.

| $J =$ | 3 | 6 | 12 | 21 | 30 | 50 |
|---|---|---|---|---|---|---|
| RMSE | 0.167 | 0.148 | 0.153 | 0.142 | 0.150 | 0.138 |

Table S.6: **Effect of $J$**. The RMSE of generated state trajectories for FlowDAS is evaluated with different Monte Carlo sampling times $J$ in Equation (S.28) for the Lorenz 1963 experiment. As $J$ increases, the RMSE initially decreases, indicating improved performance, and then stabilizes.

### C.2  Posterior Estimation Methods

We also evaluate the impact of different methods for posterior estimation: as defined in Equations (9) and (10). The results are presented in Table S.7, where "1st-order" and "2nd-order" refer to 1st-order Milstein method and 2nd-order stochastic Runge-Kutta method, respectively. "No correction" indicates forecasting purely based on the model without incorporating observation information. The results show that both 1st-order and 2nd-order estimations provide reasonable accuracy. However, the 2nd-order estimation consistently delivers better performance. This suggests that employing more accurate estimations of $p(\boldsymbol{X}_1|\boldsymbol{X}_s, \boldsymbol{X}_0)$ can effectively enhance model performance. Beyond the 2nd-order method, higher-order approaches like the Runge-Kutta 4th-order (RK4) method could further improve accuracy. However, these methods come with increased computational cost: 2nd-order estimation requires two neural network evaluations per step compared to one for 1st-order estimation, while RK4 requires four evaluations per step. In our experiments, we find that the 2nd-order estimation strikes a good balance between performance and efficiency, making it a practical choice. Further exploration of higher-order methods will be left for future research.

|  | 1ST-ORDER | 2ND-ORDER | NO CORRECTION |
|---|---|---|---|
| $32^2 \to 128^2$ | 0.048 | **0.038** | 0.206 |
| $16^2 \to 128^2$ | 0.101 | **0.067** | 0.206 |

Table S.7: **Effect of posterior estimation**. The RMSE of vorticity estimate from FlowDAS is evaluated on the incompressible Navier-Stokes task using different posterior estimation methods as defined in Equations (9) and (10), and forecasting without observations (i.e., no correction). For the super-resolution tasks $16^2 \to 128^2$ and $32^2 \to 128^2$, both posterior estimation methods significantly outperform forecasting without observations. Among them, the 2nd-order method achieves the lowest RMSE.

## C.3 Particle Collapse and Robustness of Likelihood-based Sampling

The likelihood-based sampling in Equation (8) can, in principle, suffer from particle collapse, which is a well-known issue in high-dimensional importance sampling where a few weights dominate the normalization sum, leading to an effective sample size close to one. We provide further analysis and our mitigation strategy below.

**Observed behavior.** During the stochastic interpolation process, we observed that the likelihood score $\|\boldsymbol{y} - \mathcal{A}\boldsymbol{x}_s\|_2^2$ typically decreases as the sampling proceeds, but in some cases starts to increase sharply near the end of the trajectory (typically within the final 30 to 50 steps). This abnormal increase coincides with the particle weights $w_j$ becoming heavily concentrated on a single particle $\hat{\boldsymbol{X}}_1^{(j)}$, indicating a collapse of the Monte Carlo estimator.

Empirically, we observed two distinct regimes:

- When the likelihood term decreases as expected, the weights remain balanced across particles, ensuring stable estimation.
- When the likelihood begins to rise unexpectedly, one or two particles dominate the weights (approaching 100%), leading to collapse and potential divergence.

**Mitigation strategies.** To prevent collapse, we employ a two-step remedy:

- **Resampling:** When abnormally concentrated weights are detected, we immediately perform resampling to regenerate a balanced particle set. This is a standard correction technique in sequential importance sampling.
- **Alternative sampling strategy:** If resampling does not stabilize the process, we switch to the alternative sampling strategy detailed in Appendix C.1. Although this approach may yield slightly less precise estimates, it significantly enhances robustness and prevents divergence in high-dimensional settings.

## C.4 Alternative Likelihood Score Approximation

In the main formulation, the likelihood score term $\nabla \log p(\boldsymbol{y} \mid \boldsymbol{X}_0, \boldsymbol{X}_s)$ is estimated via Monte Carlo sampling over particles $\{\boldsymbol{X}_1^{(j)}\}_{j=1}^K$, which provides an unbiased approximation at the cost of potential particle collapse in high-dimensional settings. An alternative approach is to adopt a *DPS-type approximation* [42]:

$$\nabla \log p(\boldsymbol{y} \mid \boldsymbol{X}_0, \boldsymbol{X}_s) \approx \nabla \log p(\boldsymbol{y} \mid \hat{\boldsymbol{X}}_1), \quad \text{where } \hat{\boldsymbol{X}}_1 := \mathbb{E}[\boldsymbol{X}_1 \mid \boldsymbol{X}_0, \boldsymbol{X}_s].$$

This estimator replaces the Monte Carlo sampling with the conditional expectation $\hat{\boldsymbol{X}}_1$, thereby avoiding weight collapse but introducing a bias bounded by Jensen's inequality.

To assess the effect of this approximation, we directly substituted the DPS-type term into Equation (8) of the main paper and re-ran the weather forecasting experiment. The quantitative comparison is summarized in Table S.8.

The DPS estimate can mitigate collapse and sometimes improve numerical stability. However, the denominator in Equation (S.34) (see derivation below) can approach zero when $s \to 0$, causing the

| METHOD | RMSE $\pm$ STD |
|---|---|
| FLOWDAS | $0.056 \pm 0.002$ |
| FLOWDAS-DPS | $0.050 \pm 0.002$ |

Table S.8: **Comparison of likelihood approximations on the weather forecasting task.** While the DPS approximation slightly improves RMSE, it introduces theoretical bias due to Jensen's gap and requires additional stabilization.

optimization to become unstable. To prevent this, we select a cutoff value $s_c$ and apply the DPS-type update only for $s > s_c$. In the weather forecasting experiment, we set $s_c = 0.04$ via grid search. This parameter may vary across datasets and is problem dependent, with no closed-form theoretical criterion for selection. Further tuning of parameters $(\alpha_s, \beta_s, \sigma_s)$ could improve the method, which we leave for future work.

Overall, FlowDAS-DPS provides a biased yet more stable alternative that complements our main FlowDAS formulation and demonstrates the flexibility of the framework.

**Short derivation of FlowDAS-DPS.** To derive the DPS-type approximation, we notice that

$$\boldsymbol{b}_s(\boldsymbol{X}_s, \boldsymbol{X}_0) = \dot{\alpha}_s \boldsymbol{X}_0 + \dot{\beta}_s \mathbb{E}[\boldsymbol{X}_1 \mid \boldsymbol{X}_0, \boldsymbol{X}_s] + \sqrt{s}\dot{\sigma}_s \mathbb{E}[\boldsymbol{z} \mid \boldsymbol{X}_0, \boldsymbol{X}_s], \tag{S.32}$$

Besides,

$$\boldsymbol{X}_s = \alpha_s \boldsymbol{X}_0 + \beta_s \mathbb{E}[\boldsymbol{X}_1 \mid \boldsymbol{X}_0, \boldsymbol{X}_s] + \sqrt{s}\sigma_s \mathbb{E}[\boldsymbol{z} \mid \boldsymbol{X}_0, \boldsymbol{X}_s]. \tag{S.33}$$

By combining the above two equations, we obtain

$$\mathbb{E}[\boldsymbol{X}_1 \mid \boldsymbol{X}_0, \boldsymbol{X}_s] = \frac{\sigma_s b_s(\boldsymbol{X}_s, \boldsymbol{X}_0) - \dot{\sigma}_s \boldsymbol{X}_s - (\sigma_s \dot{\alpha}_s - \alpha_s \dot{\sigma}_s)\boldsymbol{X}_0}{\sigma_s \dot{\beta}_s - \beta_s \dot{\sigma}_s}, \tag{S.34}$$

## C.5 Ablation on Conditioning Horizon

To study the effect of the conditioning horizon, we perform an ablation by varying the number of observed past states used during inference. We evaluate performance using the Continuous Ranked Probability Score (CRPS), a proper scoring rule that jointly measures accuracy and uncertainty calibration of probabilistic forecasts—the lower the CRPS, the better the performance.

| NAVIER–STOKES: CRPS VS. CONDITIONING HORIZON | | | | |
|---|---|---|---|---|
| # CONDITIONING STATES | 10 | 6 | 3 | 1 |
| CRPS | **0.538** | 0.634 | 0.663 | 0.776 |

| SEVIR: CRPS VS. CONDITIONING HORIZON | | |
|---|---|---|
| # CONDITIONING STATES | 6 | 3 | 1 |
| CRPS | **0.015** | 0.021 | 0.021 |

Table S.9: **Effect of conditioning horizon.** Longer conditioning horizons consistently improve CRPS, indicating better probabilistic forecasting performance.

The results clearly show that incorporating a longer conditioning horizon leads to consistent improvements in forecast accuracy and uncertainty estimation. In the Navier–Stokes task, extending the horizon from 1 to 10 steps yields approximately a 30% reduction in CRPS, highlighting the benefit of temporal context in capturing complex dynamics. Similarly, in the SEVIR precipitation forecasting task, conditioning on six previous frames achieves the best predictive accuracy. These findings confirm that longer temporal conditioning enables FlowDAS to leverage sequential dependencies more effectively, which is crucial for accurate data assimilation in high-dimensional and nonlinear dynamical systems.

| MODEL | PARAMS (M) | TRAINING TIME (HR) | INFERENCE TIME (S/SAMPLE) |
|---|---|---|---|
| FLOWDAS | 30.0 | 22 | 30.9 |
| PDEDIFF | 22.9 | 45 | 28.9 |
| TRANSOLVER | 11.2 | 80 | 12.7 |
| FNO | 1880.0 | 80 | 2.4 |

Table S.10: Comparison of model size, total training time, and inference time on the weather forecasting task using a single A100 GPU.

| METHOD | COMPUTATION TIME (S) | RMSE $\pm$ STD | MAX MEMORY (GB) |
|---|---|---|---|
| 1ST ORDER | 17.100 | $0.0490 \pm 0.0026$ | 1.8 |
| 2ND ORDER | 30.871 | $0.0485 \pm 0.0035$ | 3.9 |
| HYBRID | 205.271 | $0.0478 \pm 0.0033$ | 70.0 |

Table S.11: **Accuracy–speed trade-off for stochastic integrators.** The hybrid "2nd order $\times$ 400 + Flow $\times$ 100" improves RMSE only slightly (absolute gain 0.0007) but increases computation time by $\sim 7\times$ and peak memory by $\sim 18\times$ compared with the second–order integrator.

## C.6 Runtime and Parameter Comparison

We report the model sizes, total training times, and inference costs for the SEVIR weather forecasting task using a single NVIDIA A100 GPU in Table S.10. FlowDAS achieves the shortest training time among all baselines, converging roughly twice as fast as PDEDiff and nearly four times faster than Transolver, despite having a comparable model size.

The iterative correction mechanism in FlowDAS introduces a modest increase in inference time compared to purely feedforward models. However, this process is essential for producing observation-consistent predictions and stabilizing long-horizon forecasts. Importantly, FlowDAS exhibits strong training efficiency, reaching convergence significantly faster than other baselines.

## C.7 Bias–efficiency trade-offs of stochastic integrators

Our stochastic integrators in Equations (9) and (10) approximate the observation–conditioned $\hat{X}_1$ and therefore introduce discretization bias. We quantify the accuracy–speed trade-off on the weather forecasting task by comparing three settings: a first–order integrator, a second–order integrator, and a hybrid method that follows the second–order scheme for the first 400 steps and then computes the full stochastic–interpolant path to the endpoint $X_1$ for the last 100 steps ("2nd order $\times$ 400 + Flow $\times$ 100"). The hybrid reduces bias but is much more costly.

We also considered a full-path variant ("Flow$\times$500") that computes $X_1$ exactly at *every* step by rolling out the leftward stochastic interpolant. This requires $500 + (500 + 499 + \cdots + 1) \approx 2.5 \times 10^5$ forward passes and retaining the entire computation graph to backpropagate $\log p(y \mid X_1)$, which leads to out-of-memory failures even on A100 GPUs. In contrast, the first– and second–order schemes provide stable inference with much lower cost while maintaining competitive accuracy.

These results shown in Table S.11 support our choice: conditioning on observations at each interpolation step delivers most of the DA benefit, whereas reducing the small integrator bias further yields only marginal accuracy gains at a very high computational and memory cost. We include this analysis and the table above in the appendix for clarity.

## D  Additional Experiment Results

This section provides the details of our experiments, including additional experiments and results.

### D.1 Double-well Potential Problem

#### D.1.1 The Double-well Potential System

In this experiment, we investigate a 1D tracking problem in a dynamical system driven by the double-well potential. The system is governed by the following stochastic dynamics:

$$\mathrm{d}x = -4x(x^2 - 1)\,\mathrm{d}t + \beta_d\,\mathrm{d}\xi_t \tag{S.35}$$

with observation model defined by $\mathcal{A}(x) = x^3$. The observations are given by

$$y = \mathcal{A}(x) + \eta = x^3 + \eta \tag{S.36}$$

where the stochastic force $\xi_t$ is a standard Brownian motion with diffusion coefficient of $\beta_d = 0.2$. $\eta$ is standard Gaussian noise with standard deviation of $0.2$. The $\Psi$ is the derivative of the potential $U(x) = x^4 - 2x^2 + f$, where $f$ is a function independent of $x$. The system describes a particle trapped in the wells located at $x = 1$ and $x = -1$, with small fluctuations around these points, as illustrated in Figure S.6a.

#### D.1.2 Training and Testing Data Generation

We trained the model on simulated trajectories generated by numerically solving the transition equation using a temporal step size of $0.1$. The training dataset consisted of $500$ trajectories, each of length $100$, with initial points uniformly sampled from the range $[-2, 2]$. In the testing stage, we introduced stronger turbulence to the system, causing the particle to occasionally switch wells $(x \to -x)$.

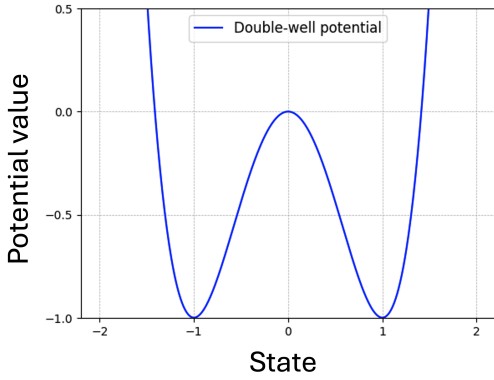
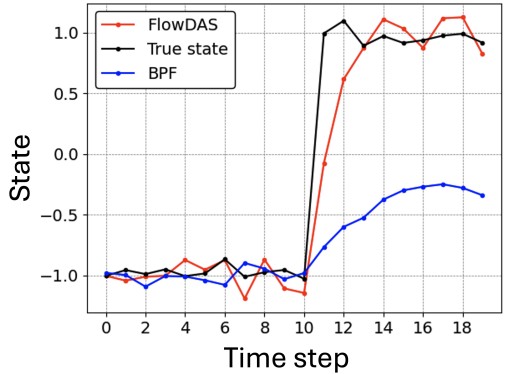

(a) **Illustration of the Double-Well Potential Problem**. In the double-well potential system, particles are typically captured at the bottoms of the wells ($x = 1$ and $x = -1$). With low probability, particles can transition between wells due to the stochastic term in the transition equation.

(b) **Visualization Results of Double-Well Potential Problem**. We show the visualization results of Flow-DAS and BPF, while the score-based solver SDA fails to produce reasonable results. FlowDAS can track the dramatic change of the particles that BPF struggles to immediately react to.

Figure S.6: Illustrations and visualization results for the double-well potential problem.

#### D.1.3 Neural Network for Learning the Drift Velocity

In this low-dimensional double-well potential task, the drift velocity $\boldsymbol{b}_s$ is approximated using a fully connected neural network with 3 hidden layers, each having a hidden dimension of 50. Both the input and output dimensions of the network are 1. For the condition $\boldsymbol{X}_0$ and timestep $s$, we empirically find that embedding $\boldsymbol{X}_0$, similar to how $s$ is embedded, outperforms directly using $\boldsymbol{X}_0$ as an additional input to the network. The intuition behind this approach is that embedding $\boldsymbol{X}_0$ helps the network better distinguish between the two variables, $\boldsymbol{X}_s$ and $\boldsymbol{X}_0$. The model is trained using the Adam optimizer with a base learning rate of $0.005$, along with a linear rate scheduler. Training is conducted for 5000 epochs.

### D.1.4 Hyperparameters During Inference

Hyperparameters of the inference procedure, specified for the double-well potential task, are presented in Table S.4.

### D.1.5 Baseline

For this task, we compare our method with SDA and the classic BPF method. For SDA, we fix the window size to two and use the same training data as FlowDAS to ensure fairness. The local score network is implemented as a fully connected neural network, following the architecture proposed in [50]. The score network consists of 3 hidden layers, each with a hidden dimension of 50. The model is trained using the AdamW optimizer with a base learning rate of $0.001$, a weight decay of $0.001$, and a linear learning rate scheduler. Training is conducted for 5000 epochs. For the BPF method, the particle density is set to $16384$.

### D.1.6 Results

The visual results are shown in Figure S.6b. Surprisingly, while both FlowDAS and the classic BPF method produce reasonable results, FlowDAS demonstrates superior performance in tracking dramatic changes in the trajectory. In contrast, the score-based solver SDA fails to produce reasonable results. This failure arises because the starting point of SDA during optimization is purely Gaussian noise, leading to a poor initial estimation of the target state. Furthermore, the cubic observation model amplifies the differences, causing the optimization gradients to explode and resulting in recovery failure.

In FlowDAS, the error is bounded by $||\boldsymbol{y}_{k+1} - \mathcal{A}(\boldsymbol{x}_k)||_2^2$ because the generation process begins with the previous state, which serves as a proximal estimate of the target state. Consequently, FlowDAS undergoes a more stable optimization process.

Compared to the classic BPF method, we find that it struggles to capture dramatic changes in the trajectory. This limitation is primarily due to the small diffusion coefficient $\beta_d$, such as $\beta_d = 0.2$, which causes the predicted filtering density in BPF to concentrate around the mean value dictated by the deterministic part of the transition equation. As a result, extreme cases lying in the tail of the future state distribution $p(\boldsymbol{x}_{k+1} \mid \boldsymbol{x}_k)$ are often missed. This phenomenon can be explained by the truncation error arising from the finite particle space [50].

In contrast, FlowDAS is capable of immediately sampling from the true conditional distribution as the steps become sufficiently large. This allows FlowDAS to better capture the tail region of the true distribution and react to dramatic changes, even those in low-probability areas. Additionally, FlowDAS can effectively incorporate observation information, enabling it to balance prediction and observation even when the true underlying state occurs in a low-probability region.

## D.2 Lorenz 1963

### D.2.1 The Lorenz 1963 Dynamic System

To simulate this system, we use the RK4 method, which updates the solution from time $t_n$ to $t_{n+1} = t_n + h$ using the following formulas for each variable $a$, $b$, and $c$:

$$
\begin{aligned}
k_{1a_n} &= h \cdot \mu(b_n - a_n), \\
k_{1b_n} &= h \cdot \left( a_n(\rho - c_n) - b_n \right), \\
k_{1c_n} &= h \cdot \left( a_n b_n - \tau c_n \right),
\end{aligned}
\tag{S.37}
$$

$$
\begin{aligned}
k_{2a_n} &= h \cdot \mu \left( (b_n + \frac{k_{1b_n}}{2}) - (a_n + \frac{k_{1a_n}}{2}) \right), \\
k_{2b_n} &= h \cdot \left( (a_n + \frac{k_{1a_n}}{2})(\rho - (c_n + \frac{k_{1c_n}}{2})) - (b_n + \frac{k_{1b_n}}{2}) \right), \\
k_{2c_n} &= h \cdot \left( (a_n + \frac{k_{1a_n}}{2})(b_n + \frac{k_{1b_n}}{2}) - \tau(c_n + \frac{k_{1c_n}}{2}) \right),
\end{aligned}
\tag{S.38}
$$

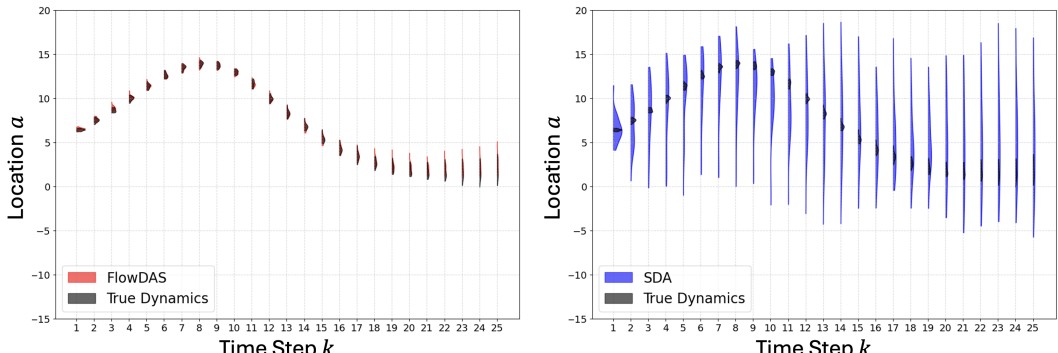

Figure S.7: **Forecaseting dynamics modeling evaluation in the Lorenz 1963 task: FlowDAS vs. SDA**. FlowDAS produces results that closely match the true states, demonstrating its ability to learn the underlying transition dynamics $p(\boldsymbol{x}_{k+1} \mid \boldsymbol{x}_k)$. In contrast, SDA exhibits rapid divergence from the true states. This divergence arises because SDA focus on modeling the joint distribution $p(\boldsymbol{x}_{k+1}, \boldsymbol{x}_k)$ rather than directly learn the transition dynamics $p(\boldsymbol{x}_{k+1} \mid \boldsymbol{x}_k)$, which is inherently less suitable for capturing the system's underlying dynamics.

$$k_{3a_n} = h \cdot \mu \left( (b_n + \frac{k_{2b_n}}{2}) - (a_n + \frac{k_{2a_n}}{2}) \right),$$

$$k_{3b_n} = h \cdot \left( (a_n + \frac{k_{2a_n}}{2})(\rho - (c_n + \frac{k_{2c_n}}{2})) - (b_n + \frac{k_{2b_n}}{2}) \right), \tag{S.39}$$

$$k_{3c_n} = h \cdot \left( (a_n + \frac{k_{2a_n}}{2})(b_n + \frac{k_{2b_n}}{2}) - \tau(c_n + \frac{k_{2c_n}}{2}) \right),$$

$$k_{4a_n} = h \cdot \mu \left( (b_n + k_{3b_n}) - (a_n + k_{3a_n}) \right),$$
$$k_{4b_n} = h \cdot \left( (a_n + k_{3a_n})(\rho - (c_n + k_{3c_n})) - (b_n + k_{3b_n}) \right), \tag{S.40}$$
$$k_{4c_n} = h \cdot \left( (a_n + k_{3a_n})(b_n + k_{3b_n}) - \tau(c_n + k_{3c_n}) \right).$$

Then, the updates for $a_n$, $b_n$, and $c_n$ are:

$$a_{n+1} = a_n + \frac{1}{6}(k_{1a_n} + 2k_{2a_n} + 2k_{3a_n} + k_{4a_n}),$$

$$b_{n+1} = b_n + \frac{1}{6}(k_{1b_n} + 2k_{2b_n} + 2k_{3b_n} + k_{4b_n}), \tag{S.41}$$

$$c_{n+1} = c_n + \frac{1}{6}(k_{1c_n} + 2k_{2c_n} + 2k_{3c_n} + k_{4c_n}).$$

After solving these deterministic updates, the stochastic force $(\xi_1, \xi_2, \xi_3)$ is added to $(a_{n+1}, b_{n+1}, c_{n+1})$, which leads to the Lorenz 1963 system dynamic equations described in Equation (11).

### D.2.2   Training and Testing Data Generation

We apply the RK4 method described in Appendix D.2.1 to generate the simulated training and testing data.

### D.2.3   Hyperparameters During Inference

Hyperparameters of the inference procedure, specified for the Lorenz 1963 task, are presented in Table S.4. Noticeably, the step size $\zeta$ is smaller than in other experiments due to the chaotic nature of the Lorenz 1963 system, which requires finer step size to accurately capture its dynamics.

### D.2.4   Neural Network for Learning the Drift Velocity

In this task, we use a fully connected neural network with 5 hidden layers, each with a hidden dimension of 256, to approximate the velocity field $\boldsymbol{b}_s$. The input and output dimensions are both 3.

For the condition $X_0$ and timestep $s$, we use embeddings of dimension 4. The model is trained using the Adam optimizer with a base learning rate of 0.005, and a linear learning rate scheduler is applied. Training is conducted over 23000 epochs.

### D.2.5 Baseline and Additional Results

We compare FlowDAS with SDA and BPF. For the SDA, we use a score neural network with a fixed window size of 2. This local score function is implemented using a fully connected neural network with 5 hidden layers, each having a hidden dimension of 256. The model is trained using the AdamW optimizer with a base learning rate of 0.001 and a linear scheduler over 23000 epochs. For BPF, the particle density is set to 16384.

We also implemented EnSF [58] and its performance is on par with BPF. This is expected since EnSF uses a training-free Monte Carlo estimator for the score function, which faces the same particle collapse issues, i.e., samples concentrate in high-probability regions of the transition dynamics, leading to biased score estimates. This reflects a train-sampling tradeoff: EnSF saves training time but cannot learn accurate scores, especially under highly nonlinear dynamics and observation models like in our noisy Lorenz-63 settings(e.g., our $atan(x_1)$ observation). FlowDAS, by learning the drift with collected data, avoids this issue and achieves superior performance. For the EnSF, we largely follow the official implementation and adopt the settings summarized below. The ensemble size is set to 20000 to ensure stable Monte Carlo estimation of the ensemble mean and covariance. The diffusion SDE is integrated using 500 steps with the Euler solver. Both $\varepsilon_a$ and $\varepsilon_b$ are fixed to 0.001. The initial data variance is set to zero, and no covariance inflation is applied (inflation factor $= 1$).

| METHOD | FLOWDAS | SDA | BPF | ENSF |
|--------|---------|-----|-----|------|
| RMSE | **0.202** | 1.114 | 0.270 | 0.298 |

Table S.12: **Comparison of FlowDAS with diffusion-based and ensemble-based filters.** Reported RMSEs are computed on the same problem setting as described in the Section 4.1. FlowDAS achieves the lowest RMSE, demonstrating robustness under non-Gaussian dynamics and non-linear observation observations.

### D.2.6 Accuracy of the Learned Dynamics

We evaluate the dynamic learning performance of FlowDAS and SDA, focusing on their ability to forecast future states **without** using observations. For BPF, we do not evaluate its performance in this context since it explicitly incorporates the true system dynamics.

As SDA is designed to rely on observations and does not work in an auto-regressive manner, we adapt it for this evaluation by using the previous state as a pseudo-observation. The SDA then forecasts the next state based on this input. For FlowDAS, we disable the observation-based update step by setting the step size $\zeta_n = 0$.

Both methods are initialized with the same initial state, and we simulate 64 independent trajectories of length 25. True states are generated by solving Equation (11) using the RK4 method, starting from the same initial point. The visualization of location $a$ across 25 time steps is shown in Figure S.7.

### D.3 Incompressible Navier-Stokes Flow

### D.3.1 Incompressible Navier-Stokes Flow Problem Settings

For the incompressible Navier-Stokes flow problem, we adopted the problem setting from [31] and used the provided training data. The choices of experiment parameters are as follows: $\nu = 10^{-3}$, $\alpha = 0.1$, $\varepsilon = 1$. The stochastic force $\xi$ is defined as:

$$\xi(t, x, y) = W_1(t)\sin(6x) + W_2(t)\cos(7x) + W_3(t)\sin(5(x + y)) + W_4(t)\cos(8(x + y))$$
$$+ W_5(t)\cos(6x) + W_6(t)\sin(7x) + W_7(t)\cos(5(x + y)) + W_8(t)\sin(8(x + y))$$
$$\text{(S.42)}$$

For more details and insights about this problem, see [31].

### D.3.2 Hyperparameters During Inference

Table S.4 presents the hyperparameters of the inference procedure for the incompressible Navier-Stokes flow task.

### D.3.3 Neural Network for Learning the Drift Velocity

We use a U-Net architecture to approximate $b_s$, following the network proposed in [31]. The conditioning on $X_0$ is implemented through channel concatenation in the input. The architectural details are as follows:

- Number of states conditioned on: 10.
- Number of initial channels: 128.
- Multiplication factor for the number of channels at each stage: (1, 2, 2, 2).
- Number of groups for group normalization in ResNet blocks: 8.
- Dimensionality of learned sinusoidal embeddings: 32.
- Dimensionality of each attention head in the self-attention mechanism: 64.
- Number of attention heads in the self-attention layers: 4.

We employ the AdamW optimizer [59] with a cosine annealing schedule to reduce the learning rate during training. The base learning rate is set to $2 \times 10^{-4}$. Training is conducted with a batch size of 32 for 1000 epochs.

### D.3.4 Baseline

We compare FlowDAS to FNO-DA, Transolver-DA, and PDEDiff.

**FNO-DA** The model is configured with 7 stacked Fourier layers. Each layer processes data in a 128-dimensional feature space and truncates the Fourier series to include 6 modes in each spatial direction. A padding of 6 is applied before the Fourier operations and removed afterward. The model is provided with the previous 10 states to predict the state at the next time step. We train the model for 1000 epochs, using a batch size of 64 and the AdamW optimizer with an initial learning rate of 0.0001.

**Transolver-DA** The model has 8 layers of transolver blocks whose hidden channels are 128 and have 32 slices, with 8 attention heads. The model is provided with the previous 10 states to predict the state at the next time step. We train the model for 1000 epochs using a batch size of 64, the AdamW optimizer, and a OneCycleLR learning rate scheduler with an initial learning rate of $5e^{-4}$.

**PDEDiff** The score neural network is configured with a temporal window of 11, of which the first 10 inputs are conditions and the forecast horizon is 1; an embedding layer dimensionality of 32; and a base number of feature channels 256. The network depth is 5, and the activation function used is SiLU [59]. We train the model using a batch size of 64 and the AdamW optimizer, with a linear learning rate scheduler (initial learning rate 0.001) and a weight decay of 0.001.

During inference, the $\alpha_1$ and $\alpha_2$ used is varied from task to task for both the FNO-DA and the Transolver-DA. We show the detailed parameter settings in Tables S.13 and S.14. We use an AdamW optimizer for Kalman update with a learning rate of $1e^{-4}$ and a maximum iteration time of 2000 to ensure the loss converges. Without other indications, we adopt the same optimizer settings for both FNO-DA and Transolver-DA.

### D.3.5 Kinetic Energy Spectrum Analysis

We evaluate the methods from a physics perspective using the kinetic energy spectrum. A better method produces results that align more closely with the kinetic energy spectrum of the true state. The kinetic energy spectrum is computed as follows:

$$E(k_n) = \sum_{\substack{p,q \\ k_{p,q} \in \text{bin } n}} E(k_{x_p}, k_{y_q}), \tag{S.43}$$

|  |  |  | $\alpha_1$ | $\alpha_2$ |
|---|---|---|---|---|
|  | SR | 4x | 0.63 | 1 |
| INCOMPRESSIBLE | SR | 8x | 1.59 | 1 |
| NAVIER STOKES | SO | 5% | 0.10 | 1 |
|  | SO | 1.5625% | 0.29 | 1 |
| WEATHER FORECASTING |  |  | 1.22 | 1 |

Table S.13: **Hyperparameters for FNO-DA in the inference stage** of all experiments presented in this study. SR stands for super-resolution task and SO represents sparse observation (inpainting) task

|  |  |  | $\alpha_1$ | $\alpha_2$ |
|---|---|---|---|---|
|  | SR | 4x | 0.53 | 1 |
| INCOMPRESSIBLE | SR | 8x | 1.21 | 1 |
| NAVIER STOKES | SO | 5% | 0.14 | 1 |
|  | SO | 1.5625% | 0.22 | 1 |
| WEATHER FORECASTING |  |  | 5.00 | 1 |

Table S.14: **Hyperparameters for Transolver-DA in the inference stage** of all experiments presented in this study. SR stands for super-resolution task and SO represents sparse observation (inpainting) task

where $k_{p,q}$ represents the wavenumbers grouped into bin $n$, $E(k_{x_p}, k_{y_q})$ is the kinetic energy at each wavenumber and $p,q$ are the index representing a specific discrete wavenumber along the $k_x$ or $k_y$ axis.

Since the direct outputs of both FlowDAS and PDEDiff are vorticity fields, it is necessary to first convert the vorticity into velocity before computing the kinetic energy spectrum. This conversion is achieved by solving the Poisson equation $\Delta\psi = -\omega$ to obtain $\psi$, and then calculating its gradient $\boldsymbol{v} = -\nabla\psi$ to derive the velocity $\boldsymbol{v}$.

We present the results of the kinetic energy spectrum analysis for the Navier-Stokes flow super-resolution and sparse observation tasks.

### D.3.6  Additional Results

In this section, we present additional results to Section 4.2, including the kinetic energy spectrum and visualization results for the Navier-Stokes flow super-resolution and sparse observation tasks.

$16^2 \rightarrow 128^2$ **Super-Resolution**   The kinetic energy spectrum is shown in Figure S.8 (a), and additional visualization results are provided in Figure S.9. FlowDAS effectively reconstructs high-resolution vorticity data from low-resolution observations, while PDEDiff struggles to capture high-frequency physics.

$32^2 \rightarrow 128^2$ **Super-Resolution**   The kinetic energy spectrum is shown in Figure S.8 (b), and additional visualization results are provided in Figure S.10. Similar to the $16^2 \rightarrow 128^2$ task, FlowDAS achieves superior alignment with the true state compared to PDEDiff.

$5\%$ **Sparse Observation**   The kinetic energy spectrum is presented in Figure S.8 (d), with additional visualization results in Figure S.11. FlowDAS accurately recovers the vorticity field despite the highly sparse observations, significantly outperforming PDEDiff, demonstrating strong recovery of physical information.

Across all tasks, FlowDAS consistently outperforms PDEDiff, demonstrating superior alignment with the true kinetic energy spectrum and effective recovery of the vorticity field. FlowDAS not only excels in super-resolution tasks but also handles sparse observation challenges with robustness, maintaining physical coherence and accurately capturing high-frequency dynamics.

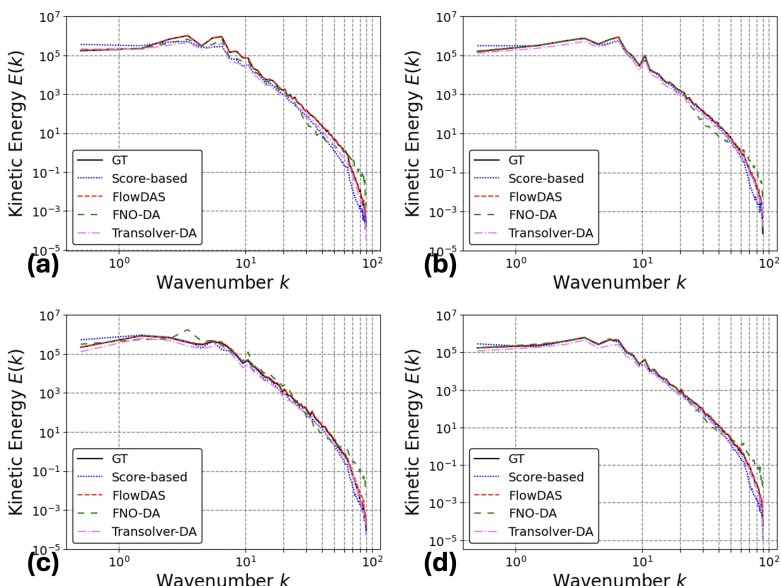

Figure S.8: **The kinetic energy spectrum of:** (a) super-resolution task: $16^2 \to 128^2$; (b) super-resolution task: $32^2 \to 128^2$; (c) sparse observation task: 1.5625%; (d) sparse observation task: 5%, in the incompressible Navier-Stokes flow task. We present the kinetic energy spectrum of the true state alongside the estimations from FlowDAS and baseline methods. FlowDAS can produce results that better aligned with the true state in terms of the kinetic energy spectrum, evidenced by the oscillations in the spectrum of baselines, indicating FlowDAS's superiority in recovering the physics information and effectiveness as a surrogate model for stochastic dynamic systems.

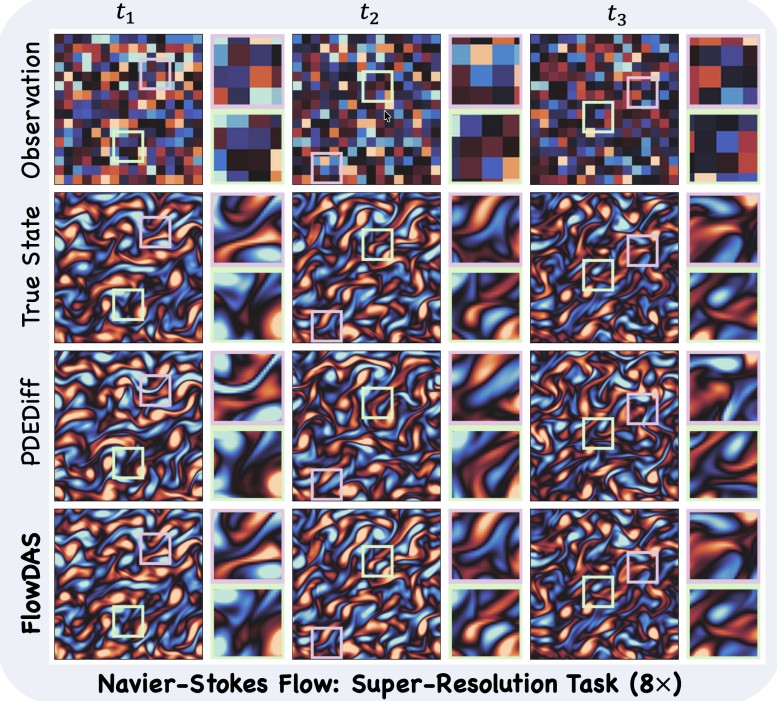

Figure S.9: **Data assimilation of incompressible Navier-Stokes flow—Super-Resolution task** (8×). Experiments were conducted at observation resolution of $16^2$. The goal of super-resolution task is to reconstruct high-resolution vorticity data ($128^2$) from low-resolution observations.

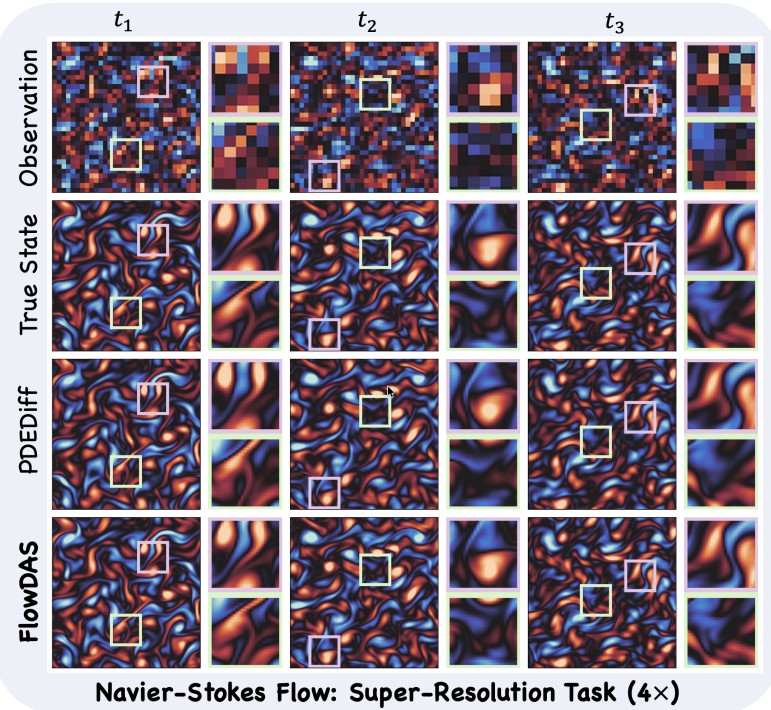

Figure S.10: **Data assimilation of incompressible Navier-Stokes flow—Super-Resolution task** ($4\times$). Experiments were conducted at observation resolution of $32^2$. The goal of super-resolution task is to reconstruct high-resolution vorticity data ($128^2$) from low-resolution observations.

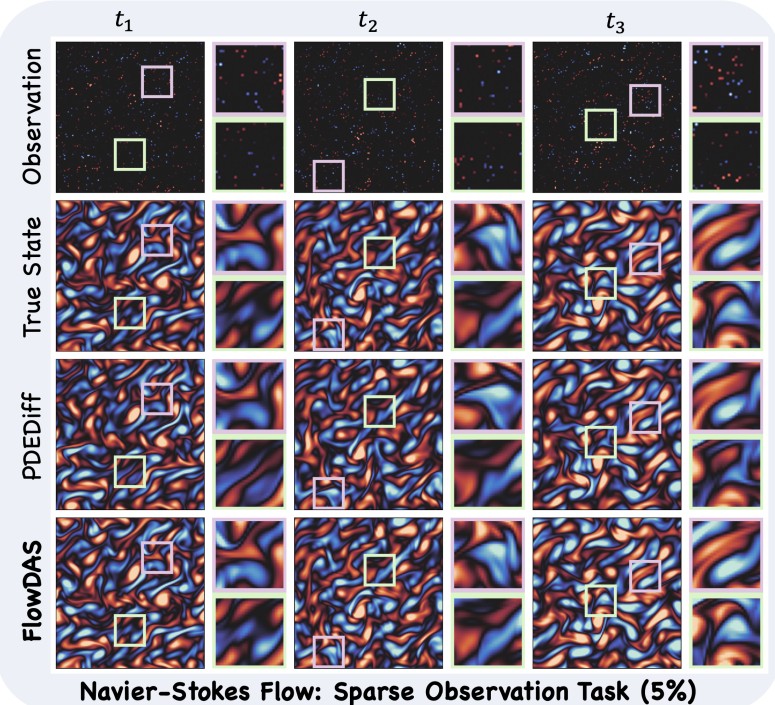

Figure S.11: **Data assimilation of incompressible Navier-Stokes flow—Sparse-Observation task** ($5\%$). Experiments were conducted at observation sparsity level at $5\%$. The goal of sparse observation task is to recover the complete vorticity field.

### D.3.7 Performance Degradation Under More Challenging Cases

In the main paper, FlowDAS was evaluated on the incompressible Navier–Stokes system under moderate super-resolution (SR) and sparse-observation (SO) conditions, specifically SR $4\times$, SR $8\times$, SO $5\%$, and SO $1.5625\%$. To evaluate the performance under more extreme conditions, we further test FlowDAS on SR $16\times$ and SO settings with observation coverages as low as $0.75\%$.

These results demonstrate a *graceful degradation* of performance as the observation coverage decreases or the upscaling factor increases, confirming the robustness of FlowDAS even under highly underdetermined measurement scenarios.

|  | SR $16\times$ | SO $1\%$ | SO $0.75\%$ |
|---|---|---|---|
| RMSE $\pm$ STD | $0.398 \pm 0.015$ | $0.395 \pm 0.016$ | $0.452 \pm 0.019$ |

Table S.15: **RMSE ($\pm$ std) of FlowDAS on Navier–Stokes under more challenging super-resolution and sparse-observation conditions.** Performance decreases smoothly as the observation ratio is reduced, supporting the stability and generalization of FlowDAS in limited-measurement regimes.

### D.3.8 Trajectory-level Evaluation

To further evaluate the temporal stability of FlowDAS, we examine its performance over full generated trajectories on the incompressible Navier–Stokes system. While the main results in Table 2 focus on single-step assimilation accuracy, this analysis quantifies how reconstruction errors and their variability evolve over multiple forecasting timesteps. At each timestep, RMSE and standard deviation are computed over 50 independently sampled initial states, each propagated through 30 stochastic realizations.

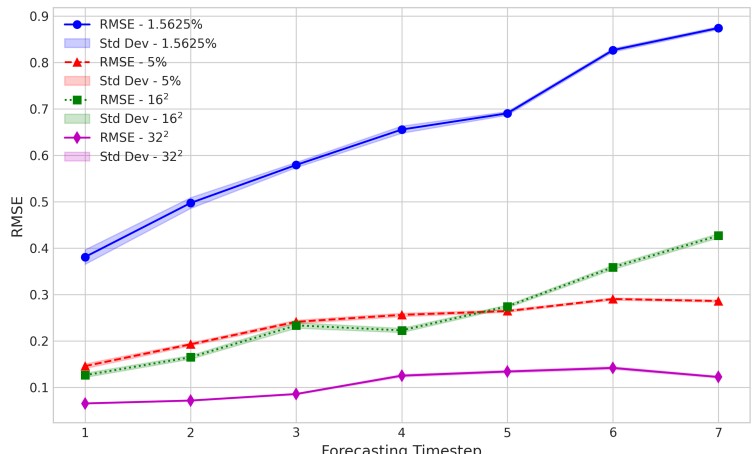

Figure S.12: **Trajectory-level RMSE and standard deviation of FlowDAS on the incompressible Navier–Stokes system.** The plot shows the evolution of RMSE (solid lines) and corresponding standard deviations (shaded areas) across forecasting timesteps. FlowDAS maintains stable performance over several assimilation steps, with gradual error accumulation consistent with stochastic propagation.

### D.4 Particle Image Velocimetry

This section presents a realistic application of our method: Particle Image Velocimetry (PIV). PIV is a widely used optical technique for measuring velocity fields in fluids, with many scientific applications in aerodynamics [60–62], biological flow studies [63–65], and medical research [66–68].

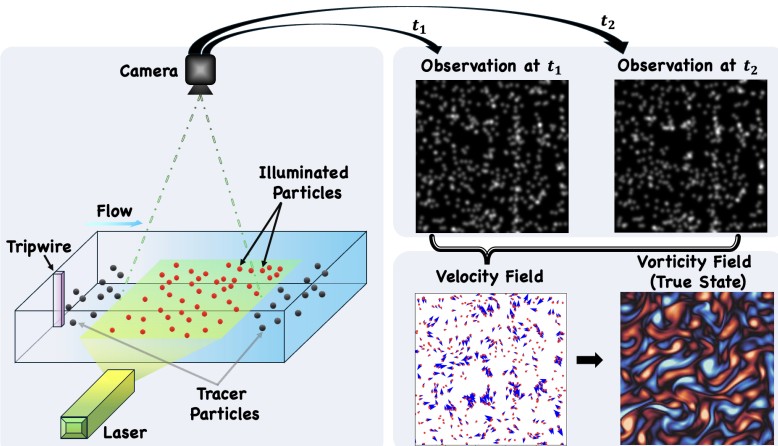

Figure S.13: **Illustration of the real-world Particle Image Velocimetry (PIV) experiment**: The flow is seeded with tracer particles illuminated by a laser sheet, and their movements are captured by a camera to derive the sparse velocity field. The goal here is to recover dense vorticity field from the sparse velocity measurement.

### D.4.1 Task Description

In a standard PIV setup, as shown in Figure S.13, fluorescent tracer particles are seeded into a fluid flowing through a channel with transparent walls. A laser sheet illuminates the fluid, and particle movements are recorded by a high-speed camera with adjustable temporal resolution. By analyzing the displacement of these tracer particles, the velocity field within the fluid can be determined at sparse locations. Unlike the task in Section 4.2, which involves recovering dense vorticity fields from sparse vorticity observations, PIV introduces a slightly different DA task: recovering dense vorticity fields ($x = \omega$) from sparse velocity measurements. This observation model is defined by

$$y = \mathcal{A}(v(\omega)) + \eta, \tag{S.44}$$

where the $\mathcal{A}(\cdot)$ sparsely samples the velocity field $v$ and the observation noise $\eta$ has a standard deviation of $\gamma = 0.25$. The relationship between the velocity $v$ and the state (vorticity) $\omega$ is given by $\omega = \nabla \times v$. To derive the velocity $v$ from the vorticity $\omega$, we first solve the Poisson equation $\Delta \psi = -\omega$ to obtain the stream function $\psi$, and then compute the velocity $v$ as the gradient of $\psi$. This process is performed using the Fast Fourier Transform.

### D.4.2 Dataset

In this experiment, we use the same fluid dynamics simulation data from the NS experiments described in Section 4.2. However, we convert the vorticity data to velocity fields via Fourier transform to create synthetic PIV datasets [69]. The particle positions in our simulation are randomly initialized and then perturbed according to the simulated flow motion pattern. In these synthetic images, we assume a particle density of 0.03 particles per pixel, a particle diameter of 3 pixels, and a peak intensity of 255 for each particle in grayscale. The images are processed through a standard PIV pipeline to extract particle locations, match corresponding particles across frames, and compute sparse velocity observations. These sparse measurements are then used in DA to reconstruct the full vorticity fields.

### D.4.3 Baselines

We compare our method against the SDA solver. Instead of training new scores or stochastic interpolant networks, we directly adopt the networks trained on vorticity data from the incompressible NS flow experiment to evaluate FlowDAS and SDA on the PIV task.

| Parameter | Value | Unit |
|---|---|---|
| Particle density | 0.03 | Particle per pixel |
| Particle diameter | 3 | Pixel |
| Peak intensity | 255 | Gray value |

Table S.16: **PIV task: parameters for simulation.** This table summarizes parameters for the PIV experiments in detail.

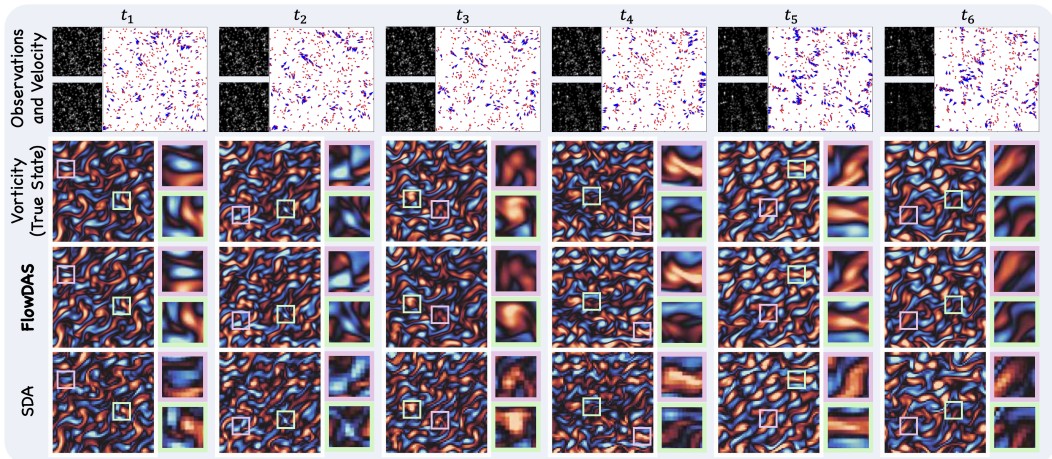

Figure S.14: **Data assimilation of Particle Image Velocimetry.** The vorticity field is visualized in the same way as in Section 4.2. FlowDAS outperforms SDA in terms of reconstruction fidelity and RMSE, recovering more detailed features even when direct observations are not available. These improvements highlight the potential of FlowDAS for real-world applications.

### D.4.4 Experiment Setting

The training data, neural network (including FlowDAS and SDA) is the same as those in the incompressible Navier-stokes flow simulation. Figure S.13 shows the standard PIV set up. Hyperparameters of the inference procedure, specified for the PIV task, are presented in Table S.4.

### D.4.5 Results

FlowDAS accurately reconstructs the underlying fluid dynamics from observed particle images, producing vorticity estimates with high precision. As shown in Figure S.14, it outperforms the baseline SDA in terms of reconstruction fidelity. Quantitatively, FlowDAS achieves a lower average RMSE of 0.118, compared to 0.154 for SDA. This performance gap highlights the robustness of FlowDAS and its potential utility in fluid dynamics applications.

## D.5 Weather Forecasting

### D.5.1 Baselines

We compared FlowDAS to FNO-DA, Transolver-DA, and PDEDiff.

**FNO-DA** The model is configured with 7 stacked Fourier layers. Each layer processes data in a 128-dimensional feature space and truncates the Fourier series to include 64 modes in each spatial direction. A padding of 6 is applied before the Fourier operations and removed afterward. The model is provided with the previous 6 states to predict the state at the next time step. We train the model for 1000 epochs, using a batch size of 200 and the AdamW optimizer with an initial learning rate of 0.0001.

| BACKBONE OF FLOWDAS | RMSE ↓ | CSI($\tau_{20}$) (0.3) ↑ | CSI($\tau_{40}$) (0.5) ↑ |
|---|---|---|---|
| FNO | **0.053±0.001** | **0.718±0.015** | **0.568±0.024** |
| TRANSOLVER | 0.056±0.002 | 0.702±0.018 | 0.540±0.028 |
| UNET | 0.056±0.002 | 0.703±0.017 | 0.540±0.023 |

Table S.17: Comparison of backbone network architectures of the drift model on the weather forecasting task. All metrics are averaged over multiple runs. Subtle improvements are observed when using a FNO network as the backbone of the drift model.

**Transolver-DA**   The model has 8 layers of transolver blocks whose hidden channels are 128 and have 32 slices, with 8 attention heads. The model is provided with the previous 6 states to predict the state at the next time step. We train the model for 1000 epochs using a batch size of 200, the AdamW optimizer, and a OneCycleLR learning rate scheduler with an initial learning rate of $5e^{-4}$.

**PDEDiff**   The score neural network is configured with a temporal window of 7, of which the first 6 inputs are conditions, and the forecast horizon is 1; an embedding layer dimensionality of 32; and a base number of feature channels 256. The network depth is 5, and the activation function used is SiLU [59]. We train the model using a batch size of 200 and the AdamW optimizer, with a linear learning rate scheduler (initial learning rate 0.001) and a weight decay of 0.001.

During inference, the $\alpha_1$ and $\alpha_2$ used is varied from task to task for both the FNO-DA and the Transolver-DA. We show the detailed parameter settings in Tables S.13 and S.14. We use an AdamW optimizer for Kalman update with a learning rate of $1e^{-4}$ and a maximum iteration time of 2000 to ensure the loss finally becomes stable. Without other indications, we adopt the same optimizer settings for both FNO-DA and Transolver-DA.

### D.5.2   Training Details

We experiment with three backbone architectures—U-Net, FNO, and Transolver—to learn the drift term and compare their performance and provide numerical results in Table S.17. The architectural details are as follows:

- UNet: as shown in Appendix D.3.3, the only change is the number of states conditioned on. Here we condition on previous 6 states.

- FNO: this model is configured with 7 stacked Fourier layers. Each layer processes data in a 128-dimensional feature space and truncates the Fourier series to include 64 modes in each spatial direction. A padding of 6 is applied before the Fourier operations and removed afterward. The model is provided with the previous 6 states, the $X_s$ and the time $s$ as additional channel concatenated to the previous states and the $X_s$ to predict the drift term.

- Transolver: this model has 5 layers of transolver blocks whose hidden channels are 128 and have 32 slices, with 8 attention heads. The sinusoidal timestep embeddings are applied. The inputs are the previous 6 states, the $X_s$ and the time $s$ and the output is the drift term.

### D.5.3   Results

We present additional visualizations to demonstrate FlowDAS's ability of capturing unknown system dynamics on this weather forecasting task in Figures S.15 to S.17. FlowDAS successfully generates accurate estimates of the weather in the future frames in an autoregressive manner. Table 3 provides numerical results of FlowDAS as well as other baseline methods. FlowDAS consistently outperformed DA baselines both in terms of RMSE and CSI metrics.

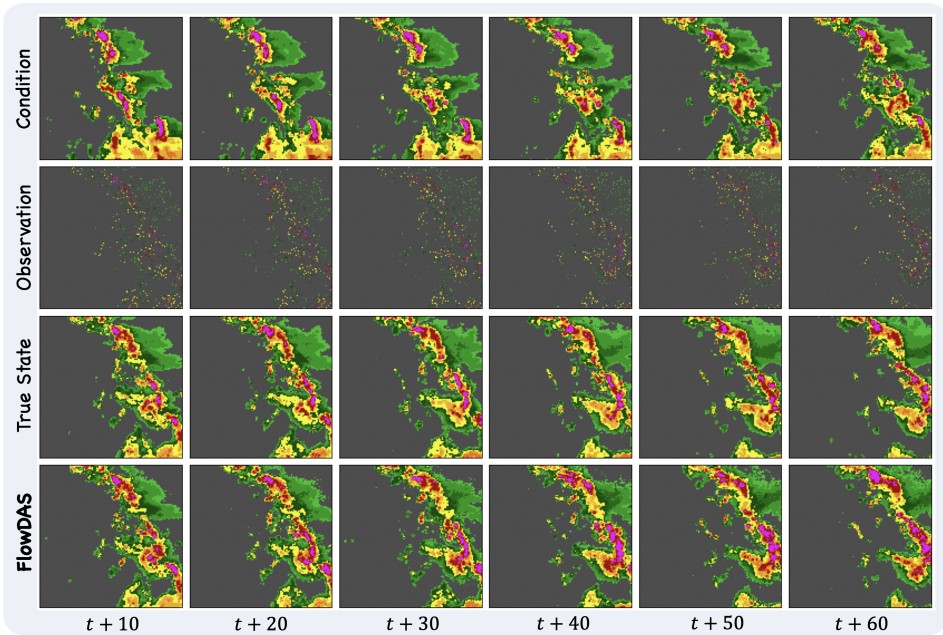

Figure S.15: **Additional result 1: Data assimilation of weather forecasting (Sparse-Observation task)**. The underlying PDE of the dynamical system is unknown. This experiment was conducted at an observation sparsity level of $10\%$, showing the capacity of long-trajectory forecasting.

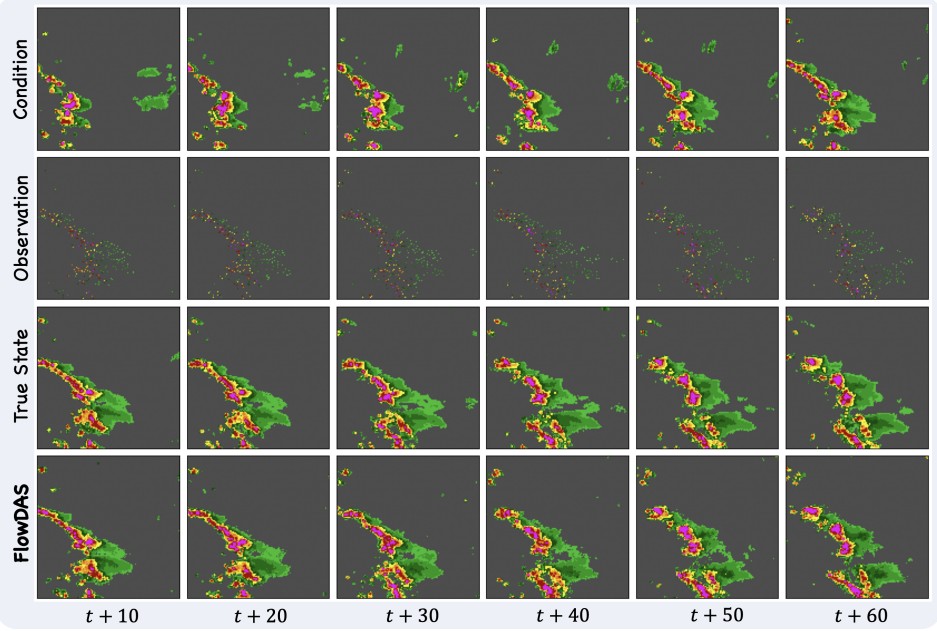

Figure S.16: **Additional result 2: Data assimilation of weather forecasting (Sparse-Observation task)**. The underlying PDE of the dynamical system is unknown. This experiment was conducted at an observation sparsity level of $10\%$, showing the capacity of long-trajectory forecasting.

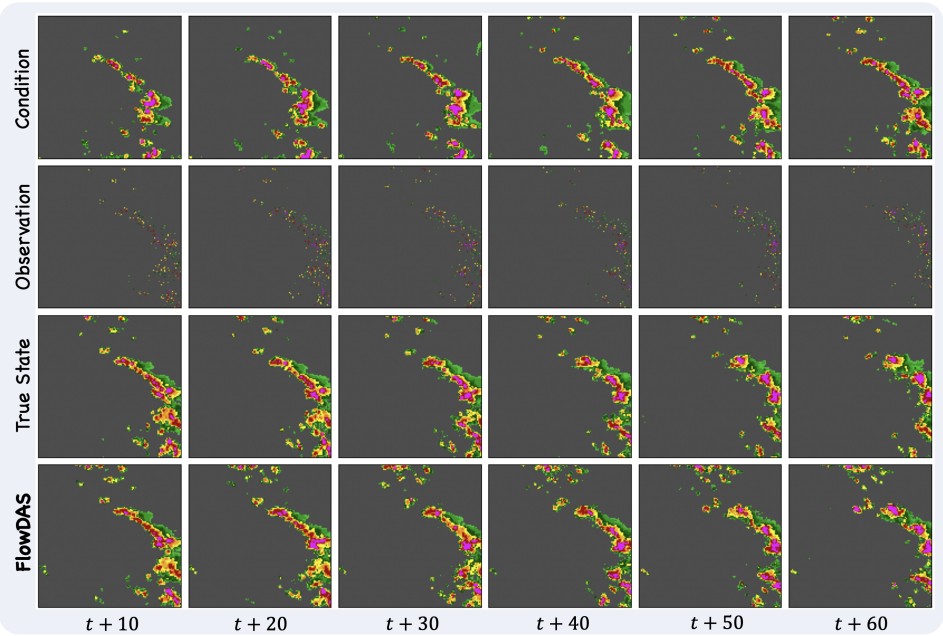

Figure S.17: **Additional result 3: Data assimilation of weather forecasting (Sparse-Observation task)**. The underlying PDE of the dynamical system is unknown. This experiment was conducted at an observation sparsity level of $10\%$, showing the capacity of long-trajectory forecasting.

