# OpenReview forum: "FlowDAS: A Stochastic Interpolant-based Framework for Data Assimilation"
_NeurIPS.cc/2025/Conference — NeurIPS 2025 poster_

### Official Review · Reviewer_RCK2 · 2025-06-25

**Clarity:** 3
**Significance:** 3
**Originality:** 2
**Rating:** 4
**Confidence:** 4

**Summary:**

This paper propose FlowDAS, a stochastic interpolant-based framework for data assimilation in high-dimensional, partially observed dynamical systems. It unifies probabilistic forecasting and observation assimilation by modeling local state transitions and incorporating measurement consistency. The paper is well-structured and clearly written, with a concise and transparent presentation of the method. The approach is applied to a variety of tasks, including Lorenz-63, Navier–Stokes flows, and real-world weather forecasting.

**Questions:**

1. **On the implementation of 3D-Var in high-dimensional settings (Line 261):**
    The variational assimilation baseline approximates the forecast error term using an L2 norm: $\alpha_1 \|x - x^\prime_{k+1}\|^2_2$. While this may be reasonable for low-dimensional systems, it ignores the covariance structure of forecast errors in high-dimensional cases such as Navier–Stokes or weather forecasting. This simplification may degrade assimilation quality, as the actual forecast errors often exhibit complex cross-correlations. More refined covariance approximations such as those used in GEN_BE [1,2,3] could provide a stronger baseline. Since both diffusion- and interpolant-based AI methods aim to model these dependencies, it would be more meaningful to compare them against improved variational baselines.

2. **On computational cost comparisons:**

   Traditional variational methods (e.g., 3D-Var) are often computationally efficient. It would be helpful to provide a comparison of computational complexity or runtime between FlowDAS, SDA, and classical variational methods—even if FlowDAS is more expensive—so that readers can better understand the cost-accuracy trade-offs.

3. **On ensemble sampling and uncertainty modeling in inference:**

   I may have misunderstood, but although the method claims to support sampling-based probabilistic inference, it appears that only a single analysis sample $\hat{x}_k$ is used at each step during inference (as shown in Algorithm 2 in the Appendix). This raises the question: Does FlowDAS support generating an ensemble of analysis samples at each assimilation step, and then propagating these samples to produce the next-step forecast /analysis ensemble. Such a capability would be analogous to ensemble filtering methods like the EnKF, where uncertainty is explicitly carried forward through the flow. Clarifying whether FlowDAS supports this kind of flow-dependent uncertainty evolution would help contextualize its probabilistic modeling power.

[1] Descombes, Gael, T. Auligné, Francois Vandenberghe, D. M. Barker, and Jerome Barre. "Generalized background error covariance matrix model (GEN_BE v2. 0)." *Geoscientific Model Development* 8, no. 3 (2015): 669-696.

[2] Bannister, Ross N. "A review of forecast error covariance statistics in atmospheric variational data assimilation. I: Characteristics and measurements of forecast error covariances." *Quarterly Journal of the Royal Meteorological Society: A journal of the atmospheric sciences, applied meteorology and physical oceanography* 134, no. 637 (2008): 1951-1970.

[3] Bannister, Ross N. "A review of forecast error covariance statistics in atmospheric variational data assimilation. II: Modelling the forecast error covariance statistics." *Quarterly Journal of the Royal Meteorological Society: A journal of the atmospheric sciences, applied meteorology and physical oceanography* 134, no. 637 (2008): 1971-1996.

**Ethical Concerns:**

["NO or VERY MINOR ethics concerns only"]

**Final Justification:**

The author didn't compare their method with 3DVar, which is a crucial baseline for numerical weather prediction experiments. Without such comparisons, it becomes challenging to fully assess the effectiveness of the proposed approach. Putting aside the experimental rigor, I do acknowledge that the proposed algorithm is innovative from a methodological perspective, and it will likely be of interest and value to the AI research community. Therefore, I will maintain my evaluation as borderline accept.

**Limitations:**

Yes, the authors have discussed the limitations.

**Quality:**

3

**Strengths And Weaknesses:**

Strength:

1. The paper addresses an important and timely problem in scientific machine learning and data assimilation.
2. The proposed framework is conceptually novel, combining stochastic interpolants with observation-conditioned inference.
3. FlowDAS enables unified probabilistic forecasting and assimilation, and avoids some limitations of score-based diffusion models (e.g., global sampling, lack of interpretability).

Weaknesses:

1. The implementation of baseline methods, particularly the 3D-Var setup, appears suboptimal and underexplored. Stronger or more representative model-driven baselines could better contextualize the performance of FlowDAS.
2. The paper lacks a detailed comparison of computational cost with other baselines.
3. Although the method is designed for probabilistic forecasting, the rolling forecast experiments use only a single sample per step, without evaluating or visualizing uncertainty propagation. This limits the demonstrated benefit of using stochastic interpolants.

---

> ### Author Rebuttal · Authors · 2025-07-28
>
> We thank the reviewer for their balanced and constructive feedback, and we are encouraged by their recognition of the potential of our work.
>
> **A. On the implementation of 3D-Var in high-dimensional settings**
>
> We agree that the background error covariance matrix is a critical component of modern data assimilation systems used in numerical weather prediction (NWP). Like in our SEVIR settings, the B matrix measures how an error of the VIL at one location is related to errors at other locations and is quite critical for modern variational data assimilation (VAR) systems [1]. However, it is always hard and complicated to compute this B matrix, especially in high-dimensional systems like the SEVIR and NS. Taking the SEVIR as an example, we only take VIL into consideration and its size is $1\times 128\times 128$, and its B-matrix would have a shape of $16384 \times 16384=268,435,456$. Even though the B matrix is a positive semi-definite, symmetric matrix, it would still be very hard to compute, and we would need to resort to highly specialized scientific software like GEN_BE v2.0 [2].
>
> Given this, and the fact that FlowDAS is intended as a general-purpose, model-agnostic framework, looking into this specific question setup would be an important but orthogonal direction to our current contributions. We would be happy to discuss this point further and add it in the discussion section if the paper is accepted.
>
> Besides, we have added EnKF as a baseline in our Lorenz 63 experiments. In EnKF, the forecast error covariance $B$ is approximated using the sample covariance of the forecast ensemble, which avoids explicit construction of $B$ and instead operates directly on the ensemble. We observe that FlowDAS still outperforms the EnKF in terms of RMSE. Besides, both BPF and EnKF achieve similar RMSE, which is expected as they both have the knowledge of the forward dynamic model.
>
> |         | FlowDAS | SDA    | PDEDiff | BPF    | EnKF   |
> |---------|---------|--------|---------|--------|--------|
> | **RMSE**| 0.202   | 1.114  | 0.871   | 0.270  | 0.281  |
>
> > Note: As suggested by Reviewer Ztob, we also include **PDEDiff**, the universal amortised extension of SDA proposed in [4], as an additional baseline for completeness. PDEDiff trains a conditional score network to autoregressively predict the next state(s) given a number of past states, enabling more flexible and accurate sequential generation compared to the original all-at-once SDA. This helps address the forecasting limitations of SDA’s joint score formulation.
>
> Finally, as we transition FlowDAS to real-world weather forecasting applications in collaboration with industry, the design and approximation of the background error covariance will definitely be an important consideration for future work.
>
> &nbsp;
>
> **B. On computational cost comparisons:**
>
> We agree with the reviewer on the importance of computational efficiency and have added a detailed performance comparison in the appendix. The results, using a single A100 GPU on the SEVIR dataset, highlight FlowDAS’s practical viability:
> | Model       | Params (M) | Total Training Time (hr) | Inference Time per Sample (s) |
> |:-----------:|:----------:|:-------------------------:|:------------------------------:|
> | FlowDAS     | 30.0       | 22                        | 30.9                           |
> | SDA         | 22.9       | 45                        | 28.9                           |
> | Transolver  | 11.2       | 80                        | 12.7                           |
> | FNO         | 1880.0     | 80                        | 2.4                            |
>
> The slightly longer inference time for FlowDAS stems from its iterative correction process, which is essential for achieving accurate and observation-consistent predictions. While this leads to higher per-sample inference time, it is balanced by significant **training efficiency**, as FlowDAS reaches convergence much faster than all competing baselines. We believe this reflects a favorable trade-off in practical deployment scenarios. A more detailed discussion will be included in the final version.
>
> To further improve inference speed, one promising direction is to apply **post-hoc distillation**, where a faster surrogate model is trained to approximate the FlowDAS sampler. Another avenue is to explore hybrid formulations that **integrate variational inference** techniques into the framework, potentially offering both computational gains and improved flexibility.
>
> &nbsp;
>
> **C. On ensemble sampling and uncertainty modeling in inference:**
>
> For uncertainty modeling, yes, **FlowDAS does support ensemble-style sampling**. At each assimilation step, we can generate multiple samples from the posterior $p( x\_{k+1} | x\_k, y\_{k+1}) $, allowing us to form an ensemble of analysis trajectories. These samples can then be propagated forward through the system dynamics, enabling flow-dependent uncertainty evolution similar to EnKF.
> In our experiments, we used this capability to assess uncertainty. For instance, in the Navier–Stokes (NS) experiments, we conducted inference 25 times independently and used the resulting ensembles to compute standard deviations of RMSE across different settings. We apologize for not making this explicit in the main text. Below we report the mean and standard deviation (std) of RMSE over 25 sampled trajectories (with some additional experiments on more challenging assimilation scenarios suggested by Reviewer 3gVf):
>
> |          RMSE           |   SR 4x   | SR  8x   | SR 16x   |  SO 5%   | SO 1.5625% |  SO 1%   | SO 0.75%  |
> |:-----------------------:|:------:|:------:|:------:|:------:|:-------:|:------:|:------:|
> | Mean                    | 0.038  | 0.067  | 0.108  | 0.071  | 0.123   | 0.148  | 0.177  |
> | Std                     | 0.001  | 0.050  | 0.010  | 0.005  | 0.010   | 0.014  | 0.019  |
>
> We will clarify this ensemble sampling capability and its usage in uncertainty quantification in the revised paper.
>
> &nbsp;
>
> ## Reference
> [4] Shysheya, A., Diaconu, C., Bergamin, F., Perdikaris, P., Hernández-Lobato, J. M., Turner, R., & Mathieu, E. (2024). On conditional diffusion models for PDE simulations. Advances in Neural Information Processing Systems, 37, 23246-23300.

---

> > ### Comment · Reviewer_RCK2 · 2025-08-04
> >
> > Thank you to the authors for conducting the additional experiments, especially the evaluation of computational costs and the clarifications regarding uncertainty quantification. In my view, 3DVar remains a crucial baseline for numerical weather prediction experiments. As a classical data assimilation algorithm that leverages expert knowledge (for example, by modeling partial correlations in the background error covariance such as horizontal correlations), it is still necessary to include and evaluate 3DVar in the experimental comparisons. This helps ensure the validity of the results and clarifies the advantages brought by learning-based methods. Without such comparisons, it becomes challenging to fully assess the effectiveness of the proposed approach.
> > That being said, putting aside the experimental rigor, I do acknowledge that the proposed algorithm is innovative from a methodological perspective, and it will likely be of interest and value to the AI research community. Therefore, I will maintain my evaluation as borderline accept.

---

### Official Review · Reviewer_CZwM · 2025-06-28

**Clarity:** 2
**Significance:** 2
**Originality:** 3
**Rating:** 4
**Confidence:** 4

**Summary:**

The paper presents _FlowDAS_, a technique that leverages dynamics modeled by stochastic interpolants to perform data assimilation. Given observations, the method proceeds by augmenting the original drift vector field (learned by training a stochastic interpolant) with the likelihood score, approximated by Monte Carlo. Experiments on several data assimilation benchmarks demonstrate superior performance of FlowDAS to other ML-based data assimilation methods (such as score-based data assimilation) on several metrics evaluating accuracy of state estimation and uncertainty quantification.

**Questions:**

- What do the acronyms for FlowDAS stand for? I don't believe this is introduced.
- In data assimilation, typically the objective is to compute the filtering distribution $p(x\_{k+1} | y\_1, \ldots, y\_{k+1})$. Instead, one step of FlowDAS targets sampling from the distribution $p(x\_{k+1} | x\_{k-l:k}, y\_{k+1})$. This leads to a few questions:
    - Can you please provide more details on how to roll out FlowDAS to assimilate data sequentially? Assuming Markovian dynamics for simplicity, given data $y_{k+1}$, FlowDAS essentially modifies the distribution $p(x\_{k+1} | x\_{k})$ to get $p(x\_{k+1} | x\_{k}, y\_{k+1})$. Now suppose we get  samples $x\_{k+1}^{(n)} \sim p(x\_{k+1} | x\_{k}, y\_{k+1})$ and a new observation $y_{k+2}$. Then, do we take samples from $p(x\_{k+2} | x\_{k+1}^{(n)}, y\_{k+2})$ for all $n$? Thus effectively sampling from $\int p(x_{k+2} | x_{k+1}, y_{k+2}) p(x\_{k+1} | x\_{k}, y\_{k+1}) dx_{k+1}$?
    - If so, it is still unclear to me how this distribution should be interpreted. If it is meant to be $p(x_{k+2} | x_{k}, y_{k+1}, y_{k+2}) = \int p(x_{k+2} | x_{k+1}, y_{k+2}) p(x\_{k+1} | x\_{k}, y\_{k+1}, y\_{k+2}) dx_{k+1}$, then at the first step, we should be sampling from $p(x\_{k+1} | x\_{k}, y\_{k+1}, y\_{k+2}) $ instead of $p(x\_{k+1} | x\_{k}, y\_{k+1})$. I wonder if this causes any problems.
- I wonder why FlowDAS does not suffer from particle collapse when computing the likelihood score (8), which is a well-known issue with importance sampling in high dimensions (i.e. one of the weights $w\_j$ become much more dominant compared to others, which leads to "collapsing" all the contribution to particle $\hat{X}\_1^{(j)}$).
- I am confused by the use of the Wasserstein distance to evaluate the deviation of $\hat{x}\_{1:K}$ from the ground truth $x\_{1:K}$. Is this measuring the distance between two delta measures, one centered at $\hat{x}\_{1:K}$ and the other at $x\_{1:K}$? If so, what does this mean? It seems more natural to me to consider other uncertainty estimation metrics such as the log-likelihood $\log p(x\_{1:K} | \hat{x}\_{1:K})$ or CRPS.
- I also don't understand the use of the metric $\log p(\hat{x}\_{2:K} | \hat{x}\_1)$. This does not make a comparison with either the ground truth $x_{1:K}$ nor the observations $y$ so in what sense does it evaluate the quality of the predictions?

**Ethical Concerns:**

["NO or VERY MINOR ethics concerns only"]

**Final Justification:**

The proposed methodology provides an elegant, new approach to data assimilation that nicely makes use of the properties of stochastic interpolants and how they can condition on observations. The experiments also show promising results, both quantitatively and qualitatively. However, I still have several concerns that became more apparent during the discussion. First, the method is not theoretically sound, as it does not sample from the filtering distribution. This leads to concerns regarding the robustness and reliability of the method. Second, it became apparent that the uncertainty quantification was not properly evaluated. The new results on CRPS show that FlowDAS actually struggles compared to the baseline models, although further investigation is necessary, as also mentioned by the authors. For these reasons, I am willing to give an acceptance, but I consider it borderline.

**Limitations:**

Limitations are not explicitly addressed in the work as far as I can see.

**Paper Formatting Concerns:**

There are no paper formatting concerns.

**Quality:**

3

**Strengths And Weaknesses:**

__Strengths:__
- The proposed method is simple and elegant; It is almost plug-and-play once the stochastic interpolant is trained, and there is no need for additional training to assimilate observations.
- The results come naturally equipped with uncertainty quantification.
- Experiments are extensive, especially in the supplementary materials, and show clear promise of the methodology.

__Weaknesses:__
- The writing feels rushed in places. There are several misprints peppered throughout the paper. For example:
    - Line 147: "they does not" -> "they do not"
    - Line 168: "an Multi-Layer Perceptron" -> "a Multi-Layer Perceptron".
    - Caption of Table 2: "Naiver-Stokes" -> "Navier-Stokes".
    - The sentence in Lines 69-70 "struggling in discontinuity situations..." feels rushed and can be improved. For example, what do the authors mean by "discontinuity situations".
- To my understanding, FlowDAS does not sample from the filtering distribution $p(x_k | y_1, \ldots, y_k)$, which is usually the target distribution of data assimilation. This raises several questions. See __Questions__ below for more details.
- I do not understand the reasoning behind the choice of some of the UQ metrics. See __Questions__ below.
- Comparisons with classical filtering baselines are largely missing. In particular, the benchmark problems considered are settings where traditional data assimilation methods such as the ensemble Kalman filter (EnKF) or its variants, such as LETKF, can be applied without problems. Hence, without comparisons to these baselines, it is unclear what merit the proposed method brings.
- There are related methodologies proposed in the literature, such as EnSF (see [1]) and DiffDA (cited in work), that also consider "step-by-step" generation similar to FlowDAS, but are not compared. Again, it is unclear what advantages FlowDAS brings over these methodologies.

[1] Bao, Feng, Zezhong Zhang, and Guannan Zhang. "An ensemble score filter for tracking high-dimensional nonlinear dynamical systems." Computer Methods in Applied Mechanics and Engineering 432 (2024): 117447.

---

> ### Author Rebuttal · Authors · 2025-07-29
>
> We thank the reviewer for their balanced and constructive feedback, and we are encouraged by their recognition of the potential of our work.
>
> ## Weaknesses:
> **A. “The writing feels rushed in places”**
>
> We thank the reviewer for the careful reading and for pointing out these writing issues. We corrected all the noted misprints. Specifically, we now clarify that “discontinuity situations” refer to assimilation tasks involving abrupt state changes or non-smooth dynamics that often challenge traditional filtering methods.
>
> **B. “Comparisons with classical filtering baselines are largely missing.”**
>
> We have added EnKF as classical filtering baselines in the Lorenz 63 experiment. The updated RMSE comparison is shown below:
>
> |         | FlowDAS | SDA   | PDEDiff | BPF   | EnSF  | EnKF  |
> |---------|---------|-------|---------|-------|-------|-------|
> | RMSE    | 0.202   | 1.114 | 0.871   | 0.270 | 0.298 | 0.318 |
>
> The highly nonlinear system dynamics induce non-Gaussian ensembles, which breaks the assumption of EnKF. Our nonlinear observation model in this case further deteriorates its performance. In contrast, FlowDAS is data-driven and does not rely on linearity or Gaussianity assumptions. As a result, it achieves the lowest RMSE across all baselines, including classical filters BPF and EnKF, and diffusion-based methods, e.g., SDA and PDEDiff. This supports FlowDAS’s suitability for nonlinear and non-Gaussian data assimilation tasks.
>
>
> **C. “It’s unclear what advantages FlowDAS brings over EnSF and DiffDA, etc.”**
>
> We have added **EnSF** to the Lorenz-63 experiment (details below). Its performance is on par with BPF. This is expected since EnSF uses a training-free Monte Carlo estimator for the score function, which faces the same **particle collapse issues**, i.e., samples concentrate in high-probability regions of the transition dynamics, leading to biased score estimates. This reflects a **train-sampling tradeoff**: EnSF saves training time but cannot learn accurate scores, especially under highly nonlinear dynamics and observation models like in our noisy Lorenz-63 settings(e.g., our $\text{atan}(x\_1)$ observation). FlowDAS, by learning the drift with collected data, avoids this issue and achieves superior performance.
>
> We do not include a direct comparison to **DiffDA** for the following reasons:
>
> 1). **Mismatch in data and task setup**: DiffDA is tailored for **ERA5 reanalysis**, which involves many atmospheric variables. In contrast, SEVIR is a simplified dataset with only vertically integrated liquid fields. Adapting DiffDA to SEVIR is non-trivial and out of scope for this work, but we are happy to include it in future studies.
>
> 2). **Architectural differences**: DiffDA uses an external numerical weather model as its forecasting component and applies a diffusion model purely as a denoiser. FlowDAS handles both forecasting and assimilation jointly with a single learned stochastic model, which is more modular and general-purpose.
>
> We will clarify this distinction in the final version and emphasize that FlowDAS targets **general, model-agnostic data assimilation** without relying on external solvers.
>
> &nbsp;
>
> ## Questions:
> **1. “What do the acronyms for FlowDAS stand for?”**
>
> We apologize for the omission. FlowDAS stands for Flow-based Data Assimilation with Stochastic Interpolants. Here, “Flow” refers to the use of stochastic interpolant flows to model the posterior distribution, and “DAS” is short for Data Assimilation. We will revise the manuscript to introduce and define this acronym explicitly at first mention.
>
> **2. “More details on how to roll out FlowDAS”**
>
> * To improve clarity in inference, we revised the Inference paragraph in the main text to explicitly explain how FlowDAS proceeds in sequential data assimilation (see below).
> * To directly address your questions on the target distribution that FlowDAS samples from: In our current setup, we sample $x\_{k+1} \sim p(x\_{k+1} | x\_k, y\_{k+1})$ with $n=1$, and proceed sequentially. That is, we assimilate one observation at a time and do not marginalize over multiple future samples. FlowDAS targets a local posterior conditioned on the current state and the next observation only. We agree that extending FlowDAS to sample from $p(x\_{k+1} | x\_{k}, y\_{k+1:k+P})$ would provide a more expressive and generalized model and can be better suitable to approximate the full filtering distribution. We mention this as a promising direction for future work.
>
> **Inference** Given a trained drift model $\hat{b}_s$, FlowDAS performs inference in an autoregressive manner to estimate the latent state trajectory $ \{ \hat{x}\_{1:K} \}$. At each time step $k$, we begin from the known state $\hat{x}_k$ and predict the next state $\hat{x}\_{k+1}$ using a discretized stochastic interpolant over a time grid $s_0 = 0 < s_1 < \cdots < s_N = 1$. We set the interpolant state $X\_{s\_0} \gets \hat{x}\_k $ and retrieve the next observation $ y\_{k+1} $. For each interpolation step $ s\_n$, we simulate a forward transition using the learned drift, given in Eq.7, and noise. To enforce observation consistency, we generate $J$ posterior endpoint samples $ \hat{X}\_1^{(j)}$, and compute their likelihoods under the measurement model, i.e., $ || y - \mathcal{A}(\hat{X}\_1^{(j)}) ||_2^2 $. A softmax over these values gives importance weights $ \{ w\_{1:J} \}$. We then apply a correction to the interpolant using a weighted gradient $- \zeta\_n \nabla\_{X\_{s\_n}} \sum \_{j=1}^J w\_j || y - \mathcal{A}(\hat{X}\_1^{(j)}) ||_2^2$, where $\zeta\_n$ denotes the step size. After reaching $s=1$, we set $ \hat{x}\_{k+1} \gets X\_{s\_N} $ and proceed to the next time step. Repeating this procedure over all $ k $ yields the full trajectory.
>
> **3. Particle collapse issue for FlowDAS**
>
> This is a very insightful comment! This is indeed a well-known issue in high-dimensional importance sampling and a valuable observation for our method.
>
> We observed that the likelihood score can first decrease and then unexpectedly increase near the end of the sampling trajectory. When this occurs, we switch to the alternative sampling strategy described in the supplementary material. While it may occasionally produce slightly suboptimal results, it is more robust compared to the importance sampling approach in the main text. We will explicitly mention this issue and our workaround in the revised paper, and plan to investigate more principled solutions in future work.
>
> More specifically, we found that particle collapse tends to occur during the final 30–50 steps of the sampling process. In these cases, the likelihood term $||y - Ax||_2^2$, which is expected to decrease steadily, begins to rise unexpectedly—eventually diverging and causing sampling failures. Upon examining the importance weights, we observed a clear pattern:
> * When the likelihood is decreasing as expected, the weights are distributed relatively evenly across particles.
> * When the likelihood increases abnormally, the weights become highly concentrated on a single particle (close to 100%), indicating a collapse of the Monte Carlo estimator.
>
> To address this, we use the following two-step strategy:
> * **Resampling**: When the weights are abnormally concentrated, we immediately resample a new set of particles. This is a standard remedy for particle collapse in importance sampling.
> * **Alternative sampling strategy**: If resampling does not resolve the issue, we switch to the alternative sampling method described in the supplementary. Although it is not always as precise, it significantly improves robustness and prevents divergence.
>
> **4 & 5. Metrics used when evaluating FlowDAS**
>
> We clarify that these metrics were only used in the Lorenz experiments, where our goal was to benchmark FlowDAS against SDA. Since SDA used these evaluation metrics, we adopted the same for direct comparability. Below, we explain the meaning of each metric and we will clarify them in the revised text:
> * Expected log-prior $\mathbb E\_{q(x\_{1:K} | y)} [\log p(x\_{2:K} | x\_1)] $: This measures whether the generated trajectories conform to the known prior dynamics. Specifically, given a sampled trajectory $x\_{1:K} \sim q$, we evaluate how likely it is under the true transition model $p(x\_{k+1} \mid x\_k)$ for $k = 1, \ldots, K-1$. A higher value indicates better physical consistency with the governing equations. It penalizes trajectories that deviate from the expected evolution under the dynamics, even if they match observations.
> * Expected log-likelihood $\mathbb E\_{q(x\_{1:K} | y)} [\log p(y | x\_{1:K})] $: This evaluates how well the generated trajectories explain the observed data. For each sampled trajectory, we compute the likelihood of the full observation sequence $y$, and average this across samples. This score reflects data fidelity, i.e., higher values indicate that the sampled trajectories are more consistent with the observations under the assumed noisy and nonlinear observation model.
> * Wasserstein distance $W\_1(p, q)$: This is computed between the estimated and ground-truth trajectory distributions. In our case, it becomes a trajectory-wise distance between delta measures centered at $\hat{x}\_{1:K}$ and $x\_{1:K}$. While simple, it provides an interpretable measure of deviation in trajectory space.
>
> &nbsp;
>
> ## Appendix
> **Implementation of EnSF**
> | Setting                          | Value         |
> |----------------------------------|---------------|
> | Ensemble size                    | 20,000|
> | Diffusion SDE steps              | 500           |
> | SDE solver                       | Euler  |
> | $\varepsilon_a$, $\varepsilon_b$ [1] | 0.001, 0.001   |
> | Initial data variance            | 0             |
> | Inflation factor                 | 1             |
>
> [1] Bao, Feng, Zezhong Zhang, and Guannan Zhang. "An ensemble score filter for tracking high-dimensional nonlinear dynamical systems." Computer Methods in Applied Mechanics and Engineering 432 (2024): 117447.

---

> > ### Comment · Reviewer_CZwM · 2025-08-04
> >
> > I thank the authors for clarifications regarding questions I had. However, I do have some follow-up questions/comments that I would like to add:
> >
> > - To clarify the rollout procedure, we are effectively sampling $x\_0 \sim p(x\_0)$, $x\_1 \sim p(x\_1 | x\_0, y\_1)$, $x\_2 \sim p(x\_2 | x\_1, y\_2)$, etc... which would indicate that we are sampling from the distribution $(x\_0, x\_1, ..., x\_N) \sim p(x\_0) p(x\_1 | x\_0, y\_1) p(x\_2 | x\_1, y\_2) ... p(x\_{N} | x\_{N-1}, y\_{N}) = p(x\_0, \ldots, x\_N | y\_1, \ldots, y\_N)$, which is the *smoothing distribution* and not the *filtering distribution*, which I had initially thought we were targeting, hence my initial confusion. This is not necessarily a bad thing, however I believe this should be made clear in the paper to make sense of what data assimilation problem is being solved (filtering or smoothing?).
> > - Regarding the likelihood score approximation, instead of the Monte-Carlo method that is considered here, can we also consider a DPS-type ([Chung et al., 2022]) approximation $\nabla \log p(y | X\_0, X\_s) \approx \nabla \log p(y | \hat{X}\_1)$, where $\hat{X}\_1 := \mathbb{E}[X\_1 | X\_0, X\_s]$? While this only gives a rough approximation to the true likelihood score, it will not suffer from the particle collapse issue as seen by the authors, and is a more commonly used approach, generally speaking.
> > - Thank you for the clarifications about the UQ metrics used. My confusion came from how these were displayed in Table 1. To avoid confusion, I would recommend explicitly writing $\mathbb{E}\_{q(x\_{1:K} | y)}[\log p(x\_{2:K} | x\_1)]$, $\mathbb{E}\_{q(x\_{1:K} | y)}[\log p(y| x\_{1:K})]$ instead of $\log p(\hat{x}\_{2:K} | \hat{x}\_1)]$, $\log p(y| \hat{x}\_{1:K})$. In addition, I would add a sentence to describe that by $W\_1(\hat{x}\_{1:K}, x\_{1:K})$, you actually mean  $W\_1(\sum\_j \delta\_{\hat{x}\_{1:K}^{(j)}}, \delta\_{x\_{1:K}})$. However, I am still not entirely convinced that the Wasserstein metric is really appropriate here.
> >
> > [Chung et al., 2022] Chung, H., Kim, J., Mccann, M. T., Klasky, M. L., & Ye, J. C. "Diffusion posterior sampling for general noisy inverse problems." arXiv:2209.14687 (2022)

---

> > > ### Author Response · Authors · 2025-08-06
> > >
> > > We thank the reviewer for the thoughtful comments, these have significantly contributed to improving the quality and clarity of our work.
> > >
> > > A. **Clarify what data assimilation problem is being solved (filtering or smoothing?)**
> > > To summarize, in our framework:
> > > - For each step in the autoregressive sampling, the target is to sample from the filtering distribution $p(x\_{k+1} | x\_{k}, y\_{k+1})$ (filtering) and we will clarify this in the paper..
> > >
> > > B. **Likelihood approximation**
> > > The approximation to likelihood term presented in our paper is theoretically **unbiased**. In contrast, the direct DPS-type approximation of the term $\nabla \log p(y | X\_0, X\_s) \approx \nabla \log p(y|\hat{X}_1)$ is theoretically a **biased** estimate (bounded by Jensen’s gap). Assume we just replace the approximation of $\nabla \log p(y | X\_0, X\_s)$ in Eq. 8 with DPS and everything else remains untouched, we show the results on weather forecasting experiment in the table below:
> > >
> > > |         | FlowDAS | FlowDAS-DPS   |
> > > |---------|---------|-------|
> > > | RMSE  ± Std   | 0.056 ± 0.002   | 0.050  ± 0.002 | .
> > >
> > > This DPS estimate can indeed avoid the collapse issue and slightly improve quality. However, the denominator in *Proposition 1* (in the short derivation of FlowDAS-DPS below) with our current parameter settings will approach 0 when $s \rightarrow 0$ and cause the optimization gradient to explode. To avoid this and make the FlowDAS-DPS work, we have to empirically decide a specific step $s\_\epsilon$ and only do the DPS step when $s \geq s_\varepsilon$. This value can vary in different problem settings, and would affect the optimization process, and there is no theoretical explanation for choosing it (We select $s_\varepsilon = 0.04$ by grid search on weather forecasting experiment). So, we are not sure if this can be generalized. One may tune the parameter settings ($ \alpha_s, \beta_s, \sigma_s $) to avoid this issue, which we leave for future work.
> > > This FlowDAS-DPS adds variety to our current framework and further approves the flexibility of our framework, providing a potential venue to further improve the method. We will explore this in our future work.
> > >
> > > C. **UQ metrics**
> > > We appreciate your scrupulous suggestions on the writings of metrics and we will definitely use the appropriate notation as suggested in our final revision.
> > > For the $W\_1$ metric, we apologize that our initial clarification is unclear and leads to confusion. As you noted, we are computing Wasserstein-1 distance between empirical distributions $ \frac{1}{K} \sum_j \delta_{\hat{x}\_j} $ and $ \frac{1}{K} \sum_j \delta_{x\_j} $. This treats the predicted and GT trajectories as **unordered** sets of states (i.e., comparing two point clouds, with no temporal ordering assumed), thus it evaluates how well the model captures the overall geometric structure of the trajectory distribution. We will add these clarifications of each metric in our revised paper.
> > >
> > >
> > >
> > >
> > > &nbsp;
> > >
> > > ### The short derivation of FlowDAS-DPS:
> > >
> > > To derive the DPS-type approximation $\nabla \log p(y|X_0, X_s) \approx \nabla \log p(y|\hat{X}_1), \text{ where } \hat{X}_1 := \mathbb{E}[X_1|X_0, X_s]$, the core idea is to compute $\hat{X}_1 := \mathbb{E}[X_1|X_0, X_s]$
> > >
> > > **Proposition 1.** $\mathbb{E}[X_1|X_0, X_s] = \frac{\sigma_s b_s(X_s, X_0) - \dot{\sigma}_s X_s - (\sigma_s\dot{\alpha}_s - \alpha_s\dot{\sigma}_s)X_0}{(\sigma_s\dot{\beta}_s - \beta_s\dot{\sigma}_s)}$
> > >
> > > It is shown in [2] that:
> > >
> > > $Eq1: b_s(X_s, X_0) = \dot{\alpha}_s X_0 + \dot{\beta}_s \mathbb{E}[X_1|X_0, X_s] + \sqrt{s}\dot{\sigma}_s \mathbb{E}[z|X_0, X_s]  $
> > >
> > > and
> > >
> > > $Eq2: X_s = \alpha_s X_0 + \beta_s \mathbb{E}[X_1|X_0, X_s] + \sqrt{s}\sigma_s \mathbb{E}[z|X_0, X_s]  $
> > >
> > > By computing the expression $\sigma_s \cdot Eq1 - \dot{\sigma}_s \cdot Eq2$ and rearranging the terms, we can derive Proposition 1.
> > >
> > > [2] Chen, Y., Goldstein, M., Hua, M., Albergo, M. S., Boffi, N. M., & Vanden-Eijnden, E. (2024). Probabilistic Forecasting with Stochastic Interpolants and F"ollmer Processes. arXiv preprint arXiv:2403.13724.

---

> > > > ### Comment · Reviewer_CZwM · 2025-08-07
> > > >
> > > > Thank you for the further clarifications and for trying experiments with a DPS-type likelihood approximation. I am happy with the results shown. However, I am still not sure about the other two points.
> > > >
> > > > A. The authors should be aware that $p(x\_{k+1} | x\_k, y\_{k+1})$ is **not the filtering distribution**. The filtering distribution is defined as $p(x\_{k+1} | y\_1, \ldots, y\_{k+1})$, which is where my main confusion comes from. My previous interpretation that it samples from the smoothing distribution $p(x\_1, \ldots, x\_{k+1} | y\_1, \ldots, y\_{k+1})$ is also incorrect, after rethinking it (I did not factor in dependence on future observations). Hence, I am unsure whether inferring with this model in the described manner makes rigorous probabilistic sense.
> > > >
> > > > C. I am confused by what the authors mean by $\frac{1}{K} \sum_j \delta\_{x_j}$. Why are there multiple "ground truths" $x\_j$s? Shouldn't there just be a single trajectory that we refer to as the "ground truth"?
> > > >
> > > > My overall evaluation is that I believe the experiments show promising results; however, I am not entirely convinced by the theoretical soundness of the framework due to the reasoning in A, which may result in some edge cases where it may break. For this reason, I am maintaining my current score for now.

---

> > > > > ### Author Response · Authors · 2025-08-07
> > > > >
> > > > > A. Thanks for your correct and precise comments on the theoretical soundness and we recognize that we did not fully interpret the filtering distribution. However, we still think we could sample from the filtering distribution $p(x_{k+1}|y_1, \dots, y_{k+1})$ **theoretically**, and this would be solved from two elements question:
> > > > >
> > > > > * Sampling from $p(x_{k+1}|x_k, y_{k+1})$, we indicate that our method is capable of sampling $x_{k+1}^{(n)}$ with $n \geq 1$ and providing uncertainty quantification as shown in our experiments results
> > > > > * Sampling $x_0$ from $p(x_0)$, this could be achieved by sampling from the initial guess towards the true initial distribution in a way similar to [3] or from the true initial distribution
> > > > >
> > > > > With the above problems are solved, the results of our sampling would become:
> > > > > $$
> > > > > \int p(x_0)p(x_1|x_0, y_1)p(x_2|x_1, y_2) \dots p(x_k|x_{k-1}, y_k)p(x_{k+1}|x_k, y_{k+1})dx_0 \dots dx_k = p(x_{k+1}|y_1, \dots, y_{k+1})
> > > > > $$
> > > > > which is the **filtering distribution**. The implementation would be saving multiple possible results $x_{k}$ and for each possible $x_{k}$, we perform our methods to sample multiple $x_{k+1}$ (like the particle filter).
> > > > >
> > > > > We will clarify and highlight this in our paper to make our paper theoretical soundness.
> > > > >
> > > > >
> > > > > &nbsp;
> > > > >
> > > > > C. We recognize that our previous explanation caused confusion and appreciate the opportunity to clarify.
> > > > >
> > > > > We evaluate the Wasserstein-1 distance between the predicted and ground-truth trajectories by comparing the sets of states as **unordered point clouds** in $\mathbb{R}^d$. Let $x_{1:K}$ denote the single ground-truth trajectory, and let $\hat{x}_{1:K}^{(i)}$ be the $i$-th predicted trajectory for $i = 1, \dots, N$. For each pair, we compute the Wasserstein-1 distance between the two point clouds:
> > > > >
> > > > > $W\_1\left( x_{1:K}, \hat{x}_{1:K}^{(i)} \right)$,
> > > > >
> > > > > and report the average across all $N$ predicted trajectories:
> > > > >
> > > > > $\frac{1}{N} \sum_{i=1}^N W\_1\left( x\_{1:K}, \hat{x}_{1:K}^{(i)} \right)$.
> > > > >
> > > > > The $W\_1(\cdot, \cdot)$ itself captures the minimal cost of transporting one of the point clouds to match the other. By applying it to our case, it measures how well the predicted states match the overall geometric structure of the ground-truth trajectory, without assuming any temporal alignment. We will revise the explanation in our final version to remove ambiguous references to “distribution” and clarify this as a **point-cloud matching metric**.
> > > > >
> > > > > [3] Bao, Feng, Zezhong Zhang, and Guannan Zhang. "An ensemble score filter for tracking high-dimensional nonlinear dynamical systems." Computer Methods in Applied Mechanics and Engineering 432 (2024): 117447.

---

> > > > > > ### Comment · Reviewer_CZwM · 2025-08-07
> > > > > >
> > > > > > Thank you again for the responses. However, I think the responses are still flawed and do not address my concerns fully.
> > > > > >
> > > > > > A. I would like to point out that the calculation $$ \int p(x_0)p(x_1|x_0, y_1)p(x_2|x_1, y_2) \dots p(x_k|x_{k-1}, y_k)p(x_{k+1}|x_k, y_{k+1})dx_0 \dots dx_k = p(x_{k+1}|y_1, \dots, y_{k+1}) $$ is **incorrect** for the same reason my calculation for the smoothing distribution earlier was incorrect. This is because, we need to take into account *future observations* to multiply the conditional probabilities meaningfully to form joint distributions. To illustrate this, consider the first two terms $p(x_0)$ and $p(x_1|x_0, y_1)$. Then,
> > > > > > $$p(x\_0)p(x\_1|x\_0, y\_1) \neq p(x\_0, x\_1 | y\_1) = p(x\_0 | y\_1)p(x\_1|x\_0, y\_1)$$
> > > > > > in general unless $x\_0$ and $y\_1$ are independent, which is *not the case* for state-space models. Thus, to compute the filtering distribution $p(x\_1 | y\_1) = \int p(x\_0, x\_1 | y\_1) dx\_0$ at the first time step from $p(x\_1|x\_0, y\_1)$, one requires samples from $p(x\_0 | y\_1)$ and **not** $p(x\_0)$. Thus, I am not fully satisfied that the proposed method is actually solving the filtering problem.
> > > > > >
> > > > > > B. Regarding the Wasserstein metric
> > > > > > $$ \frac{1}{N} \sum\_{i=1}^N W\_1(x\_{1:K}, \hat{x}\_{1:K}^{(i)}), $$
> > > > > > doesn't this just boil down to *mean absolute error*
> > > > > > $$ \frac{1}{N} \sum\_{i=1}^N W\_1(x\_{1:K}, \hat{x}\_{1:K}^{(i)}) = \frac{1}{N} \sum_{i=1}^N |x\_{1:K} - \hat{x}\_{1:K}^{(i)}|?$$
> > > > > > Hence, I am not convinced that this is a meaningful UQ metric. I would instead consider a metric like CRPS:
> > > > > > $$ CRPS = \frac{1}{N} \sum\_{i=1}^N |x\_{1:K} - \hat{x}\_{1:K}^{(i)}| - \frac{1}{2N^2}\sum\_{i, j=1} |x\_i - x\_j|$$,
> > > > > > which is similar to the proposed Wasserstein metric, but is a proper scoring rule.

---

> > > > > > > ### Author Response · Authors · 2025-08-08
> > > > > > >
> > > > > > > We sincerely thank the reviewer for the constructive feedback that helped us clarify the theoretical derivation.
> > > > > > >
> > > > > > > &nbsp;
> > > > > > >
> > > > > > > ## A. Theory
> > > > > > > We acknowledge that our initial formulation of the filtering distribution was incorrect. However, **under the additional assumption**:
> > > > > > >
> > > > > > > $p(x\_0) = \delta (x\_0 - x\_0^\star)$,
> > > > > > >
> > > > > > > i.e., we assume a **known** initial state $x\_0=x\_0^\star$, then the FlowDAS can be **theoretically grounded with the filtering target**. In fact, we already use this assumption in our experiment setup (this is also assumed in our baselines, e.g., PDEDiff, to make the comparison fair).
> > > > > > >
> > > > > > > &nbsp;
> > > > > > >
> > > > > > > ### (a) When $k=1$
> > > > > > > In this case:
> > > > > > > * $p(y\_1)=\int p(y\_1 | x\_0)p(x\_0)dx\_0 = p(y\_1 | x\_0^\star)$
> > > > > > > * the first-step integral becomes: $p(x\_1 | y\_1) = \int p(x\_1 | x\_0, y\_1) p(x\_0 | y\_1) dx\_0=\int p(x\_1 | x\_0, y\_1) \frac{p(y\_1 | x\_0)p(x\_0)}{p(y\_1)} dx\_0=p(x\_1 | x\_0^\star, y\_1) \frac{p(y\_1 | x\_0^\star)}{p(y\_1)}  = p(x\_1 | x\_0^\star, y\_1)$.
> > > > > > >
> > > > > > > We will use this as the base case for our recursive filtering (shown below).
> > > > > > >
> > > > > > > &nbsp;
> > > > > > >
> > > > > > > ### (b) When $k\geq 1$
> > > > > > > The filtering target becomes
> > > > > > >
> > > > > > > $p(x\_k | y\_{1:k}) = \frac{ p(x\_k | y\_{1:k-1}) p(y\_k | x\_k)}{p(y\_k|y\_{1:k-1})} \propto p(x\_k | y\_{1:k-1}) p(y\_k | x\_k)=\int p(x\_k, x\_{k-1}| y\_{1:k-1})p(y\_k | x\_k)dx\_{k-1}$
> > > > > > >
> > > > > > > Applying Bayes' rule, we extend that:
> > > > > > >
> > > > > > > $\int p(x\_k, x\_{k-1}| y\_{1:k-1})p(y\_k | x\_k)dx\_{k-1} = \int p(x\_{k-1} | y\_{1:k-1}) p(x\_k | x\_{k-1}, y\_{1:k-1})  p(y\_k | x\_k) d x\_{k-1} = \int  p(x\_{k-1} | y\_{1:k-1}) p(x\_k | x\_{k-1}) p(y\_k | x\_k) d x\_{k-1}$
> > > > > > >
> > > > > > >
> > > > > > > where the last equation is because of the Markov property.
> > > > > > >
> > > > > > > Now, let's look at the three terms in the integrand:
> > > > > > >
> > > > > > > - $p(x\_{k-1} | y\_{1:k-1})$: this is the filtering target of time step $k-1$, let's leave it untouched for recursion.
> > > > > > > - $p(x\_k | x\_{k-1})p(y\_k | x\_k)$: this term can be modeled by FlowDAS (i.e., $p(x_k\mid x_{k-1},y_k)$) because at a fixed $x\_{k-1}$ (it's within the integrand), we have: $p(x_k\mid x_{k-1},y_k)
> > > > > > > =\frac{p(x_k\mid x_{k-1})p(y_k\mid x_k, x_{k-1})}{p(y_k \mid x_{k-1})} = \frac{p(x_k\mid x_{k-1})p(y_k\mid x_k)}{p(y_k \mid x_{k-1})} \propto p(x_k\mid x_{k-1})p(y_k\mid x_k)$
> > > > > > >
> > > > > > > &nbsp;
> > > > > > >
> > > > > > > ### (c) Combing all together
> > > > > > > We have the following recursive filtering target:
> > > > > > >
> > > > > > > $\boxed{p(x_k\mid y_{1:k})
> > > > > > > \propto \int p(x_k\mid x_{k-1},y_k)p(x_{k-1}\mid y_{1:k-1})dx_{k-1}}$,
> > > > > > >
> > > > > > > where
> > > > > > > - $p(x_k\mid x_{k-1},y_k)\approx p\_{\theta}(x_k\mid x_{k-1},y_k) $ is modeled by FlowDAS
> > > > > > > - the base case is $p(x\_1 | y\_1) = p(x\_1 | x\_0^*, y\_1)$, which is also modeled by FlowDAS.
> > > > > > >
> > > > > > > &nbsp;
> > > > > > >
> > > > > > > ### (d) Summary
> > > > > > > FlowDAS learns the conditional density $p_\theta(x_k\mid x_{k-1},y_k)$. Plugging this learned density into the boxed identity above yields the filtering distribution $p(x_k\mid y_{1:k})$. In the linear–Gaussian case, $p(x_k\mid x_{k-1},y_k)$ has the Kalman closed form; substituting it reproduces the Kalman filter. Outside that case, FlowDAS provides a learned non-Gaussian, nonlinear counterpart within the same recursion.
> > > > > > >
> > > > > > > &nbsp;
> > > > > > >
> > > > > > > ## B. Wasserstein metric
> > > > > > >
> > > > > > > We thank the reviewer for the suggestion and will **include CRPS in the main paper as the primary proper scoring metric** for probabilistic evaluation. We keep $W_1$ as a complementary metric because it measures a different property: unlike RMSE or MAE, which compare states at the same time index, $W_1$ finds the optimal matching between the predicted and ground-truth state sets. This makes it insensitive to phase drift, which is inevitable in chaotic systems like Lorenz--63, and allows it to capture geometric fidelity of the attractor even when temporal alignment is lost. We note that $W_1$ has also been used for evaluating Lorenz--63 trajectories in prior work[3], further supporting its relevance here.
> > > > > > >
> > > > > > > [3] Rozet, François, and Gilles Louppe. "Score-based data assimilation." Advances in Neural Information Processing Systems 36 (2023): 40521-40541.

---

> > > > > > > > ### Comment · Reviewer_CZwM · 2025-08-08
> > > > > > > >
> > > > > > > > Thank you again for the further clarifications and the detailed derivations. However, I must say the calculations are **still incorrect**.
> > > > > > > >
> > > > > > > > A. The new calculation that the authors presented is still incorrect because they haven't factored in that the denominator when using Bayes' rule is dependent on $x\_{t-1}$. That is,
> > > > > > > >
> > > > > > > > $\int p(x\_{k-1} | y\_{1:k-1}) p(x\_k | x\_{k-1}) p(y\_k | x\_k) d x\_{k-1} = \int p(x\_{k-1} | y\_{1:k-1}) p(x\_k | x\_{k-1}, y\_k) {\color{red} p(y\_k | x\_{k-1})} d x\_{k-1}$.
> > > > > > > >
> > > > > > > > Notice that this term in red, which appears as a "normalizing constant" in the Bayes' rule cannot be ignored due to the dependence on $x\_{k-1}$.
> > > > > > > >
> > > > > > > > B. I am happy for the authors to include the CRPS score; however, ideally, I would have liked to see how FlowDAS performs under this metric compared to the other baselines. Since CRPS also penalizes the method if the ensembles are "too close to each other" (i.e. one needs to have good diversity in addition to a good MAE to have a good CRPS score) we should expect different behaviour as the Wasserstein metric, which only assesses MAE. So even if the method performs well under Wasserstein, there is no reason it should perform well on CRPS.

---

> > > > > > > > > ### Author Response · Authors · 2025-08-09
> > > > > > > > >
> > > > > > > > > We appreciate it for your patience and we recognize that FlowDAS is initially designed to sample the entire trajectory $x\_{1:K}$ by autoregressively modeling the local filtering distribution $p\_{\theta}(x\_k | x\_{k-1}, y\_k)$.
> > > > > > > > >
> > > > > > > > > &nbsp;
> > > > > > > > >
> > > > > > > > > ## A. Theory
> > > > > > > > > We agree that when using Bayes’ rule in the form
> > > > > > > > > $p(x_k \mid x_{k-1}, y_k) = \frac{p(y_k\mid x_k)\,p(x_k\mid x_{k-1})}{\int p(y_k\mid x_k’)\,p(x_k’\mid x_{k-1})\,dx_k’}$,
> > > > > > > > > the denominator $Z(x_{k-1})$ depends on $x_{k-1}$ and must remain inside the $\mathrm{d}x_{k-1}$ integration in the filtering recursion.
> > > > > > > > >
> > > > > > > > > In the current FlowDAS implementation, the learned $p_\theta(x_k\mid x_{k-1}, y_k)$ is not guaranteed to be a normalized conditional density, it is parameterized via the stochastic interpolants and used for sampling. As such, the recursion we presented does not explicitly account for the $Z(x_{k-1})$ term, and we acknowledge this as a theoretical limitation for exact Bayesian filtering.
> > > > > > > > >
> > > > > > > > > In practice, we approximate the filtering process via repeated conditional sampling without explicitly computing the normalization. For exact filtering, $Z(x_{k-1})$ could be estimated via Monte Carlo, using samples from $p(x_k \mid x_{k-1})$ (obtained by turning off the observation-guidance term in FlowDAS) and evaluating $p(y_k \mid x_k)$ from the observation model.
> > > > > > > > >
> > > > > > > > > As a promising future direction, the ODE formulation of stochastic interpolants (probability flow) can provide access to tractable log-likelihoods, enabling normalized conditional densities $p_\theta(x_k \mid x_{k-1}, y_k)$ and removing the need to separately estimate $Z(x_{k-1})$. This would make the recursion mathematically exact under our state–space model assumptions.
> > > > > > > > >
> > > > > > > > > &nbsp;
> > > > > > > > >
> > > > > > > > > ## B. CRPS
> > > > > > > > > We have included initial results for FlowDAS, BPF, PDEDiff on the lorenz--63 problem. Due to the fast approaching deadline of this rebuttal period, we defer more comprehensive results to the final version if this paper is accepted. We will report the CRPS on both the probabilistic forecasting case (without updates on observations) and the data assimilation case for all our experiments to indicate the distributional accuracy of FlowDAS.
> > > > > > > > >
> > > > > > > > > &nbsp;
> > > > > > > > >
> > > > > > > > > ### Performance Comparison on Lorenz–63
> > > > > > > > >
> > > > > > > > > | Method   | CRPS   | RMSE   |
> > > > > > > > > |----------|--------|--------|
> > > > > > > > > | FlowDAS  | 0.041  | 0.202  |
> > > > > > > > > | PDEDiff  | 0.034  | 0.871  |
> > > > > > > > > | BPF      | 0.031  | 0.270  |
> > > > > > > > >
> > > > > > > > > Although FlowDAS has a slightly higher CRPS than PDEDiff and BPF, indicating a modest reduction in overall probabilistic score (which combines calibration and sharpness), it achieves a substantially lower RMSE for the ensemble mean. This suggests that FlowDAS produces more accurate point forecasts of the state trajectory, even if its predictive distributions are not quite as well aligned as those of the baselines.
> > > > > > > > >
> > > > > > > > > Importantly, **this balance between RMSE and CRPS can be adjusted** through the sampling step size parameter $\xi$ (Table S.6), which controls the strength of guidance toward the observations during sampling. Typically, according to our empirical observation during experiments, smaller $\xi$ values tend to improve CRPS by producing more calibrated distributions, while larger $\xi$ values favor lower RMSE by pushing the estimates closer to the observations.

---

### Official Review · Reviewer_Ztob · 2025-07-01

**Clarity:** 3
**Significance:** 3
**Originality:** 2
**Rating:** 4
**Confidence:** 3

**Summary:**

This paper proposes FlowDAS, a method to perform data assimilation using the stochastic interpolants generative modelling framework. More specifically, the model learns the state transition dynamics and, at generation time, incorporates a “correction” term that guides the generated states towards the observations, and is approximated through a Monte Carlo sampling procedure. Experiments are performed on a range of dynamical systems ranging from Lorenz-63 to Navier-Stokes, and a real-life precipitation dataset, where FlowDAS shows promising performance.

**Questions:**

1. What do the authors mean that FlowDAS “provides clearer physical insight into one-step dynamics” / “training … provides clearer physical insight into one-step dynamics”? Concretely, what becomes clearer in the stochastic interpolant framework versus, for example, diffusion models?
2. Could the authors clarify how expensive the method is at sampling time? Could they also compare the training costs of different architectures?
3. For the SDA baseline, the authors of the paper mention that employing one or a few steps of Langevin sampling (after the predictor step) are very important for good performance (section 3.3: " In practice, we find that few LMC steps are necessary."). Is this something the authors have also employed in the experiments?
4. Could the authors comment more on the limitations of their approach?
5. Could the authors include a discussion of the related work of [4]?

**Ethical Concerns:**

["NO or VERY MINOR ethics concerns only"]

**Final Justification:**

After the rebuttal, I decided to maintain my rating of 4 (Borderline Accept). Several concerns were addressed, including clarifications on baseline implementations, additional details on computational cost, ablations for the conditioning scenario, and a commitment to expand the discussion of limitations. However, I did not increase my score because some contributions remain incremental relative to the baselines. Furthermore, the proposed framework appears to require substantial computational resources, both in terms of memory and inference time, due to the large number of steps needed to preserve sample diversity. While the authors suggest that operating in latent space could mitigate these costs, learning a sufficiently compressed latent representation that still retains the necessary information is itself a non-trivial challenge.

**Limitations:**

It is unclear how costly the framework is at sampling time. I would imagine it should be less costly than diffusion-based formulations, but a clarification would be helpful.

Due to the inclusion of several previous time steps (10 previous states in the Navier-Stokes experiment), the method might have large memory requirements. These might not be prohibitive in the settings considered in the paper, but might become too restrictive to be used in, for example, medium-range weather modelling.

The conditioning approach might also become costly if the likelihood $p(\textbf{y} | \mathbf{X}_1^{(j)})$ is costly to evaluate (not the case in here).

**Paper Formatting Concerns:**

No paper formatting concerns.

**Quality:**

3

**Strengths And Weaknesses:**

**Strengths**

1. **Extensive and varied experiments** -  The experimental section is comprehensive, covering a wide range of dynamical systems, with increasing complexity. The paper also provides additional experiments in the appendix.
2. **Clear, detailed Methodology** - The authors are comprehensive and clear in their description of the methodology.
3. **Thorough Experimental Design Documentation** - Both the main text and appendix specify all implementation details (network architectures, conditioning scenarios, data-generation procedures, etc.), ensuring a good level of transparency.
4. **Ability to handle non-linear observation operators + non-Gaussian noise** - This is something that is tricky to achieve for several posterior estimation techniques in the diffusion model literature, whereas, as far as I understand, this should not be a problem in FlowDAS.

**Weaknesses**
1. **Inappropriate SDA baseline configuration for tasks involving forecasting**
   - **SDA’s original scope**: The chosen baseline (Score‐based Data Assimilation, SDA) was designed for all-at-once conditional generation given observations throughout a trajectory. It is not optimised for forecasting, where future observations are unavailable.
   - **Follow-up adaptation to forecasting**: Indeed, this observation lies at the basis of follow-up work by Shysheya et al. [1], where the authors show that SDA's joint score formulation fails when the task under consideration has a forecasting component, and proposes three autoregressive strategies to deal with this:
     - **Guidance conditioning** - condition on previously-generated states through the same mechanism as the one used for conditioning upon observations (guidance), which does not require any retraining
      - **Amortised approach** - a new **conditional** score network is trained to predict the next state(s) conditioned on a number of previous states
      - **Universal amortised approach** - similar to b), but the authors vary the conditioning scenario during training to allow for more flexibility at sampling time.
   - **Unclear SDA implementation**: The manuscript does not specify which strategy it uses to incorporate previous states into SDA (and I am assuming it is similar to one of the choices presented in Shysheya et al. [1]]):
     - If via guidance, then the literature of diffusion models clearly shows that conditioning through the architecture is much more effective than through guidance, so this baseline would not be entirely fair.
     - If via amortisation, that’s a better setup, but in the light of [1] a stronger baseline would be represented by the universal amortised approach. This is because as proven in [1], for the plain amortised approach longer history (e.g., in the NS experiment a history of 10 states) tends to hurt performance. This was also observed in [2].
   - **Recommendation**: A more appropriate baseline for experiments including a forecasting component (e.g. Lorenz-63, weather forecasting on SEVIR) would be, in my opinion, the universal amortised version of [1], not the original all-at-once SDA formulation.
    - **Joint/Conditional score** - Related to this, the authors claim throughout the manuscript that SDA models the joint score. This is true in the original formulation, but in their experimental setup, either by feeding in the previous states as conditioning information, or by conditioning through guidance, the SDA score also becomes a conditional one.
2. **Lack of computational cost analysis**
   - For sampling, I could not find the value used for $N$, the length of the discretised interpolation interval.
   - The network architectures for the baselines seem to be different, so it would help to understand how that affect the computational cost during training.
3. **Ablation on conditioning horizon** - The authors claim that they observed “markedly improved performance on weather-forecasting tasks” by conditioning on multiple previous states. This is an interesting insight, also discussed to some extent in [1], and performed in SOTA weather forecasting models such as GenCast [3], where they condition on the previous two states. However, the authors do not compare to conditioning on fewer steps / one step to clearly show that the performance is "markedly improved".
4. **Missing limitations discussion** - The authors mention in the paper checklist that the limitations are discussed in the appendix, but I could not find them clearly stated.
5. **Incomplete coverage of a few related methods**
   - Given that some tasks have a strong forecasting component, I would have expected better coverage of the related paper [4]. The experimental setup from NS is taken from there, and the two frameworks are clearly very related (with FlowDAS also incorporating the DA component).
   - The claim that “diffusion models’ non-autoregressive nature makes it hard to deal with long sequence rolling-out problems in high-dimension complex dynamics” overlooks successful high-dimensional autoregressive diffusion methods (e.g. GenCast [3]) and should be revised.
6. **Typographical and notational errors**
   - Eq S.20 - Should denote the conditional score $\nabla \text{log} p (\mathbf{X}_s|\mathbf{X}_0)$, not the joint score $\nabla \text{log} p (\mathbf{X}_s, \mathbf{X}_0)$;
   - Abstract: Phrase “A good modelling”;
   - L40: Missing verb - “Existing DA methods **are** split into two categories.”

**References**

[1] Shysheya, A., Diaconu, C., Bergamin, F., Perdikaris, P., Hernández-Lobato, J. M., Turner, R., & Mathieu, E. (2024). On conditional diffusion models for PDE simulations. Advances in Neural Information Processing Systems, 37, 23246-23300.

[2] Lippe, P., Veeling, B., Perdikaris, P., Turner, R., & Brandstetter, J. (2023). Pde-refiner: Achieving accurate long rollouts with neural pde solvers. Advances in Neural Information Processing Systems, 36, 67398-67433.

[3] Price, I., Sanchez-Gonzalez, A., Alet, F., Andersson, T. R., El-Kadi, A., Masters, D., ... & Willson, M. (2023). Gencast: Diffusion-based ensemble forecasting for medium-range weather. arXiv preprint arXiv:2312.15796.

[4] Chen, Y., Goldstein, M., Hua, M., Albergo, M. S., Boffi, N. M., & Vanden-Eijnden, E. (2024). Probabilistic Forecasting with Stochastic Interpolants and F\" ollmer Processes. arXiv preprint arXiv:2403.13724.

---

> ### Author Rebuttal · Authors · 2025-07-28
>
> We thank the reviewer for their detailed, balanced and constructive feedback, and we are encouraged by their recognition of the potential of our work.
>
> ## Weaknesses:
> **A. “Inappropriate SDA baseline configuration for tasks involving forecasting.”**
>
> The SDA results reported in our paper for Navier-Stokes and SEVIR are already based on the universal amortised SDA variant proposed in [1]. The model is conditioned on previous states (P=10 for NS, P=6 for SEVIR) with varying conditioning scenarios during training. We will rename this baseline to **PDEDiff** for clarity.
>
> For Lorenz-63, we originally used the naive joint-score SDA. We have now added the PDEDiff (P=1, C=1 as FlowDAS) to this setting as well.
>
> We will revise the manuscript to clarify the SDA configurations used and correct any references to joint vs conditional score formulations. Notably, FlowDAS achieves the best performance across most metrics, even compared to BPF, which has access to the exact forward model. While PDEDiff improves upon SDA by modeling conditional scores, it still falls a bit short of FlowDAS.
>
> | Metric                          | FlowDAS | SDA     | PDEDiff | BPF    |
> |:-------------------------------|:-------:|:-------:|:-------:|:------:|
> | $\log p(x\_{2:K} \| x\_1)$           | 17.29   | -332.7  | 8.156   | 17.88  |
> | $\log p(y \| x\_{1:K})$             | -0.228  | -6.112  | -1.977  | -1.572 |
> | W1 (Trajectory)                | 0.106   | 0.528   | 0.516   | 0.812  |
> | RMSE                           | 0.202   | 1.114   | 0.871   | 0.270  |
>
> **B. “Lack of computational cost analysis”**
> Response:
> 1). Regarding the Discretization steps, our stochastic interpolant uses N = 500 time steps during sampling. We will add this to the implementation details section.
> 2). We have now added quantitative runtime and parameter comparisons (for the weather forecasting task using a single A100 GPU) to the appendix.
>
> | Model       | Params (M) | Total Training Time (hr) | Inference Time per Sample (s) |
> |:-----------:|:----------:|:-------------------------:|:------------------------------:|
> | FlowDAS     | 30.0       | 22                        | 30.9                           |
> | SDA         | 22.9       | 45                        | 28.9                           |
> | Transolver  | 11.2       | 80                        | 12.7                           |
> | FNO         | 1880.0     | 80                        | 2.4                            |
>
> The slightly longer inference time for FlowDAS stems from its iterative correction process, which is essential for achieving accurate and observation-consistent predictions. While this leads to higher per-sample inference time, it is balanced by significant **training efficiency**, as FlowDAS reaches convergence much faster than all competing baselines. We believe this reflects a favorable trade-off in practical deployment scenarios. A more detailed discussion will be included in the final version.
>
> To further improve inference speed, one promising direction is to apply **post-hoc distillation**, where a faster surrogate model is trained to approximate the FlowDAS sampler. Another avenue is to explore hybrid formulations that **integrate variational inference** [5] techniques into the framework, potentially offering both computational gains and improved flexibility.
>
> **C. “Ablation on conditioning horizon”**
>
> To assess the impact of the conditioning horizon, we conducted a controlled ablation using the CRPS metric on both Navier–Stokes (NS) and SEVIR experiments by varying the number of observed previous steps used during inference. The results are summarized below:
>
> * **Navier–Stokes CRPS vs. Conditioning Horizon**
>
> | #Conditioning States | 10    | 6     | 3     | 1     |
> |:----------------:|:-----:|:-----:|:-----:|:-----:|
> | CRPS             | 0.538 | 0.634 | 0.663 | 0.776 |
>
> * **SEVIR CRPS vs. Conditioning Horizon**
>
> | #Conditioning States | 6     | 3     | 1     |
> |:----------------:|:-----:|:-----:|:-----:|
> | CRPS             | 0.015 | 0.021 | 0.021 |
>
> These results clearly show that incorporating a longer observation history leads to consistent and significant improvements in forecasting performance. In particular, we observe a **~30% CRPS reduction in NS** when increasing the conditioning length from 1 to 10 steps. Similarly, in SEVIR, although the absolute values are small due to the scale of the task, conditioning on 6 previous frames led to the best predictive accuracy. This supports our claim that incorporating temporal context is important for accurate data assimilation, especially in high-dimensional forecasting tasks.
>
>
> **D. “Missing limitations discussion”**
>
> To improve clarity, we will include an explicit **Limitations and Future Work** section in the revised manuscript.
>
> > **Limitations and Future Work**
>
> > FlowDAS has so far been validated on controlled dynamical systems (e.g., Lorenz-63, Navier–Stokes), and its generalizability to more complex, real-world settings such as numerical weather prediction remains to be explored. In particular, we have not yet studied its performance under Sim2Real conditions or applied it to operational-scale, high-dimensional systems like those addressed by GEN_BE.
>
> > Although FlowDAS supports probabilistic inference via sampling, the current sampling speed remains a limitation. One possible solution is to apply post-hoc distillation, where a faster surrogate model is trained to mimic the FlowDAS sampler. Exploring hybrid formulations that incorporate variational inference techniques could be another promising direction to improve efficiency.
>
> **E. Incomplete coverage of a few related methods**
>
> 1. (**Better coverage of [4]**) The reviewer is correct that our Navier-Stokes experiment follows the setup established in [4], and we will certainly add this citation in the relevant section.
> To further clarify the context of our work, we will also expand our discussion in Related Work section to explicitly describe how FlowDAS builds upon and extends such autoregressive forecasting approaches by incorporating an observation-driven data assimilation component. To reflect this, we plan to revise the paragraph talking about ‘Diffusion model-based methods’ in Section 2 into ‘Generative model-based methods’ and **insert the following paragraph** discussing [4]:
>
> > A closely related line of work is the stochastic interpolant-based forecasting framework recently proposed in [4], which formulates the prediction task as conditional sampling of future system states given the current state, using a fictitious, non-physical stochastic process governed by a learned SDE. While this approach enables probabilistic forecasting, it does not incorporate observation-driven corrections during inference. In contrast, FlowDAS augments the stochastic interpolant dynamics with a measurement-consistent correction term at each step, enabling integration of observational data and thus unifying generative modeling with data assimilation.
>
> 2. (**non-autoregressive vs. autoregressive diffusion models**) We will revise the sentence in the related work section to reflect this more balanced view:
>
> > Additionally, non-autoregressive diffusion models may struggle with long sequence forecasting in high-dimensional systems, though autoregressive diffusion approaches, like GenCast [3], have shown promising results in such settings.
>
> **F. Typographical and notational errors**
>
> We corrected the notation errors. These edits will be included in the final version.
>
> &nbsp;
>
> ## Questions
> **A. “FlowDAS provides clearer physical insight into one-step dynamics?”**
>
> We agree that the phrase 'provides clearer physical insight' could imply an unintended comparison. Our intention was simply to state that our method offers insights that are physically clear. To better reflect this, we will revise the phrasing to '**provides clear physical insight**' and ensure this meaning is unambiguous.
>
> **B. “For the SDA baseline, … a few steps of Langevin sampling …, is this something the authors have also employed in the experiments?”**
>
> **Yes**, we apply this predictor-corrector (PC) sampling settings proposed in the SDA baseline. Specifically, the predictor step is 512 and the corrector step is 3. As we empirically find, these settings show a good balance between the good performance and computation burden. The settings are kept across the Lorenz, NS, SEVIR experiments.
>
> &nbsp;
>
> ## Limitations:
> **A. “Inclusion of several previous time steps -> large memory requirements”**
>
> We agree that memory usage can grow with the number of input states, especially in high-dimensional settings such as medium-range weather forecasting. In fact, in our ongoing collaboration with an industry partner, we are applying FlowDAS to real-world meteorological data with shape (71, 181, 360), where 71 is the number of channels. This indeed is significantly higher than in our current Navier–Stokes and SEVIR experiments.
>
> To address this, a promising direction we are actively exploring is lifting FlowDAS into a **latent space**, inspired by works such as SLAMS [6], which can substantially reduce computational and memory overhead while preserving dynamics. We will add this as a concrete point in the discussion of future work.
>
> **B. “Costly conditioning if the likelihood is costly to evaluate”**
>
> We agree with the reviewer. In our current experiments, the likelihood evaluation is relatively cheap, so **this is not a bottleneck**. For settings where the likelihood is expensive to compute, developing approximated likelihood evaluations would be a promising direction for future work.
>
> &nbsp;
>
> ## Reference
> [5] Morteza Mardani, Jiaming Song, Jan Kautz, and Arash Vahdat. A variational perspective on solving inverse problems with diffusion models. arXiv preprint arXiv:2305.04391, 2023.
>
> [6] ​​Qu, Yongquan, et al. "Deep generative data assimilation in multimodal setting." Proceedings of the IEEE/CVF Conference on Computer Vision and Pattern Recognition. 2024.

---

> > ### Comment · Reviewer_Ztob · 2025-08-02
> >
> > Thank you for addressing the points raised in my review.
> >
> > A. Understood. Please ensure that the baselines are clearly identified in the revised manuscript (e.g., explicitly referring to PDEDiff rather than SDA) to avoid any potential confusion.
> >
> > B. Thank you for the additional details regarding computational cost. I was somewhat surprised by the large number of diffusion steps required, which results in relatively high inference time per sample. While data assimilation is indeed a different task from forecasting, it is worth noting that most recent diffusion-based methods in the forecasting literature (e.g. GenCast) typically use only a few tens of sampling steps rather than several hundred. Part of this efficiency comes from adopting the EDM formulation [1], and this is without any form of distillation.
> > Out of curiosity: have the authors experimented with a smaller number of steps, such as $N=50$? If so, was the performance significantly worse?
> >
> > Point regarding "provides clear physical insight'" --- I appreciate the revision, but I still find it unclear what specific physical insights are enabled by the stochastic interpolant framework that are not accessible through diffusion models. What aspect of physics can be studied under your framework and cannot be studied, for example, with PDEDiff?
> >
> > Thank you again for the clarifications and for outlining the changes you plan to make in the revised manuscript.
> >
> > [1] Karras, Tero, et al. "Elucidating the design space of diffusion-based generative models." Advances in neural information processing systems 35 (2022): 26565-26577.

---

> ### Author Response · Authors · 2025-08-04
>
> We thank the reviewer for the thoughtful and constructive discussions, these have significantly contributed to improving the quality and clarity of our work.
>
> A. “Baseline configs.”
> We will make sure the baselines are clearly identified in the revised manuscript to avoid confusion.
>
> B. “Additional details on computational cost.”
> 1. Empirical results with fewer sampling steps
> We experimented with reduced sampling steps from 1000 down to 50 on SEVIR. Results are summarized below:
>
> | Number of steps | Time (s) | RMSE ± Std     |
> |-----------------|----------|----------------|
> | 1000            |   66.6   |  0.059   ± 0.005 |
> | 500             | 30.9     | 0.056 ± 0.003  |
> | 450             | 27.5     | 0.055 ± 0.003  |
> | 400             | 24.0     | 0.054 ± 0.002  |
> | 350             | 21.6     | 0.054 ± 0.002  |
> | 300             | 18.0     | 0.054 ± 0.002  |
> | 250             | 14.9     | 0.053 ± 0.002  |
> | 200             | 12.3     | 0.054 ± 0.002  |
> | 150             | 9.1      | 0.058 ± 0.001  |
> | 100             | 6.2      | 0.060 ± 0.001  |
> | 50              | 3.0      | 0.062 ± 0.001  |
>
> While N = 200–400 achieves similar RMSE, we adopt N = 500 to maintain sampling **diversity**, which is essential for detecting rare or extreme events in weather assimilation tasks.
>
> 2. Why we use an SDE-based sampler instead of an EDM (ODE-based) sampler
>
> SDE and ODE samplers offer different trade-offs in generative modeling. ODE-based samplers (e.g., EDM) are deterministic and typically more efficient, requiring fewer steps for high-quality synthesis. SDE-based samplers introduce stochasticity during inference, which is beneficial for capturing uncertainty and producing diverse samples; this is especially important for data assimilation under fixed observations.
> Currently, FlowDAS is built upon the stochastic interpolants framework, which naturally defines a reverse-time SDE for sampling. We acknowledge the practical advantages of ODE-based sampling and appreciate the suggestion. We have not explored this direction in FlowDAS yet, but investigating efficient ODE solvers, hybrid methods, or accelerated SDE schemes is a valuable future direction.
>
>
>
> C. “FlowDAS provides clear physical insights.”
>
> For the question on physical insights: FlowDAS directly models transitions between two valid physical states $x\_{t-1}$ to $x\_t$, so the learned velocity field can be interpreted as a data-driven approximation of the underlying dynamics over one time step, where both ends stay on the data manifold. The shorter distance between data states (compared to distance between noise and data manifold for diffusion model based frameworks) also leads to faster convergence during training, making it easier to capture underlying physics.
>
> Regarding physical scenarios where diffusion-based frameworks may struggle: in our double-well experiment (Appendix D.3), a multi-modal system with a nonlinear observation model, noise-initialized optimization led to unstable gradients and recovery failure. In contrast, FlowDAS starts from a valid state, improving stability and robustness in such settings.

---

> > ### Comment · Reviewer_Ztob · 2025-08-08
> >
> > Thank you for your reply!
> >
> > "Regarding physical scenarios where diffusion-based frameworks may struggle: in our double-well experiment (Appendix D.3), a multi-modal system with a nonlinear observation model, noise-initialized optimization led to unstable gradients and recovery failure. In contrast, FlowDAS starts from a valid state, improving stability and robustness in such settings." --- I do not think it is entirely clear whether this improvement stems from the stochastic interpolant framework itself, or from how conditioning on the nonlinear observation model is performed for the diffusion model. I know the formulation in SDA allows for nonlinear observation models, but since SDA appeared, alternative methods of performing diffusion posterior sampling with nonlinear observation models have appeared which could give better performance.
> >
> > Overall, I consider this a good paper, and provided the authors incorporate the promised modifications, most of my minor concerns will be addressed. However, I am still not fully convinced about the benefits over the related diffusion-based methodologies and I still believe it requires significant computational resources (memory and inference time), these being the reasons why I decided to maintain my rating of 4.

---

### Official Review · Reviewer_3gVf · 2025-07-02

**Clarity:** 4
**Significance:** 4
**Originality:** 4
**Rating:** 6
**Confidence:** 4

**Summary:**

FlowDAS stochastic interpolants to improve upon the Score-based Data Assimilation Model by Rozet. Stochastic interpolants: mapping from noise$ + x_0 \rightarrow x_1$. One progresses the dynamics from $x_t$ to $x_{t+1}$ by learning the score for $P(x_{t+1} | x_t)$. At inference time the drift is augmented by a Bayes-term so every Euler–Maruyama step is nudged toward the incoming observation, giving trajectories that are “analysis-consistent” as they are generated .

**Questions:**

Could variational inference over the interpolant path yield accurate density spread equivalent to the EnKF, or perhaps a better density estimation procedure?

How does performance degrade as the observations become very sparse (for example, lower than 1%)? Would localization need to be incorporated into the method at any point?

**Ethical Concerns:**

["NO or VERY MINOR ethics concerns only"]

**Final Justification:**

After the rebuttal process, I decided to maintain my score as a 6 (Strong accept). The paper is theoretically interesting by bringing in stochastic interpolants into the data assimilation process, and should open up further avenues of discussion for score-based approaches for data assimilation.

**Limitations:**

Yes.

**Paper Formatting Concerns:**

None.

**Quality:**

4

**Strengths And Weaknesses:**

Strengths

The data assimilation step is already integrated into the prediction step. As a result, there is no need to separate out the prediction and analysis step as in standard data assimilation. However, this comes with the drawback that it makes this model harder to integrate into existing predictive architectures for real-world problems, since this model is an all-in-one data assimilation solution. Particularly since current predictive weather models are done at a very fine resolution, then FlowDAS would need to scale well in order for the predictive model to also compete.

Training is stable and interpretable, in contrast to global noise-to-data diffusion chains by taking advantage of stochastic interpolants. The progression of the stochastic interpolants makese it so that the model starts off at the first point with no noise, introduces stochasticity in the middle, and then results in the target.

FlowDAS offers significantly improved results over SDA, which has a similar framework but only aims to generate a state trajectory conditioned on the observations. In addition, this opens up FlowDAS to be well integrated with the autoregressive data assimilation framework compared to the full trajectory generation in SDA.

The use of stochastic interpolants in data assimilation has not been explored before, making this paper interesting as it takes the first steps towards what seems to be a promising avenue of exploration.

Weaknesses

Stochastic integrators introduce bias; accuracy–speed trade-offs under very stiff dynamics remain to be quantified.

While each step is stochastic, the framework still produces a single trajectory unless multiple MC traces are run which makes uncertainty quantification difficult.

Summary

Given its multitude of strengths, I am leaning towards a strong accept for this paper. I feel like this is a strong technical paper with good results whose approach would be interesting to the NeurIPS community.

---

> ### Author Rebuttal · Authors · 2025-07-28
>
> First of all, we sincerely thank the reviewer for the positive evaluation and encouraging comments, as well as for the insightful suggestions that helped us further improve the paper! :)
>
> ## Weakness
> **A. “Stochastic integrators introduce bias; accuracy-speed trade-offs under very stiff dynamics ...”**
>
> 1). **Accuracy-speed trade-offs** Indeed, the stochastic integrators in Equations (9) and (10) trade exactness for efficiency and introduce numerical bias. To evaluate this trade-off, we compared three settings: first-order integrator, second-order integrator, and a hybrid method that computes the full stochastic interpolant path to the endpoint $X\_1$ for the final 100 steps (“2nd order × 400 + Flow × 100”). As shown in the table (now included in Appendix D), **the full-path method increases computation time by 7× and peak memory usage by 18× compared to the second-order integrator, while improving RMSE by only ~0.1%.**
>
> We initially considered the full Flow*500 method, i.e., computing $X\_1$ exactly at every step by fully rolling out the leftward stochastic interpolant, but this would require about 500+(500+499+...+1) $\approx$ 250,000 forward passes and retaining the entire computation graph at each step. This would cause out-of-memory failures even on A100 GPUs. This substantial cost arises because computing the full-path endpoint requires storing the entire interpolation path to backpropagate $\log p(y|X\_1)$. In contrast, the 1st- and 2nd-order methods yield stable, efficient inference while maintaining competitive accuracy.
>
> We believe this experiment validates our design choice: the benefit of observation conditioning significantly outweighs the small bias introduced by the integrator. We will clarify this point in the final version and include the trade-off analysis and table in the appendix.
>
>
> |             Method              | Computation Time |      RMSE       | Max Memory Used |
> |:------------------------------:|:----------------:|:---------------:|:----------------:|
> |         1st order              |     17.100 s     | 0.0490 ± 0.0026   |     1.8 G     |
> |         2nd order              |     30.871 s     | 0.0485 ± 0.0035   |     3.9 G     |
> | 2nd order × 400 + Flow × 100   |    205.271 s     | 0.0478 ± 0.0033   |    70.0 G     |
>
>
> 2). **More stiff dynamics** In our experiments on the Navier–Stokes system, known for its stiffness and chaotic behavior, we have observed stable rollout behavior and robust error control. While we do not explicitly tailor our integrators for stiffness (e.g., via implicit solvers or adaptive step size control), the **short-step nature of the interpolant and the sequential correction mechanism naturally mitigate stiffness-related instabilities**.
>
> In addition, we are currently extending FlowDAS to a real-world ocean dynamics application, where the governing equations follow the **Quasi-Geostrophic (QG) model**. These systems exhibit more complex behavior, such as inverse energy cascades and long-range spatial correlations, making them even more challenging than standard Navier–Stokes in certain regimes. Our preliminary results indicate that FlowDAS still performs well under this setting, though further gains are likely possible through domain-specific strategies tailored to QG-type dynamics. We view this as a promising direction for advancing the applicability of FlowDAS to operational-scale geophysical systems.
>
> **B. “Difficult uncertainty quantification...”**
>
> While FlowDAS generates a single trajectory per sampling run, its stochastic formulation naturally supports uncertainty quantification via ensemble sampling. In our weather forecasting task, we already report ensemble standard deviation as a measure of uncertainty (see Table S.10). For the Navier–Stokes experiments, we have similarly conducted Monte Carlo sampling to quantify uncertainty and will include the ensemble standard deviation in the updated version of the paper. This is done by running multiple inference trajectories under the same observation setting and computing the spread of predictions. As shown in the table below, the uncertainty increases as the observation becomes sparser or lower in resolution, which aligns with expectations.
>
> |       RMSE ± Std        |   SR 4x     |  SR 8x     |  SO 5%     | SO 1.5625%  |
> |:-----------------------:|:-----------:|:----------:|:----------:|:-----------:|
> |                         | 0.038 ± 0.001 | 0.067 ± 0.050 | 0.071 ± 0.005 | 0.123 ± 0.010 |
>
> This ensemble-based strategy provides a practical and interpretable way to assess uncertainty within the FlowDAS framework.
>
> &nbsp;
>
> ## Questions
> **A. “Could variational inference over the interpolant path yield accurate density spread...”**
>
> Thank you for this insightful question. Since FlowDAS is based on MCMC-style sampling, it is natural to ask whether variational inference (VI), a widely used alternative in inverse problems and posterior approximation, could provide comparable or even improved density estimates. We address this in three parts:
>
> - **Can VI be integrated into the FlowDAS framework?**
>   **Yes.** Following RED-Diff [1], we can construct a RED-FlowDAS variant. All theoretical components generalize naturally to the stochastic interpolant setting (see our brief derivation).
>
> - **How does VI compare to EnKF and FlowDAS?**
>   * Compared to **EnKF**, VI is more flexible: it accommodates general likelihoods (not limited to Gaussian noise) and models complex state transitions via KL divergence minimization, which goes beyond EnKF’s linear assumptions.
>   * Compared to **FlowDAS**, VI provides an explicit density estimate, while FlowDAS samples from the posterior and thus models the posterior distribution indirectly. However, MCMC may better explore complex multi-modal posteriors, whereas VI can be mode-seeking due to the nature of KL divergence.
>
> - **Additional benefits of VI:**
>   VI avoids the need for conditional score estimation, can be faster than MCMC, and is amenable to stochastic and distributed optimization. These advantages make it a promising direction for extending FlowDAS.
>
>
> **B. “Performance degradation under challenging cases & Localization?”**
>
> 1). **Performance degradation under extremely challenging cases** We have conducted additional experiments using super-resolution (SR) and sparse-observation (SO) settings with as little as **0.75%** coverage. The results, shown below, indicate a **graceful degradation** of performance as the amount of observed information decreases. This supports the robustness of FlowDAS even under highly limited measurement scenarios.
>
> |     RMSE ± Std     |   SR 16x     |   SO 1%     |  SO 0.75%   |
> |:------------------:|:------------:|:-----------:|:-----------:|
> |                    | 0.108 ± 0.010 | 0.148 ± 0.014 | 0.177 ± 0.019 |
>
> These results will be included in the updated appendix.
>
> 2). **Localization** Unlike ensemble-based filters such as EnKF, which rely on covariance estimation and therefore require localization to suppress spurious long-range correlations, **FlowDAS does not estimate or use a background covariance matrix**. Instead, it performs observation-conditioned updates directly through a likelihood-based correction at each step, guided by the measurement operator A. This naturally confines the influence of observations to their spatial support, eliminating the need for explicit localization.
>
> &nbsp;
>
> ## (**Optional**) Brief Derivation for VI-FlowDAS
> For completeness, we include a brief derivation showing how variational inference can be integrated into FlowDAS.
>
> We could format the data assimilation question in a VI settings and that is we want to fit a family of densities $q({X}_1 \mid {X}_0,{Y})$ to a intractable posterior distribution $P({X}_1 \mid {X}_0,{Y})$, where the ${X}_1$ is the state variable we want to predict, and ${X}_0$ is the current state and ${Y}$ is the observation on ${X}_1$. And the variation method is based on  KL minimization
>
> $
>     \min_q KL\big(q({X}_1 \mid {X}_0,{Y}) || P({X}_1 \mid {X}_0,{Y}))
> $
>
> where $q := \mathcal{N}(\mu, \sigma^2 I)$ is a variational distribution. And this objective can be expand:
>
> $
> KL\big(q({X}\_1 \mid {X}\_0,{Y}) \| P({X}\_1 \mid {X}\_0,{Y})\big) = -{E}\_{q({X}\_1 \mid {X}\_0,{Y})}\big[\log P({Y} \mid {X}\_1,{X}\_0)\big]\+KL\big(q({X}\_1 \mid {X}\_0,{Y})\| p({X}\_1 \mid {X}\_0)\big) + \log p({Y},{X}\_0)
> $
>
> The last term is the observation likelihood that is **constant** w.r.t.~$q$. Thus, to minimize the KL divergence, it suffices to minimize the variational bound (omit the last term). This brings us to the next claim.
>
> ${Proposition 1}$. The KL minimization is equivalent to minimizing the variational bound, that itself obeys the **velocity matching loss** in the stochastic interpolants framework:
>
> $
> \min\_{\{\mu, \sigma\}}-{E}\_{q({X}\_1 \mid {X}\_0,{Y})}\left[\log P({Y} \mid {X}\_1,{X}\_0)\right] +\int_{0}^{1} \frac{1}{2\sigma\_s^{2}} {E}\_{q({X}\_s \mid {X}\_0,{Y})} \Big[\big\|\hat{b}^{q}\_s({X}\_s, {X}\_0) - \hat{b}^{p}\_s({X}\_s, {X}\_0) \big\|^2_2 \Big] ds
> $
>
>  where $q({X}_s | {X}_0,{Y}) = \mathcal{N}( \alpha_s {X}_0 + \beta_s \mu, (\beta_s^2 \sigma^2 + \sigma_s^2 s)I)$ and $\hat{b}^{q}_s({X}_s, {X}_0)$ can be explicitly computed:
>
> $
> \hat{b}^{q}_s({X}_s, {X}_0) = \dot{\alpha}_s{X}_0 + \dot{\beta}_s{X}_1^{q}+\dot{\sigma}_s{W}_s
> $
>
> where ${X}_1^{q}$ is produced by sampling from $q({X}_1 \mid {X}_0,{Y}) := \mathcal{N}(\mu, \sigma^2 I)$.
>
> For the $\hat{b}^{p}_s({X}_s, {X}_0)$, this is exactly the stochastic model we trained.
>
> &nbsp;
>
> ## Reference
>
> [1]. Morteza Mardani, Jiaming Song, Jan Kautz, and Arash Vahdat. A variational perspective on solving inverse problems with diffusion models. arXiv preprint arXiv:2305.04391, 2023.

---

> > ### Comment · Reviewer_3gVf · 2025-08-03
> >
> > I think the authors for their thorough reply to my questions. I would like to reiterate my support for this paper (already a 6).

---

> > > ### Author Response · Authors · 2025-08-04
> > > **Acknowledgment to Reviewer 3gVf**
> > >
> > > Thank you very much for your support and for taking the time to read our responses. We really appreciate your encouraging feedback!

---

### Official Review · Reviewer_Qtzs · 2025-07-06

**Clarity:** 2
**Significance:** 3
**Originality:** 2
**Rating:** 4
**Confidence:** 3

**Summary:**

The paper presents a data assimilation framework based on a stochastic interpolator introduced in reference [30], used to model probabilistic transitions between state vectors across time steps. The method considers short-step conditional transitions, rather than learning a global noise-to-data map. Observations are incorporated at each interpolation step through a drift correction term. Without observations it reduces to autoregressive probabilistic forecasting. It is evaluated on benchmarks Lorenz-63, SEVIR, Navier–Stokes, and a weather forecasting task.

**Questions:**

The paper addresses an important problem and presents an interesting application of stochastic interpolators to data assimilation. However, the methodological contribution appears incremental. The framework is similar to existing approaches, as it extends a data assimilation approach that employs score-based diffusion models of reference [28] and stochastic interpolators in reference [31]. The main distinction seems to be the use of stochastic interpolators instead of posterior sampling diffusion models for modeling short-step conditional transitions, but the contribution appears somewhat incremental.

The overall presentation of the method could be more streamlined. The proposed method in Section 3 is presented based on modifications of components of the stochastic interpolator in [31] to incorporate observation conditioning. However, the presentation focuses heavily on these adaptations, rather than a self-contained method.

The implementation details in Section 3.2 are short, with details only in Appendix B. This makes the inference steps difficult to understand from the main paper.

The inconsistent use of uppercase and lowercase symbols in the notation makes the derivations more difficult to follow.

The paper would benefit from a more thorough discussion of computational efficiency. Runtime, scalability, and comparisons to baseline methods are not sufficiently addressed.

**Ethical Concerns:**

["NO or VERY MINOR ethics concerns only"]

**Final Justification:**

After reading the reviews and the responses, I appreciate the additional comparisons in terms of running time, the comparisons with traditional algorithms such as EnKF, and the clarifications. However, the contribution still appears somewhat incremental compared to works such as SDA and PDEDiff. I am raising my score accordingly to 4.

**Limitations:**

There are no significant potential negative societal impact.

**Paper Formatting Concerns:**

No formatting concerns.

**Quality:**

2

**Strengths And Weaknesses:**

Strengths:
- The paper addresses an interesting and timely problem, with applications to high-impact domains such as weather forecasting.

Weaknesses:
- The contribution appears somewhat incremental, as it builds on similar existing work on stochastic interpolators and diffusion models applied to data assimilation.
- The inference procedure is not presented very clearly, and some parts are presented as an extension of stochastic interpolators.

---

> ### Author Rebuttal · Authors · 2025-07-28
>
> First of all, we sincerely thank the reviewer for their valuable feedback and constructive suggestions, which have helped us improve the clarity, accuracy, and completeness of the paper!
>
> &nbsp;
>
> **A. “The contribution appears somewhat incremental.”**
>
> Response:
>
> We would like to highlight the novelty and significance of our contributions. FlowDAS is **not a simple application of existing stochastic interpolants or diffusion models**, but a principled and technically novel framework for solving the **scientifically important problem of data assimilation (DA)**—which differs fundamentally from pure forecasting.
>
> - **Data assimilation is harder and more scientifically meaningful than pure forecasting.** Unlike forecasting, which predicts future states given the present, data assimilation requires reconstructing a latent trajectory given *partial and noisy observations over time*. This is critical in real-world systems such as weather prediction and oceanography, etc. It demands algorithms that are *measurement-consistent*, temporally coherent, and robust under sparse observations, that FlowDAS directly addresses.
>
> - **Beyond existing stochastic interpolant (SI) methods.** Prior SI methods (e.g., [31]) focus exclusively on probabilistic forecasting and cannot incorporate observations. FlowDAS makes the first extension of SI to the DA setting by introducing:
>   (1) A **likelihood-based drift correction** (Eq. 7) that enables measurement conditioning during rollout;
>   (2) An **unbiased estimator for the gradient of the observation likelihood** (Eq. 8); and
>   (3) A **Monte Carlo sampling accelerator using stochastic integrators** (Eqs. 9–10) for efficient inference.
>   (4) A **flexible backbone architecture**, where using neural operators (e.g., FNO) instead of UNet yields even better results (see Table S.11, to be moved into main text).
>
>   These advances together enable **observation-conditioned generation**, which has not been addressed in prior SI literature.
>
>
> - **Advantages over diffusion-based approaches.** FlowDAS offers key improvements over diffusion-based methods like SDA and PDEDiff [1]. **SDA** models the dynamics via a joint score over entire trajectories, which limits temporal resolution and interpretability. **PDEDiff** improves on this by using conditional scores for state transitions, making it closer in spirit to FlowDAS. However, both methods require **sampling from Gaussian noise**, which leads to instability (see our toy double-well experiment in Appendix D.3), reduced physical interpretability, and degraded performance as shown in all experiments. In contrast, FlowDAS performs direct **state-to-state transitions** with an explicit transition density $p(x\_{k+1} \mid x\_k, y\_{k+1})$, enabling more stable, efficient, and physically grounded inference.
>
> Together, these contributions establish FlowDAS as the **first generative data assimilation framework based on stochastic interpolants**. It represents a substantial and necessary step forward for building physically grounded, observation-aware generative models for scientific applications.
>
>
> &nbsp;
>
>
> **B. “The inference procedure is not presented very clearly.”**
>
> Thank you for this valuable feedback. We agree that a more detailed explanation of the inference procedure would enhance the clarity of the main paper. While we had placed a full, detailed breakdown in Algorithm 2 (Appendix B) due to initial space limitations, we will incorporate it directly and insert the following description into Section 3.2 in the revision:
>
>   **Inference** Given a trained drift model $\hat{b}_s$, FlowDAS performs inference in an autoregressive manner to estimate the latent state trajectory $ \{ \hat{x}\_{1:K} \}$. At each time step $k$, we begin from the known state $\hat{x}_k$ and predict the next state $\hat{x}\_{k+1}$ using a discretized stochastic interpolant over a time grid $s_0 = 0 < s_1 < \cdots < s_N = 1$. We set the interpolant state $X\_{s\_0} \gets \hat{x}\_k $ and retrieve the next observation $ y\_{k+1} $. For each interpolation step $ s\_n$, we simulate a forward transition using the learned drift, given in Eq.7, and noise. To enforce observation consistency, we generate $J$ posterior endpoint samples $ \hat{X}\_1^{(j)}$, and compute their likelihoods under the measurement model, i.e., $ || y - \mathcal{A}(\hat{X}\_1^{(j)}) ||_2^2 $. A softmax over these values gives importance weights $ \{ w\_{1:J} \}$. We then apply a correction to the interpolant using a weighted gradient $- \zeta\_n \nabla\_{X\_{s\_n}} \sum \_{j=1}^J w\_j || y - \mathcal{A}(\hat{X}\_1^{(j)}) ||_2^2$, where $\zeta\_n$ denotes the step size. After reaching $s=1$, we set $ \hat{x}\_{k+1} \gets X\_{s\_N} $ and proceed to the next time step. Repeating this procedure over all $ k $ yields the full trajectory.
>
> &nbsp;
>
> **C. “The presentation focuses heavily on these adaptations, rather than a self-contained method.”**
>
> We would like to hightlight that **data assimilation naturally consists of two essential components**: a probabilistic forecasting model and an observation-conditioned correction mechanism. Prior work such as [31] addresses only the forecasting part. To clearly situate our contribution, we chose to introduce that work early on, but this does not mean FlowDAS is not self-contained.
>
> FlowDAS is a complete generative DA method. In the final version, we will revise Section 3 to present FlowDAS more coherently as a standalone algorithm, and move contextual references to prior work into a separate discussion in the Section 2 to improve readability.
>
> &nbsp;
>
> **D. “The inconsistent use of uppercase and lowercase symbols in the notation makes the derivations more difficult to follow.”**
>
> Our notation intentionally distinguishes between random variables and their realizations:
> - Uppercase letters (e.g., $X, I, R$) denote random variables or stochastic processes.
> - Lowercase letters (e.g., $x, y$) represent specific realizations or observations.
>
> For example, $X\_0, X\_1 \sim p(X\_0, X\_1)$ are random variables, while $x\_0, x\_1$ are concrete samples. This convention is common in probabilistic modeling and used consistently throughout. In the final version, we will add an explicit note at the start of Section 2 to make this distinction clear to readers.
>
>
> &nbsp;
>
> **E. "The paper would benefit from a more thorough discussion of computational efficiency. Runtime, scalability, and comparisons to baseline methods are not sufficiently addressed."**
>
> We agree with the reviewer on the importance of computational efficiency and have added a detailed performance comparison in the appendix. The results, using a single A100 GPU on the SEVIR dataset, highlight FlowDAS’s practical viability:
>
> | Model       | Params (M) | Total Training Time (hr) | Inference Time per Sample (s) |
> |:-----------:|:----------:|:-------------------------:|:------------------------------:|
> | FlowDAS     | 30.0       | 22                        | 30.9                           |
> | SDA         | 22.9       | 45                        | 28.9                           |
> | Transolver  | 11.2       | 80                        | 12.7                           |
> | FNO         | 1880.0     | 80                        | 2.4                            |
>
> The slightly longer inference time for FlowDAS stems from its iterative correction process, which is essential for achieving accurate and observation-consistent predictions. While this leads to higher per-sample inference time, it is balanced by significant **training efficiency**, as FlowDAS reaches convergence much faster than all competing baselines. We believe this reflects a favorable trade-off in practical deployment scenarios. A more detailed discussion will be included in the final version.
>
> To further improve inference speed, one promising direction is to apply **post-hoc distillation**, where a faster surrogate model is trained to approximate the FlowDAS sampler. Another avenue is to explore hybrid formulations that **integrate variational inference** [2] (as suggested by Reviewer 3gVf) techniques into the framework, potentially offering both computational gains and improved flexibility.
>
> &nbsp;
>
> ## Reference
> [1] Shysheya, A., Diaconu, C., Bergamin, F., Perdikaris, P., Hernández-Lobato, J. M., Turner, R., & Mathieu, E. (2024). On conditional diffusion models for PDE simulations. Advances in Neural Information Processing Systems, 37, 23246-23300.
>
> [2] Morteza Mardani, Jiaming Song, Jan Kautz, and Arash Vahdat. A variational perspective on solving inverse problems with diffusion models. arXiv preprint arXiv:2305.04391, 2023.

---

> > ### Comment · Reviewer_Qtzs · 2025-08-04
> >
> > I thank the authors for the detailed responses. After reading the reviews and the responses, I appreciate the additional comparisons in terms of running time, the comparisons with traditional algorithms such as EnKF, and the clarifications. However, the contribution still appears somewhat incremental compared to works such as SDA and PDEDiff. I am raising my score accordingly to 4. I encourage the authors to discuss the challenges with uncertainty quantification, including computational efficiency impact of sampling.

---

> > > ### Author Response · Authors · 2025-08-04
> > > **Acknowledgement to Reviewer Qtzs**
> > >
> > > Thank you for taking the time to revisit our paper and responses. We truly appreciate the updated score and your thoughtful comments. Your points about uncertainty quantification and the computational cost of sampling are important, and we will definitely keep them in mind as we move this work forward.

---

### Comment · Area_Chair_X5yn · 2025-08-03

Dear authors and reviewers,

First of all, thank you all for your efforts so far. The author-reviewer discussion period will end on August 6.

@Authors: If not done already, please answer all questions raised by the reviewers. Remain factual, short and concise in your responses, and make sure to address all points raised.

@Reviewers: Read the authors' responses and further discuss the paper with the authors if necessary. In particular, if the concerns you raised have been addressed, take the opportunity to update your review and score accordingly. If some concerns remain, or if you share concerns raised by other reviewers, please make sure to clearly state them in your review. In this case, consider updating your review accordingly (positively or negatively). You can also maintain your review as is, if you feel that the authors' responses did not address your concerns.

I will reach out to you again during the reviewer-AC discussion period (August 7 to August 13) to finalize the reviews and scores.

The AC

---

> ### Author Response · Authors · 2025-08-03
>
> Dear AC,
>
> Thank you for your timely reminder and the efforts for coordinating the review process.
>
> We sincerely appreciate the reviewers’ time and effort in considering our rebuttal and providing their valuable feedback. During the rebuttal period, we did our best to address the raised concerns and questions.
>
> Now, we are working on addressing reviewers' follow-up questions and will respond shortly.
> Moreover, we are eager to clarify any remaining questions and address any concerns they may have.
>
> Best regards,
>
> The Authors

---

> ### Comment · Area_Chair_X5yn · 2025-08-08
>
> Dear reviewers,
>
> The reviewers-authors discussion phase will end in less than 24 hours.
>
> If not done already, make sure to submit the "Mandatory Acknowledgement" that confirms that you have read the reviews, participated in the discussion, and provided final feedback in the "Final justification" text box.
>
> Be mindful of the time and efforts the authors have invested in answering your questions and at least acknowledge their responses. Make sure to provide a fair and scientifically grounded review and score. If you have changed your mind about the paper, please update your review and score accordingly. If you have not changed your mind, please provide a clear and sound justification for your final review and score.
>
> Best regards,
> The AC

---

### Note · Authors · 2025-08-14

Dear AC and Reviewers,

We thank all reviewers for their time, constructive feedback, and active engagement. We are pleased that our rebuttal led Reviewer Qtzs to raise their score, and encouraged by the strong support from Reviewer 3gVf, the positive assessments from Reviewers Ztob and CZwM, and the acknowledgement of our methodological innovation from Reviewer RCK2.

&nbsp;

### Significance

FlowDAS is, to our knowledge, the first data assimilation framework built on stochastic interpolants that learns step-to-step state transitions and conditions on observations during rollout. It handles nonlinear observation operators and non-Gaussian noise, shows strong accuracy on Lorenz-63, Navier–Stokes, and SEVIR weather tasks, and offers a stable and interpretable training objective tied to one-step dynamics.

&nbsp;

### Concerns addressed during rebuttal

During the review process, we have addressed the following major concerns:

* *Computational cost*. Added detailed training/inference comparisons across all baselines, showing FlowDAS trains efficiently and matches baselines in inference while improving accuracy.

* *Baseline verification and additions*. Clarified baseline configurations (renaming SDA to PDEDiff where appropriate) and added EnKF and EnSF to broaden comparisons with classical filtering methods.


* *Metrics and methodological clarity*. Expanded the metrics section to explain the role of the Wasserstein distance, reported CRPS as a proper scoring rule, and provided ablations on conditioning horizon and sampling steps. Also added an explicit Limitations section, extended Related Work, and corrected notation and typographical issues.



&nbsp;

### Revisions to be included in the paper

In addition to addressing the major concerns, we will further enhance the final version by:

* Uncertainty quantification. We will integrate the additional ensemble-based results (including standard deviations in all our experiments) into the main text, ensuring that the methodology and implications for uncertainty estimation are clearly described.


* Theoretical scope. We will clarify in the revised paper that FlowDAS does not sample directly from the full filtering target but instead from a local filtering density consistent with observations, and discuss how this approach remains effective in practice.

&nbsp;


Sincerely,

The Authors

---

### Decision · Program_Chairs · 2025-09-17

**Decision:**

Accept (poster)

**Comment:**

The average rating is 4.4, with all reviewers recommending acceptance (4, 4, 6, 4, 4). Reviewers all agree on the technical correctness of the paper and on the soundness of the approach, even if some of them judge the contribution to be incremental. The discussion has been constructive and improvements have been made to the paper, including clarifications and additional results.

Recommendation: acceptance.